# Generalized Smoothness in Stochastic Convex Optimization: First- and Zero-Order Methods

## Abstract

This paper is devoted to the study of stochastic optimization problems under the generalized smoothness assumption. By considering the unbiased gradient oracle in *Stochastic Gradient Descent*, we provide strategies to achieve in bounds the summands with exponential objective decrease. In particular, in the case $L_0 = 0$, we obtain in the **convex setup** the iteration complexity: $N = \mathcal{O}\left(L_1 R \log \frac{1}{\varepsilon} + \frac{L_1 c R^2}{\varepsilon}\right)$ for *Clipped Stochastic Gradient Descent* and $N = \mathcal{O}\left(L_1 R \log \frac{1}{\varepsilon}\right)$ for *Normalized Stochastic Gradient Descent*. Furthermore, we generalize the convergence results to the case with a biased gradient oracle, and show that the power of $(L_0, L_1)$-smoothness extends to *zero-order algorithms*. Finally, we demonstrate the possibility of the zero-order algorithm outperforming the first-order algorithm in the convex setup through numerical experimentation, which has aroused some interest in the machine learning community – logistic regression problem.

## 1 Introduction

In many real-world scenarios, systems are often noisy and complex, making deterministic optimization infeasible. Therefore, this work focuses on a stochastic optimization problem:

$$f^* = \min_{x \in \mathbb{R}^d} \left\{ f(x) := \mathbb{E}_{\xi \sim \mathcal{D}} \left[ f(x, \xi) \right] \right\}, \tag{1}$$

where $f : \mathbb{R}^d \to \mathbb{R}$ is a convex function. This problem configuration encompasses a broad range of applications in ML scenarios, e.g. empirical risk minimization, where $\mathcal{D}$ denotes distribution across training data points, and $f(x, \xi)$ represents loss of model $x$ on data point $\xi$. We assume that optimization algorithms only have access to the gradient oracle $\mathbf{g} : \mathbb{R}^d \times \mathcal{D} \to \mathbb{R}^d$ with stochastic gradient $\mathbb{E}\left[\nabla f(x, \xi)\right] = \nabla f(x)$ and bias $\mathbf{b}(x)$ terms:

$$\mathbf{g}(x, \xi) = \nabla f(x, \xi) + \mathbf{b}(x). \tag{2}$$

Frequently, to solve problem equation 1 one uses what is likely already a classic optimization algorithm, namely Stochastic Gradient Descent (SGD) (Bottou, 1998) or its variations, which have demonstrated their effectiveness in different settings, for instance, federated learning (Yuan & Ma, 2020; Kairouz et al., 2021; Woodworth et al., 2021), deep learning (Dean et al., 2012; Zhang et al., 2015; Dimlioglu & Choromanska, 2024) and others. Among the variants of SGD, it is worth noting the Normalized Stochastic Gradient Descent (NSGD) (Nesterov, 1984; Hazan et al., 2015; Zhao et al., 2024) which has received widely attention from the community because it addresses challenges in optimization for machine learning (Bengio et al., 1994). And it's also worth noting the Clipped Stochastic Gradient Descent (ClipSGD) (Goodfellow, 2016; Gorbunov et al., 2020), which is commonly used to stabilize the training of deep learning models (Pascanu et al., 2013).

Many standard literatures analyze stochastic optimization algorithms with unbiased gradient oracle equation 2. In particular, SGD (Lacoste-Julien et al., 2012; Bottou et al., 2018), NSGD (Zhao et al., 2021; Hübler et al., 2024a), ClipSGD (Gorbunov et al., 2020; Koloskova et al., 2023). However, there are a number of applications where gradient oracle equation 2 is biased. For example, sparsified SGD (Alistarh et al., 2018), delayed SGD (Stich & Karimireddy, 2019), etc. Zero-order algorithms (Nesterov & Spokoiny, 2017; Demidovich et al., 2023) occupy a special place in the class of stochastic methods with biased gradient oracle equation 2. They are motivated by various applications, including multi-armed bandit (Shamir, 2017; Lattimore & Gyorgy, 2021), online optimization (Agarwal et al.,

2010; Bach & Perchet, 2016; Akhavan et al., 2022), hyperparameter tuning (Hernández-Lobato et al., 2014; Nguyen & Balasubramanian, 2022).

In our work, we investigate the convergence of first-order algorithms: ClipSGD, NSGD, and zero-order algorithms: ZO-ClipSGD, ZO-NSGD, assuming convexity and $(L_0, L_1)$-smoothness, which in the case of twice differentiable functions, states that $\|\nabla^2 f(x)\| \leq L_0 + L_1\|\nabla f(x)\|$.

We emphasize the following points:

**Algorithm step size.** Zero-order algorithms do not have access to the exact (stochastic) gradient in particular, as well as every algorithm with a biased gradient oracle, so we focus on creating first-order methods whose step size does not depend on knowledge of the gradient at a given point. We use the developed first-order algorithms as a basis for creating zero-order methods (Gasnikov et al., 2023).

**Exponential objective decrease.** Historically (Nesterov, 2018), stochastic optimization first- and zero-order algorithms have achieved the desired accuracy with a linear rate of convergence only in strongly convex case and under assumption of standard smoothness. However, the work of (Lobanov et al., 2024b) showed that if the generalized smoothness assumption is satisfied in a _deterministic convex_ optimization problem, then gradient descent has two regimes: exponential objective decrease as long as $\|\nabla f(x^k)\| \geq \frac{L_0}{L_1}$, and a sublinear convergence rate in the other case (see the example of the power of

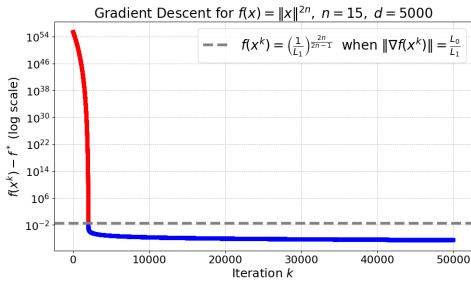

Figure 1: Changing regimes demonstration

norm function in Figure 1). Considering these points, our work answers the following question:

_Can exponential objective decrease in stochastic convex optimization be achieved in terms of iteration complexity for first- and zero-order algorithms with constant step size?_

### 1.1 MAIN CONTRIBUTIONS

More specifically, our contributions are the following:

- We provide strategies to obtain summands with exponential objective decrease. In particular, we show that using clipping or normalization techniques can achieve the desired results.

- We improve convergence results for ClipSGD and NSGD with unbiased gradient oracle equation 2 in the convex setting assuming $(L_0, L_1)$-smoothness (see Table 1). Moreover, we show that in the case $L_0 = 0$, NSGD can converge in the convex setup with exponential objective decrease to the desired accuracy in terms of iteration complexity.

- We generalize ClipSGD, and NSGD to the case of a biased gradient oracle, showing how the bias accumulates over iterations under generalized smoothness assumption.

- We provide the first convergence results for the zero-order algorithms ZO-ClipSGD (Algorithm 3), and ZO-NSGD (Algorithm 4) in the convex and $(L_0, L_1)$-smooth setting. We show that the power of generalized smoothness extends to zero-order methods as well, achieving summands with exponential objective decrease (see Table 1).

- We demonstrate on a numerical example of logistic regression that indeed, zero- and first-order stochastic algorithms can converge with exponential objective decrease in a convex setup. Moreover, we demonstrate the possibility of the zero-order (ZO-NSGD) algorithm outperforming the first-order (ClipSGD) algorithm (see Figure 2).

### 1.2 FORMAL SETTING AND ASSUMPTIONS

In this subsection, we introduce and discuss main assumptions and notations used throughout paper.

**Notations.** We use $\langle x, y \rangle := \sum_{i=1}^d x_i y_i$ to denote standard inner product of $x, y \in \mathbb{R}^d$. We denote Euclidean norm in $\mathbb{R}^d$ as $\|x\| := \sqrt{\sum_{i=1}^d x_i^2}$. In particular, this norm $\|x\| := \sqrt{\langle x, x \rangle}$ is related to the inner product. We use $\mathcal{P}[\cdot]$ to define probability measure which is always known from the context, $\mathbb{E}[\cdot]$ denotes mathematical expectation. We use the following notation $B^d(r) := \{x \in \mathbb{R}^d : \|x\| \leq r\}$

Table 1: Comparison of convergence results of SGD variants to the most related work (Gaash et al., 2025) in the convex and $(L_0, L_1)$-smooth setup. Notation: $\mathbb{E}\left[\|\nabla f(x, \xi)\|^2\right] \leq \tilde{\sigma}^2$; $\eta \leq (L_0 + L_1 c)^{-1}$ – step size; $c > 0$ – clipping radius; $\mathcal{R} = \left(\eta + \frac{L_0 R}{c^2} + \frac{R}{c} + \frac{(4L_1+1)R\varepsilon}{c}\right)$; $\varepsilon$ = desired accuracy; $d$ = dimension; $\Delta$ = Noise Level (see Section 5); SEOD = summand with exponential objective decrease.

| Algorithm | Number of Iterations #N | Batch Size #B | Maximum Noise Level #$\Delta$ | SEOD? | Reference |
|---|---|---|---|---|---|
| ClipSGD | $\mathcal{O}\left(\frac{L_0 R^2}{\varepsilon} + \frac{\sigma^2 R^2}{\varepsilon^2} + L_1^2 R^2\right)$ | ✗ | ✗ | ✗ | Gaash et al. (2025) |
| | $\mathcal{O}\left(\frac{R}{\eta c}\log\frac{1}{\varepsilon} + \frac{R^2}{\eta\varepsilon}\right)$ | $\mathcal{O}\left(\frac{\sigma^2 \mathcal{R}}{\varepsilon}\right)$ | ✗ | ✓ | Theorem 3.1 (**Ours**) |
| NSGD | $\mathcal{O}\left(\left(L_1 R + \frac{L_0 R^2}{\varepsilon}\right)\log\frac{1}{\varepsilon}\right)$ | $\mathcal{O}\left(\max\left\{\frac{\sigma^2(4L_1+1)R^3}{\varepsilon^2}, \frac{\sigma^2 L_0 R^3}{\varepsilon^3}\right\}\right)$ | ✗ | ✓ | Theorem 4.1 (**Ours**) |
| ZO-ClipSGD | $\mathcal{O}\left(\frac{R}{\eta c}\log\frac{1}{\varepsilon} + \frac{R^2}{\eta\varepsilon}\right)$ | $\mathcal{O}\left(\frac{dMR\tilde{\sigma}^2}{\varepsilon c^2}\right)$ | $\mathcal{O}\left(\frac{\varepsilon}{\sqrt{d}R(L_0+L_1 M)}\min\left\{\tilde{\sigma}, \frac{\varepsilon}{\sqrt{d}R}\right\}\right)$ | ✓ | Theorem 5.2 (**Ours**) |
| ZO-NSGD | $\mathcal{O}\left(\left(L_1 R + \frac{L_0 R^2}{\varepsilon}\right)\log\frac{1}{\varepsilon}\right)$ | $\mathcal{O}\left(\frac{dMR^3\tilde{\sigma}^2}{\varepsilon^3}\right)$ | $\mathcal{O}\left(\frac{\varepsilon^{3/2}}{\sqrt{d}R^{3/2}(L_0+L_1 M)}\min\left\{\tilde{\sigma}, \frac{\varepsilon^{3/2}}{\sqrt{d}R^{3/2}}\right\}\right)$ | ✓ | Theorem 5.4 (**Ours**) |

to denote Euclidean ball ($l_2$-ball) and $S^d(r) := \left\{x \in \mathbb{R}^d : \|x\| = r\right\}$ to denote Euclidean sphere. We denote $M < \infty$ as $M := \max_{k \in [0, N-1]}\left\|\nabla f(x^k)\right\|$. For simplicity, we denote $f^* := f(x^*)$ and $R := \max_{k \in [0, N-1]}\left\|x^k - x^*\right\|$. We use $\tilde{O}(\cdot)$ to hide the logarithmic coefficients.

**Assumptions on objective function.** Throughout this paper, we refer to the standard $L$-smoothness assumption, which is widely used in the literature (e.g. Polyak, 1987) and has the following form:

**Assumption 1.1** ($L$-smoothness). Function $f$ is $L$-smooth if for any $x, y \in \mathbb{R}^d$ is satisfied:

$$\|\nabla f(y) - \nabla f(x)\| \leq L\|y - x\|.$$

Despite the widespread use of Assumption 1.1, our work focuses on the more general smoothness assumption, which has recently attracted increased interest. In particular, in (Zhang et al., 2019) it was shown that norm of Hesse matrix correlates with norm of gradient function when training neural networks, and in (Lobanov et al., 2024b) it was shown that using generalized smoothness it is possible to significantly improve the convergence of algorithms. $(L_0, L_1)$-smoothness (Zhang et al., 2019; 2020a) has been proposed as a natural relaxation of standard smoothness assumption.

**Assumption 1.2** ($(L_0, L_1)$-smoothness). A function $f : \mathbb{R}^d \to \mathbb{R}$ is $(L_0, L_1)$-smooth if the following inequality is satisfied for any $x, y \in \mathbb{R}^d$ with $\|y - x\| \leq \frac{1}{L_1}$:

$$\|\nabla f(y) - \nabla f(x)\| \leq (L_0 + L_1\|\nabla f(x)\|)\|y - x\|.$$

Assumption 1.2 in the case $L_1 = 0$ covers the standard Assumption 1.1. Moreover, $(L_0, L_1)$-smoothness is strictly more general than $L$-smoothness, see the examples in (Zhang et al., 2019; Chen et al., 2023; Koloskova et al., 2023; Gorbunov et al., 2024).

**Remark 1.3** (Clarification regarding $L_0 = 0$). *In this paper we often emphasize the case $L_0 = 0$ in Assumption 1.2. It is worth noting that the class of functions that do not reach their infimum $x^*$ (converge to an asymptote) satisfies this case. Explicit examples of functions with $L_0 = 0$ are the exponent of the inner product and the logistic function (see (Gorbunov et al., 2024) for details).*

**Assumptions on gradient oracle.** In our analysis, we consider cases with both unbiased and biased gradient oracle equation 2. Therefore, we assume that the bias and variance of gradient oracle equation 2 are bounded:

**Assumption 1.4** (Bounded bias). There exists constant $\zeta \geq 0$ s.t. the bias is bounded if $\forall x \in \mathbb{R}^d$:

$$\|\mathbf{b}(x)\| \leq \zeta.$$

**Assumption 1.5** (Bounded variance). Exists constant $\sigma^2 \geq 0$ s.t. the variance is bounded if $\forall x \in \mathbb{R}^d$:

$$\mathbb{E}\left[\|\mathbf{g}(x, \xi) - \mathbb{E}\left[\mathbf{g}(x, \xi)\right]\|^2\right] \leq \sigma^2.$$

Assumption 1.4 is organic (see, e.g. Lobanov et al., 2024a), and the case $\zeta = 0$ corresponds to the unbiased gradient oracle equation 2. Assumption 1.5 is often used by the community (e.g. Juditsky & Nemirovski, 2010; Lan, 2012), and is sometimes called heavy-tailed noise (Gorbunov et al., 2020).

### 1.3 PAPER ORGANIZATION

Next, our paper has the following structure. In Section 2, we discuss related work. In Section 3, We start to present the main results of our work, in particular, we provide the first strategy for obtaining a summand with exponential objective decrease in the convergence estimate. In Section 4, we analyze

NSGD, showing in which regime exponential objective decrease can be observed in terms of iteration complexity. We provide the first analysis of zero-order algorithms under $(L_0, L_1)$-smoothness in Section 5. In Section 6, we discuss the results obtained. While, in Section 7, we show experimentally about the possibility of the zero-order algorithm outperforming the first-order algorithm. Finally, Section 8 concludes our paper. All missing proofs are provided in the supplementary materials.

## 2 RELATED WORKS

In this section, we will discuss the most related works.

**Algorithms under $(L_0, L_1)$-smoothness.** Generalized smoothness was first introduced in (Zhang et al., 2019), which analyzed ClipSGD in the non-convex setting. A number of works (Wang et al., 2023; Li et al., 2023b; Faw et al., 2023; Li et al., 2023a; Hong & Lin, 2024; Xie et al., 2024; Wang et al., 2024) followed that also focused on the non-convex setup, including ClipSGD (Zhang et al., 2020b;a; Koloskova et al., 2023), NSGD (Zhao et al., 2021; Hübler et al., 2024b). After that, there was interest in research on algorithms in the convex deterministic setting: Clipped Gradient Descent (Koloskova et al., 2023), Normalized Gradient Descent (Vankov et al., 2024), Gradient Descent with Polyak step size $\eta_k = \frac{f(x^k) - f^*}{\|\nabla f(x^k)\|^2}$ (Takezawa et al., 2024), and $\eta_k = \frac{1}{L_0 + L_1 \|\nabla f(x^k)\|}$ (Gorbunov et al., 2024; Vankov et al., 2024). Moreover, in (Lobanov et al., 2024b), it was theoretically shown that it is possible to significantly improve the convergence of algorithms in the (strongly) convex setting by achieving linear convergence rate. However, much less attention has been paid to the stochastic convex setting. Perhaps the only results are (Gorbunov et al., 2024; Gaash et al., 2025), which considers SGD and ClipSGD achieving only a sublinear convergence rate. Moreover, in the case $L_0 = 0$ in Assumption 1.2, the algorithms from (Gorbunov et al., 2024; Gaash et al., 2025) cannot converge to the desired accuracy. *In our work, we focus on the stochastic convex setup, showing that existing convergence results (in particular, iteration complexity) can be significantly improved.*

**Zero-order algorithms.** The work of (Gasnikov et al., 2022) showed that to achieve optimal estimates of iteration $N$ and oracle $T$ complexity in zero-order algorithms, one could base it on a first-order algorithm using a gradient approximation as the biased gradient oracle equation 2, which uses only information about the objective function $f$. Using this technique a number of works have achieved the best convergence results in various settings including distributed optmization (Akhavan et al., 2021), federated optimization (Patel et al., 2022), overparameterization (Lobanov & Gasnikov, 2023), Polyak-Lojasiewicz condition (Gasnikov et al., 2024), etc. However, all these works assumed standard smoothness (Assumption 1.1) and achieved only sublinear convergence rates. *In our work, we present convergence results for zero-order algorithms under $(L_0, L_1)$-smoothness.*

## 3 CLIPPED STOCHASTIC GRADIENT DESCENT

In this section we begin to present the main results of our work. In particular, we analyze the convergence of SGD variants under convexity and $(L_0, L_1)$-smoothness with arbitrary constant step size. We assume that the gradient oracle equation 2 is unbiased $\zeta = 0$, i.e., Assumption 1.5 takes:

$$\mathbb{E}\left[\|\nabla f(x, \xi) - \nabla f(x)\|^2\right] \leq \sigma^2.$$

As a first strategy to obtain the summands with exponential objective decrease, we consider the clipping technique. Applying this technique we produce the ClipSGD, which has the following form:

---

**Algorithm 1** Clipped Stochastic Gradient Descent Method (ClipSGD)

---

**Input:** initial point $x^0 \in \mathbb{R}^d$, iterations $N$, batch size $B$, step size $\eta_k > 0$ and clipping radius $c > 0$
**for** $k = 0$ **to** $N - 1$ **do**
    1. Draw fresh i.i.d. samples $\xi_1^k, ..., \xi_B^k$
    2. $\nabla f(x^k, \boldsymbol{\xi}^k) = \frac{1}{B} \sum_{i=1}^B \nabla f(x^k, \xi_i^k)$
    3. $\text{clip}_c(\nabla f(x^k, \boldsymbol{\xi}^k)) = \min\left\{1, \frac{c}{\|\nabla f(x^k, \boldsymbol{\xi}^k)\|}\right\} \nabla f(x^k, \boldsymbol{\xi}^k)$
    4. $x^{k+1} \leftarrow x^k - \eta_k \cdot \text{clip}_c(\nabla f(x^k, \boldsymbol{\xi}^k))$
**end for**
**Return:** $x^N$

---

Algorithm 1 uses the clipped stochastic gradient $\text{clip}_c(\nabla f(x, \boldsymbol{\xi}))$, which normalizes the gradient only if $\|\nabla f(x, \boldsymbol{\xi})\| > c$. Next theorem provides the convergence result for ClipSGD.

**Theorem 3.1.** *Let function $f$ satisfy Assumption 1.2 and unbiased gradient oracle satisfy Assumption 1.5, then Algorithm 1 with constant step size $\eta_k = \eta \leq [4(L_0 + L_1 c)]^{-1}$, $B \geq \sigma^2/\varepsilon \left[\eta + L_0 R/c^2 + R/c + (4L_1 + 1)R\varepsilon/c\right]$ and clipping radius $c$ provides error:*

$$\mathbb{E}\left[f(x^N)\right] - f^* \lesssim \left(1 - \frac{\eta c}{R}\right)^N (f(x^0) - f^*) + \frac{R^2}{\eta N} + \frac{\sigma^2}{B}\left(\eta + \frac{L_0 R}{c^2} + \frac{R}{c}\right).$$

It should be noted that the results of Theorem 3.1 are given with a choice of step size independent of the gradient at the current point. This choice of step allows us to separate the constants $L_0$ and $L_1$ in the final estimates. The summand $\frac{R^2}{\eta N}$ is a typical ClipSGD characterizing the sublinear rate (see e.g. (Gorbunov et al., 2020)). However, it is worth noting that by substituting $\eta = (4[L_0 + L_1 c])^{-1}$, then the summand with $L_1$: $\frac{L_1 c R^2}{N}$ already improves existing results both assuming standard smoothness (Gorbunov et al., 2020) and generalized smoothness (Gaash et al., 2025). The first summand $\left(1 - \frac{\eta c}{R}\right)^N (f(x^0) - f^*)$, which characterizes exponential objective decrease, deserves special attention. *To the best of our knowledge, this is the first result for ClipSGD showing such a summand in a convex setting.* Moreover, by substituting $\eta = (4[L_0 + L_1 c])^{-1}$, it is not hard to see that in the regime $L_0 = 0$ (see Remark 1.3), Algorithm 1 at a batch size $B = \mathcal{O}\left(\frac{\sigma^2}{\varepsilon}\left[\eta + \frac{L_0 R}{c^2} + \frac{R}{c} + \frac{(4L_1+1)R\varepsilon}{c}\right]\right)$ requires only $N = \mathcal{O}\left(L_1 R \log\frac{1}{\varepsilon} + \frac{L_1 c R^2}{\varepsilon}\right)$ iterations. This iteration complexity significantly outperforms standard results in the $L$-smoothness setting (Assumption 1.1), since (Lan, 2012) shows a lower bound consisting only of a sublinear convergence rate. Moreover, comparing to the closest work to the problem setting, then even when $\sigma = 0$ (Gaash et al., 2025) does not guarantee convergence to the desired accuracy, offering an estimate of $N = \mathcal{O}\left(L_1^2 R^2\right)$ that is independent of accuracy.

## 4 NORMALIZED STOCHASTIC GRADIENT DESCENT

In the previous section, we showed that it is possible to obtain in the final convergence estimate a summand with exponential objective decrease. In addition, we highlighted the regime $L_0 = 0$, in which ClipSGD has the following iteration complexity: $N = \mathcal{O}\left(L_1 R \log\frac{1}{\varepsilon} + \frac{L_1 c R^2}{\varepsilon}\right)$. However, with this iteration complexity, it cannot be said that the algorithm can converge with linear rate to the desired accuracy. Such an estimate can characterize that the algorithm converges with linear rate as long as the gradient norm is large $\left\|\nabla f(x^k)\right\| \geq c$, and then slows down to the sublinear rate. However, it is worth noting that the summand responsible for the sublinear rate depends on the clipping radius: $\frac{L_1 c R^2}{\varepsilon}$. That is, if we take $c$ smaller, the ClipSGD will take longer to converge to the linear rate. Thus, noticing that the regime $\left\|\nabla f(x^k)\right\| \geq c$ is a gradient normalization, then considering NSGD (see Algorithm 2), it seems that one can achieve a true linear convergence rate to the desired accuracy.

---

**Algorithm 2** Normalized Stochastic Gradient Descent Method (NSGD)

**Input:** initial point $x^0 \in \mathbb{R}^d$, iterations number $N$, batch size $B$ and step size $\eta_k > 0$
**for** $k = 0$ **to** $N - 1$ **do**
    1. Draw fresh i.i.d. samples $\xi_1^k, ..., \xi_B^k$
    2. $\nabla f(x^k, \boldsymbol{\xi}^k) = \frac{1}{B}\sum_{i=1}^B \nabla f(x^k, \xi_i^k)$
    3. $x^{k+1} \leftarrow x^k - \eta_k \cdot \frac{\nabla f(x^k, \boldsymbol{\xi}^k)}{\|\nabla f(x^k, \boldsymbol{\xi}^k)\|}$
**end for**
**Return:** $x^N$

---

The following theorem provides a convergence result for Algorithm 2.

**Theorem 4.1.** *Let function $f$ satisfy Assumption 1.2 and unbiased gradient oracle equation 2 satisfy Assumption 1.5, then Algorithm 2 with $\lambda \leq \varepsilon/R$, $B \geq \max\left\{\sigma^2(4L_1 + 1)R^3/\varepsilon^2, \sigma^2 L_0 R^3/\varepsilon^3\right\}$ and $\eta_k = \eta \leq \lambda/\left[2(L_0 + L_1\lambda)\right]$ guarantees:*

$$\mathbb{E}\left[f(x^N)\right] - f^* \lesssim \left(1 - \frac{\eta}{R}\right)^N (f(x^0) - f^*) + \frac{\sigma^2 L_0 R}{B\lambda^2} + \lambda R.$$

From Theorem 4.1 we can see that we have indeed got rid of the summand characterizing the sublinear rate from the deterministic part. *Thus, we see that it is normalization that allows us to achieve the summand with exponential objective decrease* $\left(1 - \frac{\eta}{R}\right)^N \left(f(x^0) - f^*\right)$. However, note that by substituting $\eta = \lambda / \left[2(L_0 + L_1\lambda)\right]$, we obtain a summand with $L_0 : \left(1 - \frac{\lambda}{RL_0}\right)^N \left(f(x^0) - f^*\right)$, which is in fact sublinear since it depends on the hyperparameter $\lambda$ (it follows from the third summand that $\lambda \sim \varepsilon / R$), and with $L_1 : \left(1 - \frac{1}{RL_1}\right)^N \left(f(x^0) - f^*\right)$, which is indeed linear since it does not depend on $\lambda$ in any way. That is, Algorithm 2, which uses batch parallelization, requires $N = \mathcal{O}\left(\left(L_1 R + \frac{L_0 R^2}{\varepsilon}\right) \log \frac{1}{\varepsilon}\right)$ iterations. Similar to the reasoning in the previous section, it is worth highlighting the regime $L_0 = 0$ (see Remark 1.3). Then we obtain a very surprising result on iteration complexity, namely, to achieve the desired accuracy NSGD converge with a linear rate of $N = \mathcal{O}\left(L_1 R \log \frac{1}{\varepsilon}\right)$ iterations. *This estimate breaks all existing bounds on first-order algorithms (Lan, 2012), given the specificity of the problem formulation, namely convexity.* However, to achieve this rate over iterations, NSGD requires a batch size $B = \mathcal{O}\left(\frac{\sigma^2 L_0 R^3}{\varepsilon^3}\right)$. We emphasize that the fact that NSGD requires a large batch size is not surprising (see, e.g., (Cutkosky & Mehta, 2020)), in contrast to the true linear convergence rate.

## 5 Zero-Order Algorithms

In this section, we consider another class of algorithms: optimization algorithms that have access only to an objective function value $f$ possibly with some bounded adversarial noise $|\delta(x)| \leq \Delta$:

$$\tilde{f}(x, \xi) = f(x, \xi) + \delta(x). \tag{3}$$

In equation 3, $\Delta$ means the maximum possible allowable noise level at which the desired accuracy can still be achieved. In (Lobanov, 2025), the importance of considering $\Delta$ as a third optimality criterion for zero-order algorithms was shown. In particular, in some applications (Bogolubsky et al., 2016), the larger noise level $\Delta$ is, the cheaper the call to the inexact oracle $\tilde{f}$ in equation 3.

Since this class of algorithms does not have access to the stochastic gradient $\nabla f(x, \xi)$, the gradient oracle equation 2 will be the gradient approximation (Shamir, 2017; Nesterov & Spokoiny, 2017):

$$\mathbf{g}(x, \{e, \xi\}) = \frac{d}{2\gamma} \left(\tilde{f}(x + \gamma e, \xi) - \tilde{f}(x - \gamma e, \xi)\right) e, \tag{4}$$

where $\gamma > 0$ is a smoothing parameter, $e$ is a random vector uniformly distributed in $S^d(1)$.

Due to the fact that the gradient approximation is the biased gradient oracle equation 2, in order to create zero-order algorithms by basing on the results in Sections 3 and 4, it is necessary to first generalize the results of Theorems 3.1, 4.1 (note that in these regimes it is not necessary to know the (stochastic) norm of the gradient with step size) to the case of gradient oracle with bias.

### 5.1 ZO-ClipSGD Method

The first algorithm we consider in this section is ZO-ClipSGD. This algorithm is a modification of ClipSGD (Algorithm 1), which uses instead of the original $\|\nabla f(x, \xi)\|$, the stochastic gradient approximation equation 4, which is the biased gradient oracle equation 2. The ZO-ClipSGD has form:

---

**Algorithm 3** Zero-Order Clipped Stochastic Gradient Descent Method (ZO-ClipSGD)

---

**Input:** initial point $x^0 \in \mathbb{R}^d$, iterations $N$, batch size $B$, step size $\eta_k > 0$ and clipping radius $c > 0$
**for** $k = 0$ **to** $N - 1$ **do**
    1. Draw fresh i.i.d. samples $\xi_1^k, ..., \xi_B^k$ and $e_1^k, ..., e_B^k$
    2. $\mathbf{g}(x^k, \{\mathbf{e}^k, \boldsymbol{\xi}^k\}) = \frac{1}{B} \sum_{i=1}^{B} \mathbf{g}(x^k, \{e_i^k, \xi_i^k\})$ via equation 4
    3. $\text{clip}_c(\mathbf{g}(x^k, \{\mathbf{e}^k, \boldsymbol{\xi}^k\})) = \min\left\{1, \frac{c}{\|\mathbf{g}(x^k, \{\mathbf{e}^k, \boldsymbol{\xi}^k\})\|}\right\} \mathbf{g}(x^k, \{\mathbf{e}^k, \boldsymbol{\xi}^k\})$
    4. $x^{k+1} \leftarrow x^k - \eta_k \cdot \text{clip}_c(\mathbf{g}(x^k, \{\mathbf{e}^k, \boldsymbol{\xi}^k\}))$
**end for**
**Return:** $x^N$

---

Before proceeding to present the convergence results of Algorithm 3, we note that the gradient approximation equation 4 is a biased gradient oracle, so we cannot directly use the results obtained in Theorem 3.1. Thus, in order to obtain estimates for the iteration complexity $N$, oracle complexity $T$ and maximum noise $\Delta$, we first need to generalize the results of Theorem 3.1 to the case with a biased gradient oracle equation 2.

**Lemma 5.1.** *Let function $f$ satisfy Assumption 1.2 and biased gradient oracle equation 2 ($\zeta > 0$) satisfy Assumption 1.5, then Algorithm 1 with step size $\eta_k = \eta \leq [4(L_0 + L_1 c)]^{-1}$ guarantees error:*

$$\mathbb{E}\left[f(x^N)\right] - f^* \lesssim \left(1 - \frac{\eta c}{R}\right)^N (f(x^0) - f^*) + \frac{R^2}{\eta N} + \mathcal{R} \cdot \left(\frac{\sigma^2}{B} + \zeta^2\right) + R\zeta,$$

*where $c$ is arbitrary clipping radius, $\mathcal{R} = \left(\eta + \frac{MR}{c^2} + \frac{R}{c}\right)$.*

Lemma 5.1 shows how the bias accumulates over iterations, thus converging to the error floor. By estimating the second moment and the bias of the gradient approximation equation 4 and substituting them into Lemma 5.1, we find three optimality criteria for ZO-ClipSGD.

---

**Theorem 5.2.** *Let function $f$ satisfy Assumption 1.2, gradient approximation equation 4 satisfy Assumption 1.5, then Algorithm 3 with step size $\eta_k = \eta \leq [4(L_0 + L_1 c)]^{-1}$ converges to desired $\varepsilon$ accuracy after:*

$$N = \mathcal{O}\left(\frac{R}{\eta c} \log \frac{1}{\varepsilon} + \frac{R^2}{\eta \varepsilon}\right); \qquad T = \mathcal{O}\left(\frac{d\tilde{\sigma}^2 M R^2}{\varepsilon c^2 \eta}\left(\frac{1}{c}\log\frac{1}{\varepsilon} + \frac{R}{\varepsilon}\right)\right)$$

*number of iterations and zero-order oracle calls at*

$$\Delta \lesssim \frac{\varepsilon}{\sqrt{d}R(L_0 + L_1 M)} \min\left\{\tilde{\sigma}, \frac{\varepsilon}{\sqrt{d}R}\right\}$$

*maximum noise level, where $c > 0$ is clipping radius, $\mathbb{E}\left[\|\nabla f(x, \xi)\|^2\right] \leq \tilde{\sigma}^2$.*

---

It is not hard to see from Theorem 5.2 that in the generalized smoothness condition, the iteration complexity at ZO-ClipSGD is the same as the first-order method of Algorithm 1. This effect is similar in the standard smoothness condition as well. The oracle complexity is $d$ times worse than its first-order counterpart due to the restriction to the oracle (Algorithm 3 uses only the zero-order oracle equation 3). It is worth noting that the maximum noise level $\Delta$ outperforms (Kornilov et al., 2023), showing that generalized smoothness not only allows us to reach the summands with exponential objective decrease, but also improves the estimate on the maximum noise level (it is $\Delta$ that affects the error floor, in other words, the accuracy of the solution, to control the asymptote).

### 5.2 ZO-NSGD METHOD

Similar to the first-order algorithms, in this subsection we answer the question whether linear convergence can be achieved by the zero-order method in a convex setup. To answer this question, we consider the Zero-Order Normalized Stochastic Gradient Descent Method.

---

**Algorithm 4** Zero-Order Normalized Stochastic Gradient Descent Method (ZO-NSGD)

---

**Input:** initial point $x^0 \in \mathbb{R}^d$, iterations number $N$, batch size $B$, step size $\eta_k > 0$
**for** $k = 0$ to $N - 1$ **do**
    1. Draw fresh i.i.d. samples $\xi_1^k, ..., \xi_B^k$ and $e_1^k, ..., e_B^k$
    2. $\mathbf{g}(x^k, \{\mathbf{e}^k, \boldsymbol{\xi}^k\}) = \frac{1}{B}\sum_{i=1}^B \mathbf{g}(x^k, \{e_i^k, \xi_i^k\})$ via equation 4
    3. $x^{k+1} \leftarrow x^k - \eta_k \cdot \frac{\mathbf{g}(x^k, \{\mathbf{e}^k, \boldsymbol{\xi}^k\})}{\|\mathbf{g}(x^k, \{\mathbf{e}^k, \boldsymbol{\xi}^k\})\|}$
**end for**
**Return:** $x^N$

---

In a similar way as in the previous subsection, we first generalize Theorem 4.1 to the case with a biased gradient oracle to substitute estimates on the bias and the second moment of the gradient approximation to find optimality criteria for the gradient-free algorithm ZO-NSGD.

**Lemma 5.3.** *Let function $f$ satisfy Assumption 1.2 ($(L_0, L_1)$-smoothness) and biased gradient oracle equation 2 ($\zeta > 0$) satisfy Assumption 1.5 (bounded variance), then Algorithm 4 with hyperparameter $\lambda > 0$ and step size $\eta_k = \eta \leq \lambda / \left[2(L_0 + L_1\lambda)\right]$ guarantees:*

$$\mathbb{E}\left[f(x^N)\right] - f^* \lesssim \left(1 - \frac{\eta}{R}\right)^N (f(x^0) - f^*) + \frac{MR}{\lambda^2}\left(\frac{\sigma^2}{B} + \zeta^2\right) + \lambda R.$$

From Lemma 5.3 we can see how the inaccuracy accumulates over the iteration. The summand with $\zeta^2$ is unimprovable for first-order unaccelerated algorithms (see, e.g., (Devolder, 2013; Gasnikov et al., 2024)). Now, having obtained the results for the biased NSGD we can use them to derive convergence results for Algorithm 4.

**Theorem 5.4.** *Let function $f$ satisfy Assumption 1.2 ($(L_0, L_1)$-smoothness) and gradient approximation equation 4 satisfy Assumption 1.5 (bounded variance), then ZO-NSGD with $\eta_k = \eta \leq \lambda / \left[2(L_0 + L_1\lambda)\right]$ converges to desired $\varepsilon$ accuracy $\left(\mathbb{E}\left[f(x^N)\right] - f^* \leq \varepsilon\right)$ after:*

$$N = \mathcal{O}\left(\frac{R}{\eta}\log\frac{1}{\varepsilon}\right); \qquad T = \tilde{\mathcal{O}}\left(\frac{d\tilde{\sigma}^2 MR^4}{\varepsilon^3 \eta}\right)$$

*number of iterations and zero-order oracle calls equation 3 at*

$$\Delta \lesssim \frac{\varepsilon^{3/2}}{\sqrt{d}R^{3/2}(L_0 + L_1 M)}\min\left\{\tilde{\sigma}, \frac{\varepsilon^{3/2}}{\sqrt{d}R^{3/2}}\right\}$$

*maximum noise level, where $\lambda > 0$ is hyperparameter, $\mathbb{E}\left[\|\nabla f(x, \xi)\|^2\right] \leq \tilde{\sigma}^2$.*

It is not hard to see that given a restricted oracle equation 3, Theorem 5.4 shows that ZO-NSGD requires $N = \mathcal{O}\left(\frac{R}{\eta}\log\frac{1}{\varepsilon}\right)$ iterations to achieve the desired accuracy, which corresponds to a linear rate. However, it is worth noting that, as in Theorem 4.1, if we take the maximum step size $\eta = \lambda / \left[2(L_0 + L_1\lambda)\right]$, then the summand with $L_0$ still shows sublinear convergence $\tilde{\mathcal{O}}\left(\frac{L_0 R^2}{\varepsilon}\right)$, despite the presence of the summand with $L_1$. But in the $L_0 = 0$ regime, *we can say unambiguously that the zero-order algorithm ZO-NSGD can converge with a linear rate to the desired accuracy in the convex setup if we take the batch size $B = \mathcal{O}\left(\frac{d\tilde{\sigma}^2 MR^3}{\varepsilon^3}\right)$. Theorem 5.4 shows that the power of generalized smoothness (together with Batch parallelization) extends to zero-order algorithms.* Comparing the result of ZO-NSGD with Theorem 5.2, we can see that for a significant improvement in iteration complexity $N$ we "pay" for a deterioration in both oracle complexity $T$ and maximum noise level $\Delta$, which seems quite natural. See the proof of Lemma 5.3 and Theorem 5.4 in Appendix E.

## 6 DISCUSSION AND FUTURE WORKS

In Sections 3-5, we gave two strategies for obtaining summands with exponential objective decrease in convergence estimates of algorithms for the convex setting: clipping (see Section 3) and normalization (see Section 4) techniques. Although these summands are quite unexpected for the convex setup and improve the estimates on the iteration complexity, we cannot claim linear convergence in general, since the convergence is dominated by the summands characterizing the sublinear rate. However, as we noted in Theorems 4.1 and 5.4, in the regime $L_0 = 0$, the NSGD and ZO-NSGD methods break all existing bounds on iteration complexity, demonstrating that it is possible to converge with linear rate to the desired accuracy with the condition of using Batch paralllization. This result is pleasantly surprising and opens up a number of directions for future research.

In this paper we have focused on iteration complexity, so we see a careful analysis of the optimality criterion in the aggregate as future work. In particular, it seems interesting to show that $M$ is indeed bounded, e.g., using the technique from (Li et al., 2023a) and evaluating with respect to the smoothness constants $L_0$ and $L_1$. The existence of the regime $L_0 = 0$ which allows one to achieve a linear convergence rate in the convex setting prompts the following question: can the iteration complexity be improved by assuming, for example, strong convexity, the PL condition, etc.? It is also interesting to see if similar effects are found in accelerated, adaptive algorithms, variational inequalities, distributed learning, nonsmooth (or increased smoothness) problems, overparameterization, online optimization.

## 7 NUMERICAL EXPERIMENTS

In this section, we numerically analyze the algorithms presented in this paper and show that linear convergence in stochastic convex optimization is possible. For this illustration, we have chosen a problem that is of particular interest in the machine learning community: the logistic regression problem on w1a dataset (Platt, 1998). We consider the following convex problem statement equation 1:

$$\min_{x \in \mathbb{R}^d} f(x) = \frac{1}{V} \sum_{i=1}^{V} f_i(x), \qquad f_i(x) = \log\left(1 + \exp(-y_i \cdot (Ax)_i)\right),$$

where $f_i(x)$ is the loss on the $i$-th data point, $A \in \mathbb{R}^{V \times d}$ is an instances matrix, $y \in \{-1, 1\}^V$ is a label vector and $x \in \mathbb{R}^d$ is a vector of weights. It is easy to show, that logistic regression function is $L$-smooth (see Assumption 1.1) with $L = \frac{1}{4V}\lambda_{\max}(A^T A)$, where $\lambda_{\max}(A^T A)$ denotes the largest eigenvalue of the matrix $A^T A$. Moreover, such a problem statement is a special case of equation 1 with $\xi$ being a random variable with the uniform distribution on $\{1, ..., V\}$.

In all tests we used the following parameters: $V = 2477$ - number of data, $d = 300$ - problem dimension, $B = 10$ - batch size, $\gamma = 10^{-5}$ - smoothing parameter, $\Delta = 10^{-9}$ - noise level. Figure 2 demonstrates the convergence dynamics of all the algorithms considered in this paper. In particular, NSGD (see red line) *demonstrates that indeed the first-order algorithm can converge with linear rate to the desired accuracy in a convex setup!* ClipSGD (see green line), which used a step size $\eta = \frac{1}{c \cdot \|A\|_1}$, where $c = 10^{-1}$ is the clipping radius, shows two modes of

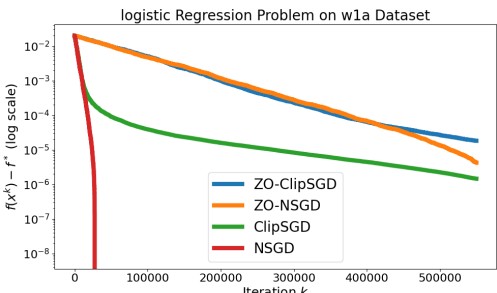

Figure 2: Comparison of the considered algorithms.

convergence: as long as $\|\nabla f(x^k)\| \geq c$ the algorithm converges with a linear rate matching the NSGD (which used a step size $\eta = \frac{1}{\|A\|_1}$), as soon as $\|\nabla f(x^k)\| < c$ the algorithm slows down to the sublinear rate. The dynamics are similar for zero-order algorithms: ZO-NSGD (see orange line), ZO-ClipSGD (see blue line). Expectedly, these algorithms converge slower on the first iterations than their first-order counterparts due to restricted access to oracle equation 3. However, it is worth noting that *ZO-NSGD also exhibits linear convergence, potentially outperforming the first-order ClipSGD algorithm after* 55000 *iterations (taking into account the convergence dynamics of the algorithms).*

## 8 CONCLUSION

In this paper, we considered a stochastic convex optimization problem under the generalized smoothness condition of the objective function. We are the first who have provided strategies to achieve summands with exponential objective decrease, thereby improving the iteration complexity (see Sections 3 and 4). In Section 5, we showed that this effect of generalized smoothness extends to zero-order algorithms as well. Moreover, we highlight the regime $L_0 = 0$, under which we theoretically guarantee linear convergence for the NSGD and ZO-NSGD algorithms (subject to the use of batch paralllization). This is the first result demonstrating linear convergence in such a problem setting, thus opening up a number of future works (see Section 6). Finally, in Section 7 we showed using numerical experiments that linear convergence in such a problem formulation is also possible in practice, and moreover we demonstrated the possibility of the zero-order (ZO-NSGD) algorithm outperforming the first-order (ClipSGD) algorithm (see Figure 2).

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
