# APPENDIX
# Generalized Smoothness in Stochastic Convex Optimization: First- and Zero-Order Methods

## A  AUXILIARY RESULTS

In this section we provide auxiliary materials that are used in the proof of Theorems.

### A.1  BASIC INEQUALITIES AND ASSUMPTIONS

**Basic inequalities.**  For all $a, b \in \mathbb{R}^d$ $(d \geq 1)$ the following equality holds:

$$2 \langle a, b \rangle - \|b\|^2 = \|a\|^2 - \|a - b\|^2, \tag{5}$$

$$\langle a, b \rangle \leq \|a\| \cdot \|b\|. \tag{6}$$

**Squared norm of the sum**  For all $a_1, ..., a_n \in \mathbb{R}^d$

$$\|a_1 + ... + a_n\|^2 \leq n\|a_1\|^2 + ... + n\|a_n\|^2. \tag{7}$$

**Generalized-Lipschitz-smoothness.**  Throughout this paper, we assume that the $(L_0, L_1)$-smoothness condition (Assumption 1.2) is satisfied. This inequality can be represented in the equivalent form for any $x, y \in \mathbb{R}^d$:

$$f(y) - f(x) \leq \langle \nabla f(x), y - x \rangle + \frac{L_0 + L_1 \|\nabla f(x)\|}{2} \|y - x\|^2, \tag{8}$$

where $L_0, L_1 \geq 0$ for any $x \in \mathbb{R}^d$ and $\|y - x\| \leq \frac{1}{L_1}$.

**Variance decomposition.**  If $\xi$ is random vector in $\mathbb{R}^d$ with bounded second moment, then

$$\mathbb{E}\left[\|\xi + a\|^2\right] = \mathbb{E}\left[\|\xi - \mathbb{E}[\xi]\|^2\right] + \mathbb{E}\left[\|\mathbb{E}[\xi] - a\|^2\right], \tag{9}$$

for any deterministic vector $a \in \mathbb{R}^d$.

### A.2  AUXILIARY LEMMA ABOUT GENERALIZED SMOOTHNESS

If Assumption 1.2 holds, then it also holds that $\forall x \in \mathbb{R}^d$:

$$\|\nabla f(x)\|^2 \leq 2(L_0 + L_1 \|\nabla f(x)\|)(f(x) - f^*), \tag{10}$$

where $f^* = \inf_x f(x)$.

*Proof.* We start the proof by applying equation 8 for $y = x - \frac{1}{L_0 + L_1 \|\nabla f(x)\|} \nabla f(x)$, where $\|y - x\| = \frac{\|\nabla f(x)\|}{L_0 + L_1 \|\nabla f(x)\|} \leq \frac{1}{L_1}$. Then we can obtain:

$$f^* \leq f\left(x - \frac{1}{L_0 + L_1 \|\nabla f(x)\|} \nabla f(x)\right) \overset{equation\ 8}{\leq} f(x) - \frac{1}{2(L_0 + L_1 \|\nabla f(x)\|)} \|\nabla f(x)\|^2.$$

$\square$

## A.3 AN UPPER BOUND ON THE GRADIENT NORM

If Assumption 1.2 holds, then it also holds that $\forall x \in \mathbb{R}^d$:

$$\|\nabla f(x)\| \leq L_0 + \frac{4L_1 + 1}{2}(f(x) - f^*). \tag{11}$$

*Proof.* We start with equation 10:

$$\|\nabla f(x)\|^2 \leq 2(L_0 + L_1 \|\nabla f(x)\|)(f(x) - f^*)$$
$$\iff \|\nabla f(x)\|^2 - 2L_1 \|\nabla f(x)\| (f(x) - f^*) - 2L_0 \|\nabla f(x)\| \leq 0.$$

We need to solve this quadratic inequality w.r.t. $\|\nabla f(x)\|$. The discriminant is

$$4L_1^2(f(x) - f^*) + 8L_0(f(x) - f^*) > 0,$$

i.e., it is positive. Since $\|\nabla f(x)\| \geq 0$, we should also satisfy

$$\|\nabla f(x)\| \leq \frac{2L_1(f(x) - f^*) + \sqrt{4L_1^2(f(x) - f^*) + 8L_0(f(x) - f^*)}}{2}$$

$$\leq L_1(f(x) - f^*) + \sqrt{L_1^2(f(x) - f^*)} + \sqrt{2L_0(f(x) - f^*)}$$

$$\leq 2L_1(f(x) - f^*) + L_0 + \frac{1}{2}(f(x) - f^*)$$

$$= L_0 + \frac{4L_1 + 1}{2}(f(x) - f^*)$$

$\square$

## A.4 WIRTINGER-POINCARE INEQUALITY

Let $f$ is differentiable, then for all $x \in \mathbb{R}^d, \gamma e \in S^d(\gamma)$:

$$\mathbb{E}\left[f(x + \gamma e)^2\right] \leq \frac{\gamma^2}{d}\mathbb{E}\left[\|\nabla f(x + \gamma e)\|^2\right]. \tag{12}$$

# B CLIPPED STOCHASTIC GRADIENT DESCENT (PROOF OF THE THEOREM 3.1)

We start by using $(L_0, L_1)$-smoothness (see Assumption 1.2):

$$f(x^{k+1}) - f(x^k) \overset{equation\ 8}{\leq} \langle \nabla f(x^k), x^{k+1} - x^k \rangle + \frac{L_0 + L_1 \|\nabla f(x^k)\|}{2} \|x^{k+1} - x^k\|^2$$

$$= -\eta \langle \nabla f(x^k), \text{clip}_c \left(\nabla f(x^k, \boldsymbol{\xi}^k)\right) \rangle$$

$$+ \frac{\eta^2 (L_0 + L_1 \|\nabla f(x^k)\|)}{2} \|\text{clip}_c \left(\nabla f(x^k, \boldsymbol{\xi}^k)\right)\|^2. \tag{13}$$

Next, we consider three cases depending on the gradient norm: $\|\nabla f(x^k)\| \geq c$ – the full gradient is clipped and $\frac{c}{2} \leq \|\nabla f(x^k)\| \leq c$ and $\|\nabla f(x^k)\| \leq \frac{c}{2}$ – the full gradient is not clipped.

## B.1 FIRST CASE: $\|\nabla f(x^k)\| \geq c$

In this case $\alpha \nabla f(x^k) = \text{clip}_c \left(\nabla f(x^k)\right)$ with $\alpha = \min\left\{1, \frac{c}{\|\nabla f(x^k)\|}\right\} = \frac{c}{\|\nabla f(x^k)\|}$, therefore we have the following

$$-\eta \langle \nabla f(x^k), \text{clip}_c \left(f(x^k, \boldsymbol{\xi}^k)\right) \rangle \overset{equation\ 5}{=} -\frac{\alpha \eta}{2} \|\nabla f(x^k)\|^2 - \frac{\eta}{2\alpha} \|\text{clip}_c \left(\nabla f(x^k, \boldsymbol{\xi}^k)\right)\|^2$$

$$+ \frac{\eta}{2\alpha} \|\text{clip}_c \left(\nabla f(x^k, \boldsymbol{\xi}^k)\right) - \alpha \nabla f(x^k)\|^2$$

$$= -\frac{\alpha\eta}{2}\left\|\nabla f(x^k)\right\|^2 - \frac{\eta}{2\alpha}\left\|\text{clip}_c\left(\nabla f(x^k,\boldsymbol{\xi}^k)\right)\right\|^2$$

$$+ \frac{\eta}{2\alpha}\left\|\text{clip}_c\left(\nabla f(x^k,\boldsymbol{\xi}^k)\right) - \text{clip}_c\left(\nabla f(x^k)\right)\right\|^2$$

$$= -\frac{c\eta}{2}\left\|\nabla f(x^k)\right\| - \frac{\eta}{2\alpha}\left\|\text{clip}_c\left(\nabla f(x^k,\boldsymbol{\xi}^k)\right)\right\|^2$$

$$+ \frac{\eta}{2\alpha}\left\|\text{clip}_c\left(\nabla f(x^k,\boldsymbol{\xi}^k)\right) - \text{clip}_c\left(\nabla f(x^k)\right)\right\|^2.$$

Using that clipping is a projection on onto a convex set, namely ball with radius $c$, and thus is Lipshitz operator with Lipshitz constant 1, we can obtain:

$$-\eta\left\langle\nabla f(x^k), \mathbb{E}\left[\text{clip}_c\left(f(x^k,\boldsymbol{\xi}^k)\right)\right]\right\rangle \leq -\frac{c\eta}{2}\left\|\nabla f(x^k)\right\| - \frac{\eta}{2\alpha}\mathbb{E}\left[\left\|\text{clip}_c\left(\nabla f(x^k,\boldsymbol{\xi}^k)\right)\right\|^2\right]$$

$$+ \frac{\eta}{2\alpha}\mathbb{E}\left[\left\|\nabla f(x^k,\boldsymbol{\xi}^k) - \nabla f(x^k)\right\|^2\right]$$

$$\leq -\frac{c\eta}{2}\left\|\nabla f(x^k)\right\| - \frac{\eta}{2\alpha}\mathbb{E}\left[\left\|\text{clip}_c\left(\nabla f(x^k,\boldsymbol{\xi}^k)\right)\right\|^2\right]$$

$$+ \frac{\eta\sigma^2}{2\alpha B}$$

$$= -\frac{c\eta}{2}\left\|\nabla f(x^k)\right\| - \frac{\eta}{2\alpha}\mathbb{E}\left[\left\|\text{clip}_c\left(\nabla f(x^k,\boldsymbol{\xi}^k)\right)\right\|^2\right]$$

$$+ \frac{\eta\left\|\nabla f(x^k)\right\|\sigma^2}{2cB}. \tag{14}$$

We now consider the cases depending on the relation between $c$ and $\sigma$:

**In the case $c \geq \sqrt{2}\sigma$**    We have in equation 14:

$$-\eta\left\langle\nabla f(x^k), \mathbb{E}\left[\text{clip}_c\left(\nabla f(x^k,\boldsymbol{\xi}^k)\right)\right]\right\rangle \overset{equation\ 14}{\leq} -\frac{c\eta}{2}\left\|\nabla f(x^k)\right\| - \frac{\eta}{2\alpha}\mathbb{E}\left[\left\|\text{clip}_c\left(\nabla f(x^k,\boldsymbol{\xi}^k)\right)\right\|^2\right]$$

$$+ \frac{\eta\left\|\nabla f(x^k)\right\|\sigma^2}{2cB}$$

$$= -\frac{\eta}{2\alpha}\mathbb{E}\left[\left\|\text{clip}_c\left(\nabla f(x^k,\boldsymbol{\xi}^k)\right)\right\|^2\right]$$

$$- \frac{c\eta}{2}\left\|\nabla f(x^k)\right\|\left(1 - \frac{\sigma^2}{c^2 B}\right)$$

$$\leq -\frac{\eta}{2\alpha}\mathbb{E}\left[\left\|\text{clip}_c\left(\nabla f(x^k,\boldsymbol{\xi}^k)\right)\right\|^2\right]$$

$$- \frac{c\eta}{4}\left\|\nabla f(x^k)\right\|$$

$$= -\frac{\eta\left\|\nabla f(x^k)\right\|}{2c}\mathbb{E}\left[\left\|\text{clip}_c\left(\nabla f(x^k,\boldsymbol{\xi}^k)\right)\right\|^2\right]$$

$$- \frac{c\eta}{4}\left\|\nabla f(x^k)\right\|.$$

Plugging this into equation 13 and choosing $\eta \leq \frac{1}{4(L_0 + L_1 c)}$ we have:

$$\mathbb{E}\left[f(x^{k+1})\right] - f(x^k) \overset{equation\ 13}{\leq} -\frac{\eta\left\|\nabla f(x^k)\right\|}{2c}\mathbb{E}\left[\left\|\text{clip}_c\left(\nabla f(x^k,\boldsymbol{\xi}^k)\right)\right\|^2\right] - \frac{c\eta}{4}\left\|\nabla f(x^k)\right\|$$

$$+ \frac{\eta^2(L_0 + L_1\left\|\nabla f(x^k)\right\|)}{2}\mathbb{E}\left[\left\|\text{clip}_c\left(\nabla f(x^k,\boldsymbol{\xi}^k)\right)\right\|^2\right]$$

$$= -\frac{\eta\left\|\nabla f(x^k)\right\|}{2c}\mathbb{E}\left[\left\|\text{clip}_c\left(\nabla f(x^k,\boldsymbol{\xi}^k)\right)\right\|^2\right](1 - \eta L_1 c)$$

$$- \frac{c\eta}{4}\left\|\nabla f(x^k)\right\| + \frac{\eta^2 L_0}{2}\mathbb{E}\left[\left\|\text{clip}_c\left(\nabla f(x^k,\boldsymbol{\xi}^k)\right)\right\|^2\right]$$

$$\leq -\frac{c\eta}{4}\left\|\nabla f(x^k)\right\| - \frac{\eta}{2}\mathbb{E}\left[\left\|\text{clip}_c\left(\nabla f(x^k,\boldsymbol{\xi}^k)\right)\right\|^2\right](1 - \eta(L_0 + L_1 c))$$

$$\leq -\frac{c\eta}{4}\left\|\nabla f(x^k)\right\|. \tag{15}$$

Using the convexity assumption of the function, we have the following:

$$f(x^k) - f^* \leq \left\langle \nabla f(x^k), x^k - x^* \right\rangle$$
$$\overset{equation\ 6}{\leq} \left\|\nabla f(x^k)\right\|\left\|x^k - x^*\right\|$$
$$\leq \left\|\nabla f(x^k)\right\|\underbrace{\max_{k\in[0,N-1]}\left\|x^k - x^*\right\|}_{R}.$$

Hence we have:

$$\left\|\nabla f(x^k)\right\| \geq \frac{f(x^k) - f^*}{R}. \tag{16}$$

Then substituting equation 16 into equation 15 we obtain:

$$\mathbb{E}\left[f(x^{k+1})\right] - f(x^k) \leq -\frac{\eta c}{4}\left\|\nabla f(x^k)\right\| \leq -\frac{\eta c}{4R}(f(x^k) - f^*).$$

This inequality is equivalent to the trailing inequality:

$$\mathbb{E}\left[f(x^{k+1})\right] - f^* \leq \left(1 - \frac{\eta c}{4R}\right)(f(x^k) - f^*).$$

Then for $k = 0, 1, 2, ..., N-1$ iterations that satisfy the conditions $\left\|\nabla f(x^k)\right\| \geq c \geq \sqrt{2}\sigma$, then ClipSGD has

$$\mathbb{E}\left[f(x^N)\right] - f^* \leq \left(1 - \frac{\eta}{2R}\right)^N(f(x^0) - f^*).$$

$\boxed{\textbf{In the case } c \leq \sqrt{2}\sigma}$  We have in equation 14:

$$-\eta\left\langle \nabla f(x^k), \mathbb{E}\left[\text{clip}_c\left(f(x^k, \boldsymbol{\xi}^k)\right)\right]\right\rangle \overset{equation\ 14}{\leq} -\frac{c\eta}{2}\left\|\nabla f(x^k)\right\| - \frac{\eta}{2\alpha}\mathbb{E}\left[\left\|\text{clip}_c\left(\nabla f(x^k, \boldsymbol{\xi}^k)\right)\right\|^2\right]$$
$$+ \frac{\eta\left\|\nabla f(x^k)\right\|\sigma^2}{2cB}$$
$$= -\frac{c\eta}{2}\left\|\nabla f(x^k)\right\| - \frac{\eta}{2\alpha}\mathbb{E}\left[\left\|\text{clip}_c\left(\nabla f(x^k, \boldsymbol{\xi}^k)\right)\right\|^2\right]$$
$$+ \frac{\eta M\sigma^2}{2cB}.$$

Plugging this into equation 13 and choosing $\eta \leq \frac{1}{4(L_0 + L_1 c)}$ we have:

$$\mathbb{E}\left[f(x^{k+1})\right] - f(x^k) \overset{equation\ 13}{\leq} -\frac{c\eta}{2}\left\|\nabla f(x^k)\right\| - \frac{\eta\left\|\nabla f(x^k)\right\|}{2c}\mathbb{E}\left[\left\|\text{clip}_c\left(\nabla f(x^k, \boldsymbol{\xi}^k)\right)\right\|^2\right]$$
$$+ \frac{\eta^2(L_0 + L_1\left\|\nabla f(x^k)\right\|)}{2}\mathbb{E}\left[\left\|\text{clip}_c\left(\nabla f(x^k, \boldsymbol{\xi}^k)\right)\right\|^2\right] + \frac{\eta M\sigma^2}{2cB}$$
$$= -\frac{c\eta}{2}\left\|\nabla f(x^k)\right\| - \frac{\eta\left\|\nabla f(x^k)\right\|}{2c}\mathbb{E}\left[\left\|\text{clip}_c\left(\nabla f(x^k, \boldsymbol{\xi}^k)\right)\right\|^2\right](1 - \eta L_1 c)$$
$$+ \frac{\eta^2 L_0}{2}\mathbb{E}\left[\left\|\text{clip}_c\left(\nabla f(x^k, \boldsymbol{\xi}^k)\right)\right\|^2\right] + \frac{\eta M\sigma^2}{2cB}$$
$$\leq -\frac{c\eta}{2}\left\|\nabla f(x^k)\right\| - \frac{\eta}{2}\mathbb{E}\left[\left\|\text{clip}_c\left(\nabla f(x^k, \boldsymbol{\xi}^k)\right)\right\|^2\right](1 - \eta(L_0 + L_1 c))$$
$$+ \frac{\eta M\sigma^2}{2cB}$$
$$\leq -\frac{c\eta}{2}\left\|\nabla f(x^k)\right\| + \frac{\eta M\sigma^2}{2cB}. \tag{17}$$

Using the convexity assumption of the function, we have the following:

$$f(x^k) - f^* \leq \langle \nabla f(x^k), x^k - x^* \rangle$$
$$\overset{equation\ 6}{\leq} \|\nabla f(x^k)\| \|x^k - x^*\|$$
$$\leq \|\nabla f(x^k)\| \underbrace{\max_{k \in [0, N-1]} \|x^k - x^*\|}_{R}.$$

Hence we have:

$$\|\nabla f(x^k)\| \geq \frac{f(x^k) - f^*}{R}. \tag{18}$$

Then substituting equation 18 into equation 17 we obtain:

$$\mathbb{E}\left[f(x^{k+1})\right] - f(x^k) \leq -\frac{\eta c}{2} \|\nabla f(x^k)\| + \frac{\eta M \sigma^2}{2cB} \leq -\frac{\eta c}{2R}(f(x^k) - f^*) + \frac{\eta M \sigma^2}{2cB}.$$

This inequality is equivalent to the trailing inequality:

$$\mathbb{E}\left[f(x^{k+1})\right] - f^* \leq \left(1 - \frac{\eta c}{2R}\right)(f(x^k) - f^*) + \frac{\eta M \sigma^2}{2cB}$$
$$\overset{equation\ 11}{\leq} \left(1 - \frac{\eta c}{2R} + \frac{\eta \sigma^2 (4L_1 + 1)}{4cB}\right)(f(x^k) - f^*) + \frac{\eta \sigma^2 L_0}{2cB}$$
$$\overset{①}{\leq} \left(1 - \frac{\eta}{4R}\right)(f(x^k) - f^*) + \frac{\eta \sigma^2 L_0}{2cB},$$

where in ① we used $B = \mathcal{O}\left(\frac{\sigma^2}{\varepsilon}\left[\eta + \frac{L_0 R}{c^2} + \frac{R}{c} + \frac{(4L_1+1)R\varepsilon}{c}\right]\right).$

Then for $k = 0, 1, 2, ..., N - 1$ iterations that satisfy the conditions $\|\nabla f(x^k)\| \geq c$ and $c \leq \sqrt{2}\sigma$, then ClipSGD has

$$\mathbb{E}\left[f(x^N)\right] - f^* \leq \left(1 - \frac{\eta c}{4R}\right)^N (f(x^0) - f^*) + \frac{L_0 R \sigma^2}{c^2 B}.$$

## B.2   SECOND CASE: $\frac{c}{2} \leq \|\nabla f(x^k)\| \leq c$

In this case $\nabla f(x^k) = \text{clip}_c\left(\nabla f(x^k)\right)$ with $\alpha = \min\left\{1, \frac{c}{\|\nabla f(x^k)\|}\right\} = 1$, therefore we have the following

$$-\eta \langle \nabla f(x^k), \text{clip}_c\left(\nabla f(x^k, \boldsymbol{\xi}^k)\right)\rangle \overset{equation\ 5}{=} -\frac{\alpha \eta}{2} \|\nabla f(x^k)\|^2 - \frac{\eta}{2\alpha} \|\text{clip}_c\left(\nabla f(x^k, \boldsymbol{\xi}^k)\right)\|^2$$
$$+ \frac{\eta}{2\alpha} \|\text{clip}_c\left(\nabla f(x^k, \boldsymbol{\xi}^k)\right) - \alpha \nabla f(x^k)\|^2$$
$$= -\frac{\eta}{2} \|\nabla f(x^k)\|^2 - \frac{\eta}{2} \|\text{clip}_c\left(\nabla f(x^k, \boldsymbol{\xi}^k)\right)\|^2$$
$$+ \frac{\eta}{2} \|\text{clip}_c\left(\nabla f(x^k, \boldsymbol{\xi}^k)\right) - \text{clip}_c\left(\nabla f(x^k)\right)\|^2$$
$$\leq -\frac{c\eta}{4} \|\nabla f(x^k)\| - \frac{\eta}{2} \|\text{clip}_c\left(\nabla f(x^k, \boldsymbol{\xi}^k)\right)\|^2$$
$$+ \frac{\eta}{2} \|\text{clip}_c\left(\nabla f(x^k, \boldsymbol{\xi}^k)\right) - \text{clip}_c\left(\nabla f(x^k)\right)\|^2.$$

Using that clipping is a projection on onto a convex set, namely ball with radius $c$, and thus is Lipshitz operator with Lipshitz constant 1, we can obtain:

$$-\eta \langle \nabla f(x^k), \mathbb{E}\left[\text{clip}_c\left(\nabla f(x^k, \boldsymbol{\xi}^k)\right)\right]\rangle \leq -\frac{c\eta}{4} \|\nabla f(x^k)\| - \frac{\eta}{2}\mathbb{E}\left[\|\text{clip}_c\left(\nabla f(x^k, \boldsymbol{\xi}^k)\right)\|^2\right]$$
$$+ \frac{\eta}{2}\mathbb{E}\left[\|\nabla f(x^k, \boldsymbol{\xi}^k) - \nabla f(x^k)\|^2\right]$$

$$\leq -\frac{c\eta}{4} \left\| \nabla f(x^k) \right\| - \frac{\eta}{2} \mathbb{E} \left[ \left\| \text{clip}_c \left( \nabla f(x^k, \boldsymbol{\xi}^k) \right) \right\|^2 \right] + \frac{\eta \sigma^2}{2B}$$

$$= -\frac{c\eta}{4} \left\| \nabla f(x^k) \right\| - \frac{\eta}{2} \mathbb{E} \left[ \left\| \text{clip}_c \left( \nabla f(x^k, \boldsymbol{\xi}^k) \right) \right\|^2 \right] + \frac{\eta \sigma^2}{2B}.$$

Plugging this into equation 13 and choosing $\eta \leq \frac{1}{4(L_0 + L_1 c)}$ we have:

$$\mathbb{E} \left[ f(x^{k+1}) \right] - f(x^k) \overset{equation\ 13}{\leq} -\frac{c\eta}{4} \left\| \nabla f(x^k) \right\| - \frac{\eta}{2} \mathbb{E} \left[ \left\| \text{clip}_c \left( \nabla f(x^k, \boldsymbol{\xi}^k) \right) \right\|^2 \right]$$

$$+ \frac{\eta^2 (L_0 + L_1 \left\| \nabla f(x^k) \right\|)}{2} \mathbb{E} \left[ \left\| \text{clip}_c \left( \nabla f(x^k, \boldsymbol{\xi}^k) \right) \right\|^2 \right] + \frac{\eta \sigma^2}{2B}$$

$$= -\frac{c\eta}{4} \left\| \nabla f(x^k) \right\| + \frac{\eta \sigma^2}{2B}$$

$$- \frac{\eta}{2} \mathbb{E} \left[ \left\| \text{clip}_c \left( \nabla f(x^k, \boldsymbol{\xi}^k) \right) \right\|^2 \right] \left( 1 - \eta(L_0 + L_1 \left\| \nabla f(x^k) \right\|) \right)$$

$$\leq -\frac{c\eta}{4} \left\| \nabla f(x^k) \right\| + \frac{\eta \sigma^2}{2B}. \tag{19}$$

Using the convexity assumption of the function, we have the following:

$$f(x^k) - f^* \leq \left\langle \nabla f(x^k), x^k - x^* \right\rangle$$

$$\overset{equation\ 6}{\leq} \left\| \nabla f(x^k) \right\| \left\| x^k - x^* \right\|$$

$$\leq \left\| \nabla f(x^k) \right\| \underbrace{\max_{k \in [0, N-1]} \left\| x^k - x^* \right\|}_{R}.$$

Hence we have:

$$\left\| \nabla f(x^k) \right\| \geq \frac{f(x^k) - f^*}{R}. \tag{20}$$

Then substituting equation 20 into equation 19 we obtain:

$$\mathbb{E} \left[ f(x^{k+1}) \right] - f(x^k) \leq -\frac{\eta c}{4} \left\| \nabla f(x^k) \right\| + \frac{\eta \sigma^2}{2B} \leq -\frac{\eta c}{4R} (f(x^k) - f^*) + \frac{\eta \sigma^2}{2B}.$$

This inequality is equivalent to the trailing inequality:

$$\mathbb{E} \left[ f(x^{k+1}) \right] - f^* \leq \left( 1 - \frac{\eta c}{4R} \right) (f(x^k) - f^*) + \frac{\eta \sigma^2}{2B}.$$

Then for $k = 0, 1, 2, ..., N-1$ iterations that satisfy the conditions $\frac{c}{2} \leq \left\| \nabla f(x^k) \right\| \leq c$, then ClipSGD has

$$\mathbb{E} \left[ f(x^N) \right] - f^* \leq \left( 1 - \frac{\eta c}{4R} \right)^N (f(x^0) - f^*) + \frac{2\sigma^2 R}{cB}.$$

## B.3 THIRD CASE: $\left\| \nabla f(x^k) \right\| \leq \frac{c}{2}$

We introduce an indicative function:

$$\aleph_k = \mathbb{1} \left\{ \left\| \nabla f(x^k, \boldsymbol{\xi}^k) \right\| > c \right\}. \tag{21}$$

Then the following is true:

$$\mathbb{E} \left[ \aleph_k \right] = \mathbb{E} \left[ \aleph_k^2 \right] = \mathcal{P} \left[ \left\| \nabla f(x^k, \boldsymbol{\xi}^k) \right\| > c \right] \overset{①}{\leq} \mathcal{P} \left[ \left\| \nabla f(x^k, \boldsymbol{\xi}^k) - \nabla f(x^k) \right\| > \frac{c}{2} \right] \overset{②}{\leq} \frac{4\sigma^2}{c^2 B}, \tag{22}$$

where in ① we used $\left\| \nabla f(x^k, \boldsymbol{\xi}^k) \right\| \leq \left\| \nabla f(x^k, \boldsymbol{\xi}^k) - \nabla f(x^k) \right\| + \left\| \nabla f(x^k) \right\| \leq \left\| \nabla f(x^k, \boldsymbol{\xi}^k) - \nabla f(x^k) \right\| + \frac{c}{2}$, and in ② we used Markov's inequality.

Let $r_{k+1} = \mathbb{E}\left[\left\|x^{k+1} - x^*\right\|\right]$ and $F_{k+1} = \mathbb{E}\left[f(x^{k+1}) - f^*\right]$, then given that

$$\text{clip}_c\left(\nabla f(x^k, \boldsymbol{\xi}^k)\right) = \nabla f(x^k, \boldsymbol{\xi}^k)(1 - \aleph_k) + \frac{c}{\|\nabla f(x^k, \boldsymbol{\xi}^k)\|}\nabla f(x^k, \boldsymbol{\xi}^k)\aleph_k$$

$$= \nabla f(x^k, \boldsymbol{\xi}^k) + \left(\frac{c}{\|\nabla f(x^k, \boldsymbol{\xi}^k)\|} - 1\right)\nabla f(x^k, \boldsymbol{\xi}^k)\aleph_k$$

we get with $\eta \leq \frac{1}{4(L_0 + L_1 c)}$:

$$r_{k+1}^2 = r_k^2 - 2\eta\left\langle\mathbb{E}\left[\text{clip}_c\left(\nabla f(x^k, \boldsymbol{\xi}^k)\right)\right], x^k - x^*\right\rangle + \eta^2\mathbb{E}\left[\left\|\text{clip}_c\left(\nabla f(x^k, \boldsymbol{\xi}^k)\right)\right\|^2\right]$$

$$= r_k^2 - 2\eta\left\langle\nabla f(x^k), x^k - x^*\right\rangle - 2\eta\left\langle\mathbb{E}\left[\left(\frac{c}{\|\nabla f(x^k, \boldsymbol{\xi}^k)\|} - 1\right)\nabla f(x^k, \boldsymbol{\xi}^k)\aleph_k\right], x^k - x^*\right\rangle$$

$$+ \eta^2\mathbb{E}\left[\left\|\text{clip}_c\left(\nabla f(x^k, \boldsymbol{\xi}^k)\right)\right\|^2\right]$$

$$\overset{equation\ 6}{\leq} r_k^2 - 2\eta\left\langle\nabla f(x^k), x^k - x^*\right\rangle + 2\eta\left\|\mathbb{E}\left[\left(\frac{c}{\|\nabla f(x^k, \boldsymbol{\xi}^k)\|} - 1\right)\nabla f(x^k, \boldsymbol{\xi}^k)\aleph_k\right]\right\|\left\|x^k - x^*\right\|$$

$$+ \eta^2\mathbb{E}\left[\left\|\text{clip}_c\left(\nabla f(x^k, \boldsymbol{\xi}^k)\right)\right\|^2\right]$$

$$\overset{①}{\leq} r_k^2 - 2\eta F_k + 2\eta\left\|\mathbb{E}\left[\left(\frac{c}{\|\nabla f(x^k, \boldsymbol{\xi}^k)\|} - 1\right)\nabla f(x^k, \boldsymbol{\xi}^k)\aleph_k\right]\right\|\left\|x^0 - x^*\right\|$$

$$+ \eta^2\mathbb{E}\left[\left\|\text{clip}_c\left(\nabla f(x^k, \boldsymbol{\xi}^k)\right)\right\|^2\right]$$

$$\overset{equation\ 7}{\leq} r_k^2 - 2\eta F_k + 2\eta\left\|\mathbb{E}\left[\left(\frac{c}{\|\nabla f(x^k, \boldsymbol{\xi}^k)\|} - 1\right)\nabla f(x^k, \boldsymbol{\xi}^k)\aleph_k\right]\right\|\left\|x^0 - x^*\right\|$$

$$+ 2\eta^2\mathbb{E}\left[\left\|\text{clip}_c\left(\nabla f(x^k, \boldsymbol{\xi}^k)\right) - \nabla f(x^k)\right\|^2\right] + 2\eta^2\left\|\nabla f(x^k)\right\|^2$$

$$= r_k^2 - 2\eta F_k + 2\eta\left\|\mathbb{E}\left[\left(\frac{c}{\|\nabla f(x^k, \boldsymbol{\xi}^k)\|} - 1\right)\nabla f(x^k, \boldsymbol{\xi}^k)\aleph_k\right]\right\|R$$

$$+ 2\eta^2\mathbb{E}\left[\left\|\text{clip}_c\left(\nabla f(x^k, \boldsymbol{\xi}^k)\right) - \text{clip}_c\left(\nabla f(x^k)\right)\right\|^2\right] + 2\eta^2\left\|\nabla f(x^k)\right\|^2$$

$$\overset{②}{\leq} r_k^2 - 2\eta F_k + 2\eta\left\|\mathbb{E}\left[\left(\frac{c}{\|\nabla f(x^k, \boldsymbol{\xi}^k)\|} - 1\right)\nabla f(x^k, \boldsymbol{\xi}^k)\aleph_k\right]\right\|R$$

$$+ 2\eta^2\mathbb{E}\left[\left\|\nabla f(x^k, \boldsymbol{\xi}^k) - \nabla f(x^k)\right\|^2\right] + 2\eta^2\left\|\nabla f(x^k)\right\|^2$$

$$\leq r_k^2 - 2\eta F_k + 2\eta\left\|\mathbb{E}\left[\left(\frac{c}{\|\nabla f(x^k, \boldsymbol{\xi}^k)\|} - 1\right)\nabla f(x^k, \boldsymbol{\xi}^k)\aleph_k\right]\right\|R$$

$$+ \frac{2\eta^2\sigma^2}{B} + 2\eta^2\left\|\nabla f(x^k)\right\|^2$$

$$\overset{equation\ 10}{\leq} r_k^2 - 2\eta F_k + 2\eta\left\|\mathbb{E}\left[\left(\frac{c}{\|\nabla f(x^k, \boldsymbol{\xi}^k)\|} - 1\right)\nabla f(x^k, \boldsymbol{\xi}^k)\aleph_k\right]\right\|R$$

$$+ \frac{2\eta^2\sigma^2}{B} + 4\eta^2\left(L_0 + L_1\left\|\nabla f(x^k)\right\|\right)F_k$$

$$\leq r_k^2 - 2\eta F_k + 2\eta\left\|\mathbb{E}\left[\left(\frac{c}{\|\nabla f(x^k, \boldsymbol{\xi}^k)\|} - 1\right)\nabla f(x^k, \boldsymbol{\xi}^k)\aleph_k\right]\right\|R$$

$$+ \frac{2\eta^2\sigma^2}{B} + 4\eta^2\left(L_0 + L_1 c\right)F_k$$

$$= r_k^2 - 2\eta F_k\left(1 - 2\eta\left(L_0 + L_1 c\right)\right) + \frac{2\eta^2\sigma^2}{B}$$

$$+ 2\eta\left\|\mathbb{E}\left[\left(\frac{c}{\|\nabla f(x^k, \boldsymbol{\xi}^k)\|} - 1\right)\nabla f(x^k, \boldsymbol{\xi}^k)\aleph_k\right]\right\|R$$

$$\leq r_k^2 - \eta F_k + \frac{2\eta^2\sigma^2}{B} + 2\eta\left\|\mathbb{E}\left[\left(\frac{c}{\|\nabla f(x^k, \boldsymbol{\xi}^k)\|} - 1\right)\nabla f(x^k, \boldsymbol{\xi}^k)\aleph_k\right]\right\|R. \tag{23}$$

Let's find the upper bound of the last summand:

$$2\eta R \left\| \mathbb{E} \left[ \left( \frac{c}{\|\nabla f(x^k, \boldsymbol{\xi}^k)\|} - 1 \right) \nabla f(x^k, \boldsymbol{\xi}^k) \aleph_k \right] \right\|$$

$$\overset{equation\ 21}{\leq} 2\eta R \mathbb{E} \left[ \|\nabla f(x^k, \boldsymbol{\xi}^k)\| \cdot \left( 1 - \frac{c}{\|\nabla f(x^k, \boldsymbol{\xi}^k)\|} \right) \aleph_k \right]$$

$$\leq 2\eta R \mathbb{E} \left[ \|\nabla f(x^k, \boldsymbol{\xi}^k)\| \cdot \aleph_k \right]$$

$$\leq 2\eta R \left( \mathbb{E} \left[ \|\nabla f(x^k, \boldsymbol{\xi}^k) - \nabla f(x^k)\| \cdot \aleph_k \right] + \|\nabla f(x^k)\| \mathbb{E} [\aleph_k] \right)$$

$$\leq 2\eta R \left( \sqrt{\mathbb{E} \left[ \|\nabla f(x^k, \boldsymbol{\xi}^k) - \nabla f(x^k)\|^2 \right] \cdot \mathbb{E} [\aleph_k^2]} + \|\nabla f(x^k)\| \mathbb{E} [\aleph_k] \right)$$

$$\overset{equation\ 22}{\leq} 2\eta R \left( \frac{2\sigma^2}{cB} + \frac{c}{2} \cdot \frac{4\sigma^2}{c^2 B} \right)$$

$$= \frac{8\eta\sigma^2 R}{cB}. \tag{24}$$

Substituting into the initial formula and rearrange the summands, we obtain

$$\eta F_k \overset{equation\ 23}{\leq} r_k^2 - r_{k+1}^2 + \frac{2\eta^2\sigma^2}{B} + 2\eta \left\| \mathbb{E} \left[ \left( \frac{c}{\|\nabla f(x^k, \boldsymbol{\xi}^k)\|} - 1 \right) \nabla f(x^k, \boldsymbol{\xi}^k) \aleph_k \right] \right\| R$$

$$\overset{equation\ 24}{\leq} r_k^2 - r_{k+1}^2 + \frac{2\eta^2\sigma^2}{B} + \frac{8\eta\sigma^2 R}{cB}$$

Combining all cases we have:

$$\mathbb{E} \left[ f(x^N) \right] - f^* \leq F_N \cdot \mathbb{1} [\mathcal{T}_1] + F_N \cdot \mathbb{1} [\mathcal{T}_2]$$

$$\leq \left( 1 - \frac{\eta c}{4R} \right)^N F_0 + \frac{R^2}{\eta N} + \frac{\sigma^2 L_0 R}{c^2 B} + \frac{2\eta\sigma^2}{B} + \frac{8\sigma^2 R}{cB},$$

where $\mathcal{T}_1$ describes case $\|\nabla f(x^k)\| \geq \frac{c}{2}$, and $\mathcal{T}_2$ describes case $\|\nabla f(x^k)\| < \frac{c}{2}$.

# C NORMALIZED STOCHASTIC GRADIENT DESCENT (PROOF OF THE THEOREM 4.1)

Let's introduce the notation $G(x^k, \boldsymbol{\xi}^k) = \frac{\nabla f(x^k, \boldsymbol{\xi}^k)}{\|\nabla f(x^k, \boldsymbol{\xi}^k)\|}$, then using $(L_0, L_1)$-smoothness (see Assumption 1.2):

$$f(x^{k+1}) - f(x^k) \overset{equation\ 8}{\leq} \langle \nabla f(x^k), x^{k+1} - x^k \rangle + \frac{L_0 + L_1 \|\nabla f(x^k)\|}{2} \|x^{k+1} - x^k\|^2$$

$$= -\eta \langle \nabla f(x^k), G(x^k, \boldsymbol{\xi}^k) \rangle + \frac{\eta^2 (L_0 + L_1 \|\nabla f(x^k)\|)}{2} \|G(x^k, \boldsymbol{\xi}^k)\|^2. \tag{25}$$

Next, we consider 4 cases of the relation $\|\nabla f(x^k)\|$ and $\|\nabla f(x^k, \boldsymbol{\xi}^k)\|$ with respect to the hyperparameter $\lambda$.

## C.1 FIRST CASE: $\|\nabla f(x^k)\| \geq \lambda$ AND $\|\nabla f(x^k, \boldsymbol{\xi}^k)\| \geq \lambda$

Let us evaluate first summand of equation 25 with $\alpha = \|\nabla f(x^k)\|^{-1}$:

$$-\eta \langle \nabla f(x^k), G(x^k, \boldsymbol{\xi}^k) \rangle \overset{equation\ 5}{=} -\frac{\alpha\eta}{2} \|\nabla f(x^k)\|^2 - \frac{\eta}{2\alpha} \|G(x^k, \boldsymbol{\xi}^k)\|^2$$

$$+ \frac{\eta}{2\alpha} \|G(x^k, \boldsymbol{\xi}^k) - \alpha \nabla f(x^k)\|^2$$

$$= -\frac{\eta}{2} \left\| \nabla f(x^k) \right\| - \frac{\eta}{2\alpha} \left\| G(x^k, \boldsymbol{\xi}^k) \right\|^2$$

$$+ \frac{\eta}{2\lambda^2\alpha} \left\| \lambda G(x^k, \boldsymbol{\xi}^k) - \lambda\alpha\nabla f(x^k) \right\|^2$$

$$= -\frac{\eta}{2} \left\| \nabla f(x^k) \right\| - \frac{\eta}{2\alpha} \left\| G(x^k, \boldsymbol{\xi}^k) \right\|^2$$

$$+ \frac{\eta}{2\lambda^2\alpha} \left\| \mathrm{clip}_\lambda \left( \nabla f(x^k, \boldsymbol{\xi}^k) \right) - \mathrm{clip}_\lambda \left( \nabla f(x^k) \right) \right\|^2$$

Using that clipping is a projection on onto a convex set, namely ball with radius $\lambda$, and thus is Lipshitz operator with Lipshitz constant 1, we can obtain:

$$-\eta \left\langle \nabla f(x^k), \mathbb{E}\left[ G(x^k, \boldsymbol{\xi}^k) \right] \right\rangle \leq -\frac{\eta}{2} \left\| \nabla f(x^k) \right\| - \frac{\eta}{2\alpha} \mathbb{E}\left[ \left\| G(x^k, \boldsymbol{\xi}^k) \right\|^2 \right]$$

$$+ \frac{\eta}{2\lambda^2\alpha} \mathbb{E}\left[ \left\| \nabla f(x^k, \boldsymbol{\xi}^k) - \nabla f(x^k) \right\|^2 \right]. \qquad (26)$$

**In the case:** $0 \leq \sigma \leq \frac{\lambda}{\sqrt{2}}$. Using this in equation 26, we have the following with $\eta_k \leq \frac{\left\| \nabla f(x^k) \right\|}{2(L_0 + L_1 \left\| \nabla f(x^k) \right\|)}$:

$$\mathbb{E}\left[ f(x^{k+1}) \right] - f(x^k) \overset{equation\ 25}{\leq} -\eta \left\langle \nabla f(x^k), \mathbb{E}\left[ G(x^k, \boldsymbol{\xi}^k) \right] \right\rangle + \frac{\eta^2 (L_0 + L_1 \left\| \nabla f(x^k) \right\|)}{2} \mathbb{E}\left[ \left\| G(x^k, \boldsymbol{\xi}^k) \right\|^2 \right]$$

$$\overset{equation\ 26}{\leq} -\frac{\eta}{2} \left\| \nabla f(x^k) \right\| - \frac{\eta}{2\alpha} \mathbb{E}\left[ \left\| G(x^k, \boldsymbol{\xi}^k) \right\|^2 \right] + \frac{\eta}{2\lambda^2\alpha} \mathbb{E}\left[ \left\| \nabla f(x^k, \boldsymbol{\xi}) - \nabla f(x^k) \right\|^2 \right]$$

$$+ \frac{\eta^2 (L_0 + L_1 \left\| \nabla f(x^k) \right\|)}{2} \mathbb{E}\left[ \left\| G(x^k, \boldsymbol{\xi}^k) \right\|^2 \right]$$

$$= -\frac{\eta}{2} \left\| \nabla f(x^k) \right\| + \frac{\eta}{2\lambda^2\alpha} \mathbb{E}\left[ \left\| \nabla f(x^k, \boldsymbol{\xi}^k) - \nabla f(x^k) \right\|^2 \right]$$

$$- \frac{\eta}{2} \mathbb{E}\left[ \left\| G(x^k, \boldsymbol{\xi}^k) \right\|^2 \right] \left( 1 - \frac{\eta (L_0 + L_1 \left\| \nabla f(x^k) \right\|)}{\left\| \nabla f(x^k) \right\|} \right)$$

$$\leq -\frac{\eta}{2} \left\| \nabla f(x^k) \right\| + \frac{\eta\sigma^2}{2\lambda^2\alpha}$$

$$\leq -\frac{\eta}{2} \left\| \nabla f(x^k) \right\| + \frac{\eta}{4} \left\| \nabla f(x^k) \right\|$$

$$= -\frac{\eta}{4} \left\| \nabla f(x^k) \right\|. \qquad (27)$$

The step size will be constant, depending on the hyperparameter $\lambda$:

$$\frac{\left\| \nabla f(x^k) \right\|}{2 \left( L_0 + L_1 \left\| \nabla f(x^k) \right\| \right)} = \frac{1}{2 \left( L_0 \frac{1}{\left\| \nabla f(x^k) \right\|} + L_1 \right)} = \frac{\lambda}{2 \left( L_0 \frac{\lambda}{\left\| \nabla f(x^k) \right\|} + L_1\lambda \right)} \geq \frac{\lambda}{2 \left( L_0 + L_1\lambda \right)}.$$

Thus, $\eta_k = \eta \leq \frac{\lambda}{2(L_0 + L_1\lambda)}$.

Using the convexity assumption of the function, we have the following:

$$f(x^k) - f^* \leq \left\langle \nabla f(x^k), x^k - x^* \right\rangle$$

$$\overset{equation\ 6}{\leq} \left\| \nabla f(x^k) \right\| \left\| x^k - x^* \right\|$$

$$\leq \left\| \nabla f(x^k) \right\| \underbrace{\max_{k \in [0, N-1]} \left\| x^k - x^* \right\|}_{R}.$$

Hence we have:

$$\left\| \nabla f(x^k) \right\| \geq \frac{f(x^k) - f^*}{R}. \qquad (28)$$

Then substituting equation 28 into equation 27 we obtain:

$$\mathbb{E}\left[f(x^{k+1})\right] - f(x^k) \leq -\frac{\eta}{4}\left\|\nabla f(x^k)\right\| \leq -\frac{\eta}{4R}(f(x^k) - f^*).$$

This inequality is equivalent to the trailing inequality:

$$\mathbb{E}\left[f(x^{k+1})\right] - f^* \leq \left(1 - \frac{\eta}{4R}\right)\left(f(x^k) - f^*\right).$$

Then for $k = 0, 1, 2, ..., N-1$ iterations that satisfy the conditions $\left\|\nabla f(x^k, \boldsymbol{\xi}^k)\right\| \geq \sqrt{2}\sigma$ and $\left\|\nabla f(x^k)\right\| \geq \sqrt{2}\sigma$ NSGD shows linear convergence:

$$\mathbb{E}\left[f(x^N)\right] - f^* \leq \left(1 - \frac{\eta}{4R}\right)^N (f(x^0) - f^*).$$

**In the case:** $\frac{\lambda}{\sqrt{2}} \leq \sigma$**.** Using this in equation 26, we have the following with $\eta_k \leq \frac{\left\|\nabla f(x^k)\right\|}{2(L_0 + L_1\|\nabla f(x^k)\|)}$:

$$\mathbb{E}\left[f(x^{k+1})\right] - f(x^k) \overset{equation\ 25}{\leq} -\eta\left\langle\nabla f(x^k), \mathbb{E}\left[G(x^k, \boldsymbol{\xi}^k)\right]\right\rangle + \frac{\eta^2(L_0 + L_1\left\|\nabla f(x^k)\right\|)}{2}\mathbb{E}\left[\left\|G(x^k, \boldsymbol{\xi}^k)\right\|^2\right]$$

$$\overset{equation\ 26}{\leq} -\frac{\eta}{2}\left\|\nabla f(x^k)\right\| - \frac{\eta}{2\alpha}\mathbb{E}\left[\left\|G(x^k, \boldsymbol{\xi}^k)\right\|^2\right] + \frac{\eta}{2\lambda^2\alpha}\mathbb{E}\left[\left\|\nabla f(x^k, \boldsymbol{\xi}^k) - \nabla f(x^k)\right\|^2\right]$$

$$+ \frac{\eta^2(L_0 + L_1\left\|\nabla f(x^k)\right\|)}{2}\mathbb{E}\left[\left\|G(x^k, \boldsymbol{\xi}^k)\right\|^2\right]$$

$$= -\frac{\eta}{2}\left\|\nabla f(x^k)\right\| + \frac{\eta}{2\lambda^2\alpha}\mathbb{E}\left[\left\|\nabla f(x^k, \boldsymbol{\xi}^k) - \nabla f(x^k)\right\|^2\right]$$

$$- \frac{\eta}{2}\mathbb{E}\left[\left\|G(x^k, \boldsymbol{\xi}^k)\right\|^2\right]\left(1 - \frac{\eta(L_0 + L_1\left\|\nabla f(x^k)\right\|)}{\left\|\nabla f(x^k)\right\|}\right)$$

$$\leq -\frac{\eta}{2}\left\|\nabla f(x^k)\right\| + \frac{\eta\sigma^2}{2\lambda^2\alpha B}$$

$$\leq -\frac{\eta}{2}\left\|\nabla f(x^k)\right\| + \frac{\eta\sigma^2 M}{2\lambda^2 B}. \tag{29}$$

The step size will be constant, depending on the hyperparameter $\lambda$:

$$\frac{\left\|\nabla f(x^k)\right\|}{2\left(L_0 + L_1\left\|\nabla f(x^k)\right\|\right)} = \frac{1}{2\left(L_0\frac{1}{\|\nabla f(x^k)\|} + L_1\right)} = \frac{\lambda}{2\left(L_0\frac{\lambda}{\|\nabla f(x^k)\|} + L_1\lambda\right)} \geq \frac{\lambda}{2\left(L_0 + L_1\lambda\right)}.$$

Thus, $\eta_k = \eta \leq \frac{\lambda}{2(L_0 + L_1\lambda)}$.

Using the convexity assumption of the function, we have the following:

$$f(x^k) - f^* \leq \left\langle\nabla f(x^k), x^k - x^*\right\rangle$$

$$\overset{equation\ 6}{\leq} \left\|\nabla f(x^k)\right\|\left\|x^k - x^*\right\|$$

$$\leq \left\|\nabla f(x^k)\right\|\underbrace{\max_{k\in[0,N-1]}\left\|x^k - x^*\right\|}_{R}.$$

Hence we have:

$$\left\|\nabla f(x^k)\right\| \geq \frac{f(x^k) - f^*}{R}. \tag{30}$$

Then substituting equation 30 into equation 29 we obtain:

$$\mathbb{E}\left[f(x^{k+1})\right] - f(x^k) \leq -\frac{\eta}{2}\left\|\nabla f(x^k)\right\| + \frac{\eta\sigma^2 M}{2\lambda^2 B} \leq -\frac{\eta}{2R}(f(x^k) - f^*) + \frac{\eta\sigma^2 M}{2\lambda^2 B}.$$

This inequality is equivalent to the trailing inequality:

$$
\mathbb{E}\left[f(x^{k+1})\right] - f^* \leq \left(1 - \frac{\eta}{2R}\right)\left(f(x^k) - f^*\right) + \frac{\eta\sigma^2 M}{2\lambda^2 B}
$$

$$
\overset{equation\ 11}{\leq} \left(1 - \frac{\eta}{2R} + \frac{\eta\sigma^2(4L_1+1)}{4\lambda^2 B}\right)\left(f(x^k) - f^*\right) + \frac{\eta\sigma^2 L_0}{2\lambda^2 B}
$$

$$
\overset{①}{\leq} \left(1 - \frac{\eta}{4R}\right)\left(f(x^k) - f^*\right) + \frac{\eta\sigma^2 L_0}{2\lambda^2 B},
$$

where in ① we used $B \geq \max\left\{\frac{\sigma^2 R(4L_1+1)}{\lambda^2}, \frac{\sigma^2 L_0 R}{\lambda^2 \varepsilon}\right\}$.

Then for $k = 0, 1, 2, ..., N - 1$ iterations that satisfy the conditions $\left\|\nabla f(x^k, \boldsymbol{\xi}^k)\right\| \geq \lambda$ and $\left\|\nabla f(x^k)\right\| \geq \lambda$ and $\sigma \geq \sqrt{2}\lambda$ NSGD shows linear convergence:

$$
\mathbb{E}\left[f(x^N)\right] - f^* \leq \left(1 - \frac{\eta}{4R}\right)^N \left(f(x^0) - f^*\right) + \frac{\sigma^2 L_0 R}{\lambda^2 B}.
$$

## C.2  SECOND CASE: $\left\|\nabla f(x^k)\right\| \leq \lambda$ AND $\left\|\nabla f(x^k, \boldsymbol{\xi}^k)\right\| \geq \lambda$

Let us evaluate first summand of equation 25 with $\alpha = \lambda^{-1}$:

$$
-\eta\left\langle\nabla f(x^k), G(x^k, \boldsymbol{\xi}^k)\right\rangle \overset{equation\ 5}{=} -\frac{\alpha\eta}{2}\left\|\nabla f(x^k)\right\|^2 - \frac{\eta}{2\alpha}\left\|G(x^k, \boldsymbol{\xi}^k)\right\|^2
$$

$$
+ \frac{\eta}{2\alpha}\left\|G(x^k, \boldsymbol{\xi}^k) - \alpha\nabla f(x^k)\right\|^2
$$

$$
\leq -\frac{\eta}{2}\left\|\nabla f(x^k)\right\| - \frac{\eta}{2\alpha}\left\|G(x^k, \boldsymbol{\xi}^k)\right\|^2
$$

$$
+ \frac{\eta}{2\lambda}\left\|\lambda G(x^k, \boldsymbol{\xi}^k) - \nabla f(x^k)\right\|^2
$$

$$
= -\frac{\eta}{2}\left\|\nabla f(x^k)\right\| - \frac{\eta}{2\alpha}\left\|G(x^k, \boldsymbol{\xi}^k)\right\|^2
$$

$$
+ \frac{\eta}{2\lambda}\left\|\mathrm{clip}_\lambda\left(\nabla f(x^k, \boldsymbol{\xi}^k)\right) - \mathrm{clip}_\lambda\left(\nabla f(x^k)\right)\right\|^2
$$

Using that clipping is a projection on onto a convex set, namely ball with radius $\lambda$, and thus is Lipshitz operator with Lipshitz constant $1$, we can obtain:

$$
-\eta\left\langle\nabla f(x^k), \mathbb{E}\left[G(x^k, \boldsymbol{\xi}^k)\right]\right\rangle \leq -\frac{\eta}{2}\left\|\nabla f(x^k)\right\| - \frac{\eta}{2\alpha}\mathbb{E}\left[\left\|G(x^k, \boldsymbol{\xi}^k)\right\|^2\right]
$$

$$
+ \frac{\eta}{2\lambda}\mathbb{E}\left[\left\|\nabla f(x^k, \boldsymbol{\xi}^k) - \nabla f(x^k)\right\|^2\right]. \tag{31}
$$

Using this, we have the following with $\eta_k \leq \frac{\left\|\nabla f(x^k)\right\|}{2(L_0 + L_1\left\|\nabla f(x^k)\right\|)}$:

$$
\mathbb{E}\left[f(x^{k+1})\right] - f(x^k) \overset{equation\ 25}{\leq} -\eta\left\langle\nabla f(x^k), \mathbb{E}\left[G(x^k, \boldsymbol{\xi}^k)\right]\right\rangle + \frac{\eta^2(L_0 + L_1\left\|\nabla f(x^k)\right\|)}{2}\mathbb{E}\left[\left\|G(x^k, \boldsymbol{\xi}^k)\right\|^2\right]
$$

$$
\overset{equation\ 31}{\leq} -\frac{\eta}{2}\left\|\nabla f(x^k)\right\| - \frac{\eta}{2\alpha}\mathbb{E}\left[\left\|G(x^k, \boldsymbol{\xi}^k)\right\|^2\right] + \frac{\eta}{2\lambda}\mathbb{E}\left[\left\|\nabla f(x^k, \boldsymbol{\xi}^k) - \nabla f(x^k)\right\|^2\right]
$$

$$
+ \frac{\eta^2(L_0 + L_1\left\|\nabla f(x^k)\right\|)}{2}\mathbb{E}\left[\left\|G(x^k, \boldsymbol{\xi}^k)\right\|^2\right]
$$

$$
= -\frac{\eta}{2}\left\|\nabla f(x^k)\right\| + \frac{\eta}{2\lambda}\mathbb{E}\left[\left\|\nabla f(x^k, \boldsymbol{\xi}^k) - \nabla f(x^k)\right\|^2\right]
$$

$$
- \frac{\eta}{2\alpha}\mathbb{E}\left[\left\|G(x^k, \boldsymbol{\xi}^k)\right\|^2\right]\left(1 - \frac{\eta(L_0 + L_1\left\|\nabla f(x^k)\right\|)}{\left\|\nabla f(x^k)\right\|}\right)
$$

$$
\leq -\frac{\eta}{2}\left\|\nabla f(x^k)\right\| + \frac{\eta\sigma^2}{2\lambda B}
$$

$$\leq -\frac{\eta}{2} \left\| \nabla f(x^k) \right\| + \frac{\eta \sigma^2}{2\lambda B}. \tag{32}$$

The step size will be constant, depending on the hyperparameter $\lambda$:

$$\frac{\left\| \nabla f(x^k) \right\|}{2 \left( L_0 + L_1 \left\| \nabla f(x^k) \right\| \right)} = \frac{1}{2 \left( L_0 \frac{1}{\left\| \nabla f(x^k) \right\|} + L_1 \right)} = \frac{\lambda}{2 \left( L_0 \frac{\lambda}{\left\| \nabla f(x^k) \right\|} + L_1 \lambda \right)} \geq \frac{\lambda}{2 \left( L_0 + L_1 \lambda \right)}.$$

Thus, $\eta_k = \eta \leq \frac{\lambda}{2(L_0 + L_1 \lambda)}$.

Using the convexity assumption of the function, we have the following:

$$f(x^k) - f^* \leq \left\langle \nabla f(x^k), x^k - x^* \right\rangle$$
$$\overset{equation\ 6}{\leq} \left\| \nabla f(x^k) \right\| \left\| x^k - x^* \right\|$$
$$\leq \left\| \nabla f(x^k) \right\| \underbrace{\max_{k \in [0, N-1]} \left\| x^k - x^* \right\|}_{R}.$$

Hence we have:

$$\left\| \nabla f(x^k) \right\| \geq \frac{f(x^k) - f^*}{R}. \tag{33}$$

Then substituting equation 33 into equation 32 we obtain:

$$\mathbb{E}\left[ f(x^{k+1}) \right] - f(x^k) \leq -\frac{\eta}{2} \left\| \nabla f(x^k) \right\| + \frac{\eta \sigma^2}{2\lambda B} \leq -\frac{\eta}{2R} (f(x^k) - f^*) + \frac{\eta \sigma^2}{2\lambda B}.$$

This inequality is equivalent to the trailing inequality:

$$\mathbb{E}\left[ f(x^{k+1}) \right] - f^* \leq \left( 1 - \frac{\eta}{2R} \right) (f(x^k) - f^*) + \frac{\eta \sigma^2}{2\lambda B}.$$

Then for $k = 0, 1, 2, ..., N - 1$ iterations that satisfy the conditions $\left\| \nabla f(x^k) \right\| \leq \lambda$ and $\left\| \nabla f(x^k, \boldsymbol{\xi}^k) \right\| \geq \lambda$ NSGD shows linear convergence:

$$\mathbb{E}\left[ f(x^N) \right] - f^* \leq \left( 1 - \frac{\eta}{2R} \right)^N (f(x^0) - f^*) + \frac{\sigma^2 R}{\lambda B}.$$

## C.3 THIRD CASE: $\left\| \nabla f(x^k) \right\| \leq \lambda$ AND $\left\| \nabla f(x^k, \boldsymbol{\xi}^k) \right\| \leq \lambda$

Using this in equation 25, we have the following with $\eta_k \leq \frac{\left\| \nabla f(x^k) \right\|}{2(L_0 + L_1 \left\| \nabla f(x^k) \right\|)}$ and $\alpha = \left\| \nabla f(x^k) \right\|^{-1}$:

$$\mathbb{E}\left[ f(x^{k+1}) \right] - f(x^k) \overset{equation\ 25}{\leq} -\eta \left\langle \nabla f(x^k), \mathbb{E}\left[ G(x^k, \boldsymbol{\xi}^k) \right] \right\rangle$$
$$+ \frac{\eta^2 (L_0 + L_1 \left\| \nabla f(x^k) \right\|)}{2} \mathbb{E}\left[ \left\| G(x^k, \boldsymbol{\xi}^k) \right\|^2 \right]$$
$$= -\frac{\eta \alpha}{2} \left\| \nabla f(x^k) \right\|^2 - \frac{\eta}{2\alpha} \mathbb{E}\left[ \left\| G(x^k, \boldsymbol{\xi}^k) \right\|^2 \right]$$
$$+ \frac{\eta}{2\alpha} \mathbb{E}\left[ \left\| G(x^k, \boldsymbol{\xi}^k) - \alpha \nabla f(x^k) \right\|^2 \right]$$
$$+ \frac{\eta^2 (L_0 + L_1 \left\| \nabla f(x^k) \right\|)}{2} \mathbb{E}\left[ \left\| G(x^k, \boldsymbol{\xi}^k) \right\|^2 \right]$$
$$= -\frac{\eta}{2} \left\| \nabla f(x^k) \right\| + \frac{\eta}{2\alpha} \mathbb{E}\left[ \left\| G(x^k, \boldsymbol{\xi}^k) - \alpha \nabla f(x^k) \right\|^2 \right]$$

$$- \frac{\eta}{2} \mathbb{E} \left[ \left\| G(x^k, \boldsymbol{\xi}^k) \right\|^2 \right] \left( 1 - \frac{\eta(L_0 + L_1 \left\| \nabla f(x^k) \right\|)}{\left\| \nabla f(x^k) \right\|} \right)$$

$$\leq -\frac{\eta}{2} \left\| \nabla f(x^k) \right\| + \frac{\eta}{2\alpha} \mathbb{E} \left[ \left\| G(x^k, \boldsymbol{\xi}^k) - \alpha \nabla f(x^k) \right\|^2 \right]$$

$$\leq -\frac{\eta}{2} \left\| \nabla f(x^k) \right\| + \frac{\eta}{\alpha} \mathbb{E} \left[ \left\| G(x^k, \boldsymbol{\xi}^k) \right\|^2 + \left\| \alpha \nabla f(x^k) \right\|^2 \right]$$

$$= -\frac{\eta}{2} \left\| \nabla f(x^k) \right\| + \frac{\eta}{\alpha} \mathbb{E} \left[ \left\| \frac{\nabla f(x^k, \boldsymbol{\xi}^k)}{\left\| \nabla f(x^k, \boldsymbol{\xi}^k) \right\|} \right\|^2 + \left\| \frac{\nabla f(x^k)}{\left\| \nabla f(x^k) \right\|} \right\|^2 \right]$$

$$= -\frac{\eta}{2} \left\| \nabla f(x^k) \right\| + \frac{2\eta\lambda \left\| \nabla f(x^k) \right\|}{\lambda}$$

$$\leq -\frac{\eta}{2} \left\| \nabla f(x^k) \right\| + 2\eta\lambda. \tag{34}$$

The step size will be constant, depending on the hyperparameter $\lambda$:

$$\frac{\left\| \nabla f(x^k) \right\|}{2 \left( L_0 + L_1 \left\| \nabla f(x^k) \right\| \right)} = \frac{1}{2 \left( L_0 \frac{1}{\left\| \nabla f(x^k) \right\|} + L_1 \right)} = \frac{\lambda}{2 \left( L_0 \frac{\lambda}{\left\| \nabla f(x^k) \right\|} + L_1 \lambda \right)} \geq \frac{\lambda}{2 \left( L_0 + L_1 \lambda \right)}.$$

Thus, $\eta_k = \eta \leq \frac{\lambda}{2(L_0 + L_1 \lambda)}$.

Using the convexity assumption of the function, we have the following:

$$f(x^k) - f^* \leq \left\langle \nabla f(x^k), x^k - x^* \right\rangle$$

$$\overset{equation\ 6}{\leq} \left\| \nabla f(x^k) \right\| \left\| x^k - x^* \right\|$$

$$\leq \left\| \nabla f(x^k) \right\| \underbrace{\max_{k \in [0, N-1]} \left\| x^k - x^* \right\|}_{R}.$$

Hence we have:

$$\left\| \nabla f(x^k) \right\| \geq \frac{f(x^k) - f^*}{R}. \tag{35}$$

Then substituting equation 35 into equation 34 we obtain:

$$\mathbb{E} \left[ f(x^{k+1}) \right] - f(x^k) \leq -\frac{\eta}{2} \left\| \nabla f(x^k) \right\| + 2\eta\lambda \leq -\frac{\eta}{2R} (f(x^k) - f^*) + 2\eta\lambda.$$

This inequality is equivalent to the trailing inequality:

$$\mathbb{E} \left[ f(x^{k+1}) \right] - f^* \leq \left( 1 - \frac{\eta}{2R} \right) (f(x^k) - f^*) + 2\eta\lambda.$$

Then for $k = 0, 1, 2, ..., N-1$ iterations that satisfy the conditions $\left\| \nabla f(x^k) \right\| \leq \lambda$ NSGD shows linear convergence:

$$\mathbb{E} \left[ f(x^N) \right] - f^* \leq \left( 1 - \frac{\eta}{2R} \right)^N (f(x^0) - f^*) + \lambda R.$$

### C.4 FOURTH CASE: $\left\| \nabla f(x^k) \right\| \geq \lambda$ AND $\left\| \nabla f(x^k, \boldsymbol{\xi}^k) \right\| \leq \lambda$

Using this in equation 25, we have the following with $\eta_k \leq \frac{\left\| \nabla f(x^k) \right\|}{2(L_0 + L_1 \left\| \nabla f(x^k) \right\|)}$ and $\alpha = \lambda^{-1}$:

$$\mathbb{E} \left[ f(x^{k+1}) \right] - f(x^k) \overset{equation\ 25}{\leq} -\eta \left\langle \nabla f(x^k), \mathbb{E} \left[ G(x^k, \boldsymbol{\xi}^k) \right] \right\rangle$$

$$+ \frac{\eta^2 (L_0 + L_1 \left\| \nabla f(x^k) \right\|)}{2} \mathbb{E} \left[ \left\| G(x^k, \boldsymbol{\xi}^k) \right\|^2 \right]$$

$$
\begin{aligned}
&= -\frac{\eta\alpha}{2}\left\|\nabla f(x^k)\right\|^2 - \frac{\eta}{2\alpha}\left\|\mathbb{E}\left[G(x^k,\boldsymbol{\xi}^k)\right]\right\|^2 \\
&\quad + \frac{\eta}{2\alpha}\left\|\mathbb{E}\left[G(x^k,\boldsymbol{\xi}^k)\right] - \alpha\nabla f(x^k)\right\|^2 \\
&\quad + \frac{\eta^2(L_0 + L_1\left\|\nabla f(x^k)\right\|)}{2}\mathbb{E}\left[\left\|G(x^k,\boldsymbol{\xi}^k)\right\|^2\right] \\
&= -\frac{\eta}{2\lambda}\left\|\nabla f(x^k)\right\|^2 + \frac{\eta}{2\lambda}\left\|\mathbb{E}\left[\lambda G(x^k,\boldsymbol{\xi}^k)\right] - \nabla f(x^k)\right\|^2 \\
&\quad + \frac{\eta^2(L_0 + L_1\left\|\nabla f(x^k)\right\|)}{2} \\
&= -\frac{\eta}{2\lambda}\left\|\nabla f(x^k)\right\|^2 + \frac{\eta}{2\lambda}\left\|\mathbb{E}\left[\frac{\lambda\nabla f(x^k,\boldsymbol{\xi}^k)}{\left\|\nabla f(x^k,\boldsymbol{\xi}^k)\right\|} - \nabla f(x^k,\boldsymbol{\xi}^k)\right]\right\|^2 \\
&\quad + \frac{\eta^2(L_0 + L_1\left\|\nabla f(x^k)\right\|)}{2} \\
&= -\frac{\eta}{2\lambda}\left\|\nabla f(x^k)\right\|^2 + \frac{\eta}{2\lambda}\left\|\mathbb{E}\left[\left(\frac{\lambda}{\left\|\nabla f(x^k,\boldsymbol{\xi}^k)\right\|} - 1\right)\nabla f(x^k,\boldsymbol{\xi}^k)\right]\right\|^2 \\
&\quad + \frac{\eta^2(L_0 + L_1\left\|\nabla f(x^k)\right\|)}{2} \\
&\leq -\frac{\eta}{2\lambda}\left\|\nabla f(x^k)\right\|^2 + \frac{\eta}{2\lambda}\mathbb{E}\left[\left(\frac{\lambda}{\left\|\nabla f(x^k,\boldsymbol{\xi}^k)\right\|} - 1\right)^2\left\|\nabla f(x^k,\boldsymbol{\xi}^k)\right\|^2\right] \\
&\quad + \frac{\eta^2(L_0 + L_1\left\|\nabla f(x^k)\right\|)}{2} \\
&\leq -\frac{\eta}{2\lambda}\left\|\nabla f(x^k)\right\|^2 + \frac{\eta}{2\lambda}\mathbb{E}\left[\frac{\lambda^2}{\left\|\nabla f(x^k,\boldsymbol{\xi}^k)\right\|^2}\left\|\nabla f(x^k,\boldsymbol{\xi}^k)\right\|^2\right] \\
&\quad + \frac{\eta^2(L_0 + L_1\left\|\nabla f(x^k)\right\|)}{2} \\
&= -\frac{\eta}{2\lambda}\left\|\nabla f(x^k)\right\|^2 + \frac{\eta^2(L_0 + L_1\left\|\nabla f(x^k)\right\|)}{2} + \frac{\eta\lambda}{2} \\
&\leq -\frac{\eta}{2}\left\|\nabla f(x^k)\right\| + \frac{\eta^2(L_0 + L_1\left\|\nabla f(x^k)\right\|)}{2} + \frac{\eta\lambda}{2} \\
&= -\frac{\eta}{2}\left\|\nabla f(x^k)\right\|\left(1 - \frac{\eta(L_0 + L_1\left\|\nabla f(x^k)\right\|)}{\left\|\nabla f(x^k)\right\|}\right) + \frac{\eta\lambda}{2} \\
&\leq -\frac{\eta}{4}\left\|\nabla f(x^k)\right\| + \frac{\eta\lambda}{2}.
\end{aligned}
\tag{36}
$$

The step size will be constant, depending on the hyperparameter $\lambda$:

$$
\frac{\left\|\nabla f(x^k)\right\|}{2\left(L_0 + L_1\left\|\nabla f(x^k)\right\|\right)} = \frac{1}{2\left(L_0\frac{1}{\left\|\nabla f(x^k)\right\|} + L_1\right)} = \frac{\lambda}{2\left(L_0\frac{\lambda}{\left\|\nabla f(x^k)\right\|} + L_1\lambda\right)} \geq \frac{\lambda}{2\left(L_0 + L_1\lambda\right)}.
$$

Thus, $\eta_k = \eta \leq \frac{\lambda}{2(L_0 + L_1\lambda)}$.

Using the convexity assumption of the function, we have the following:

$$
\begin{aligned}
f(x^k) - f^* &\leq \left\langle\nabla f(x^k), x^k - x^*\right\rangle \\
&\overset{equation\ 6}{\leq} \left\|\nabla f(x^k)\right\|\left\|x^k - x^*\right\| \\
&\leq \left\|\nabla f(x^k)\right\|\underbrace{\max_{k\in[0,N-1]}\left\|x^k - x^*\right\|}_{R}.
\end{aligned}
$$

Hence we have:

$$\left\|\nabla f(x^k)\right\| \geq \frac{f(x^k) - f^*}{R}. \tag{37}$$

Then substituting equation 37 into equation 36 we obtain:

$$\mathbb{E}\left[f(x^{k+1})\right] - f(x^k) \leq -\frac{\eta}{4}\left\|\nabla f(x^k)\right\| + \frac{\eta\lambda}{2} \leq -\frac{\eta}{4R}(f(x^k) - f^*) + \frac{\eta\lambda}{2}.$$

This inequality is equivalent to the trailing inequality:

$$\mathbb{E}\left[f(x^{k+1})\right] - f^* \leq \left(1 - \frac{\eta}{4R}\right)(f(x^k) - f^*) + \frac{\eta\lambda}{2}.$$

Then for $k = 0, 1, 2, ..., N - 1$ iterations that satisfy the conditions $\left\|\nabla f(x^k)\right\| \geq \lambda$ and $\left\|\nabla f(x^k, \boldsymbol{\xi}^k)\right\| \leq \lambda$ NSGD shows linear convergence:

$$\mathbb{E}\left[f(x^N)\right] - f^* \leq \left(1 - \frac{\eta}{4R}\right)^N (f(x^0) - f^*) + 2\lambda R.$$

Combining all the cases considered, we obtain the convergence rate Normalized Stochastic Gradient Descent with batch size $B \geq \max\left\{\frac{\sigma^2 R(4L_1+1)}{\lambda^2}, \frac{\sigma^2 L_0 R}{\lambda^2 \varepsilon}\right\}$:

$$\mathbb{E}\left[f(x^N)\right] - f^* \lesssim \left(1 - \frac{\eta}{R}\right)^N (f(x^0) - f^*)) + \frac{\sigma^2 L_0 R}{B\lambda^2} + \lambda R.$$

## D   ZERO-ORDER CLIPPED STOCHASTIC GRADIENT DESCENT METHOD

This section consists of two parts: 1) a generalization of the convergence result of ClipSGD (Algorithm 1) to the biased gradient oracle $\mathbf{g}(x^k, \boldsymbol{\xi}^k) = \nabla f(x^k, \boldsymbol{\xi}^k) + \mathbf{b}(x^k)$, where $\mathbf{b}(x^k)$ is biased bounded by $\zeta \geq 0 : \left\|\mathbf{b}(x^k)\right\| \leq \zeta$; 2) deriving convergence estimates of ZO-ClipSGD directly.

### D.1   BIASED CLIPPED STOCHASTIC GRADIENT DESCENT METHOD (PROOF OF THE LEMMA 5.1)

We start by using $(L_0, L_1)$-smoothness (see Assumption 1.2):

$$f(x^{k+1}) - f(x^k) \overset{equation\ 8}{\leq} \left\langle \nabla f(x^k), x^{k+1} - x^k \right\rangle + \frac{L_0 + L_1 \left\|\nabla f(x^k)\right\|}{2} \left\|x^{k+1} - x^k\right\|^2$$

$$= -\eta \left\langle \nabla f(x^k), \mathrm{clip}_c\left(\mathbf{g}(x^k, \boldsymbol{\xi}^k)\right) \right\rangle$$

$$+ \frac{\eta^2 (L_0 + L_1 \left\|\nabla f(x^k)\right\|)}{2} \left\|\mathrm{clip}_c\left(\mathbf{g}(x^k, \boldsymbol{\xi}^k)\right)\right\|^2. \tag{38}$$

Next, we consider three cases depending on the gradient norm: $\left\|\nabla f(x^k)\right\| \geq c$ – the full gradient is clipped and $\frac{c}{3} \leq \left\|\nabla f(x^k)\right\| \leq c$ and $\left\|\nabla f(x^k)\right\| \leq \frac{c}{3}$ – the full gradient is not clipped.

#### D.1.1   FIRST CASE: $\left\|\nabla f(x^k)\right\| \geq c$

In this case $\alpha\nabla f(x^k) = \mathrm{clip}_c\left(\nabla f(x^k)\right)$ with $\alpha = \min\left\{1, \frac{c}{\|\nabla f(x^k)\|}\right\} = \frac{c}{\|\nabla f(x^k)\|}$, therefore we have the following

$$-\eta \left\langle \nabla f(x^k), \mathrm{clip}_c\left(\mathbf{g}(x^k, \boldsymbol{\xi}^k)\right) \right\rangle \overset{equation\ 5}{=} -\frac{\alpha\eta}{2}\left\|\nabla f(x^k)\right\|^2 - \frac{\eta}{2\alpha}\left\|\mathrm{clip}_c\left(\mathbf{g}(x^k, \boldsymbol{\xi}^k)\right)\right\|^2$$

$$+ \frac{\eta}{2\alpha}\left\|\mathrm{clip}_c\left(\mathbf{g}(x^k, \boldsymbol{\xi}^k)\right) - \alpha\nabla f(x^k)\right\|^2$$

$$= -\frac{\alpha\eta}{2}\left\|\nabla f(x^k)\right\|^2 - \frac{\eta}{2\alpha}\left\|\mathrm{clip}_c\left(\mathbf{g}(x^k, \boldsymbol{\xi}^k)\right)\right\|^2$$

$$+ \frac{\eta}{2\alpha} \left\| \mathrm{clip}_c \left( \mathbf{g}(x^k, \boldsymbol{\xi}^k) \right) - \mathrm{clip}_c \left( \nabla f(x^k) \right) \right\|^2$$

$$= -\frac{c\eta}{2} \left\| \nabla f(x^k) \right\| - \frac{\eta}{2\alpha} \left\| \mathrm{clip}_c \left( \mathbf{g}(x^k, \boldsymbol{\xi}^k) \right) \right\|^2$$

$$+ \frac{\eta}{2\alpha} \left\| \mathrm{clip}_c \left( \mathbf{g}(x^k, \boldsymbol{\xi}^k) \right) - \mathrm{clip}_c \left( \nabla f(x^k) \right) \right\|^2.$$

Using that clipping is a projection on onto a convex set, namely ball with radius $c$, and thus is Lipshitz operator with Lipshitz constant 1, we can obtain:

$$-\eta \left\langle \nabla f(x^k), \mathbb{E} \left[ \mathrm{clip}_c \left( \mathbf{g}(x^k, \boldsymbol{\xi}^k) \right) \right] \right\rangle \leq -\frac{c\eta}{2} \left\| \nabla f(x^k) \right\| - \frac{\eta}{2\alpha} \mathbb{E} \left[ \left\| \mathrm{clip}_c \left( \mathbf{g}(x^k, \boldsymbol{\xi}^k) \right) \right\|^2 \right]$$

$$+ \frac{\eta}{2\alpha} \mathbb{E} \left[ \left\| \mathbf{g}(x^k, \boldsymbol{\xi}^k) - \nabla f(x^k) \right\|^2 \right]$$

$$\overset{equation\ 9}{=} -\frac{c\eta}{2} \left\| \nabla f(x^k) \right\| - \frac{\eta}{2\alpha} \mathbb{E} \left[ \left\| \mathrm{clip}_c \left( \mathbf{g}(x^k, \boldsymbol{\xi}^k) \right) \right\|^2 \right]$$

$$+ \frac{\eta}{2\alpha} \mathbb{E} \left[ \left\| \mathbf{g}(x^k, \boldsymbol{\xi}^k) - \mathbb{E} \left[ \mathbf{g}(x^k, \boldsymbol{\xi}^k) \right] \right\|^2 \right]$$

$$+ \frac{\eta}{2\alpha} \left\| \mathbf{b}(x^k) \right\|^2$$

$$\leq -\frac{c\eta}{2} \left\| \nabla f(x^k) \right\| - \frac{\eta}{2\alpha} \mathbb{E} \left[ \left\| \mathrm{clip}_c \left( \nabla f(x^k, \boldsymbol{\xi}^k) \right) \right\|^2 \right]$$

$$+ \frac{\eta \sigma^2 M}{2cB} + \frac{\eta \left\| \nabla f(x^k) \right\| \zeta^2}{2c}. \tag{39}$$

We now consider the cases depending on the relation between $c$ and $\zeta$:

---

**In the case $c \geq \sqrt{2}\zeta$**   We have in equation 39:

$$-\eta \left\langle \nabla f(x^k), \mathbb{E} \left[ \mathrm{clip}_c \left( \mathbf{g}(x^k, \boldsymbol{\xi}^k) \right) \right] \right\rangle \overset{equation\ 39}{\leq} -\frac{c\eta}{2} \left\| \nabla f(x^k) \right\| - \frac{\eta}{2\alpha} \mathbb{E} \left[ \left\| \mathrm{clip}_c \left( \mathbf{g}(x^k, \boldsymbol{\xi}^k) \right) \right\|^2 \right]$$

$$+ \frac{\eta \sigma^2 M}{2cB} + \frac{\eta \left\| \nabla f(x^k) \right\| \zeta^2}{2c}$$

$$= -\frac{\eta}{2\alpha} \mathbb{E} \left[ \left\| \mathrm{clip}_c \left( \mathbf{g}(x^k, \boldsymbol{\xi}^k) \right) \right\|^2 \right]$$

$$- \frac{c\eta}{2} \left\| \nabla f(x^k) \right\| \left( 1 - \frac{\zeta^2}{c^2} \right) + \frac{\eta \sigma^2 M}{2cB}$$

$$\leq -\frac{\eta}{2\alpha} \mathbb{E} \left[ \left\| \mathrm{clip}_c \left( \mathbf{g}(x^k, \boldsymbol{\xi}^k) \right) \right\|^2 \right] - \frac{c\eta}{4} \left\| \nabla f(x^k) \right\| + \frac{\eta \sigma^2 M}{2cB}$$

$$= -\frac{\eta \left\| \nabla f(x^k) \right\|}{2c} \mathbb{E} \left[ \left\| \mathrm{clip}_c \left( \nabla f(x^k, \boldsymbol{\xi}^k) \right) \right\|^2 \right]$$

$$- \frac{c\eta}{4} \left\| \nabla f(x^k) \right\| + \frac{\eta \sigma^2 M}{2cB}.$$

Plugging this into equation 38 and choosing $\eta \leq \frac{1}{4(L_0 + L_1 c)}$ we have:

$$\mathbb{E} \left[ \nabla f(x^{k+1}) \right] - f(x^k) \overset{equation\ 13}{\leq} -\frac{\eta \left\| \nabla f(x^k) \right\|}{2c} \mathbb{E} \left[ \left\| \mathrm{clip}_c \left( \mathbf{g}(x^k, \boldsymbol{\xi}^k) \right) \right\|^2 \right] - \frac{c\eta}{4} \left\| \nabla f(x^k) \right\| + \frac{\eta \sigma^2 M}{2cB}$$

$$+ \frac{\eta^2 (L_0 + L_1 \left\| \nabla f(x^k) \right\|)}{2} \mathbb{E} \left[ \left\| \mathrm{clip}_c \left( \mathbf{g}(x^k, \boldsymbol{\xi}^k) \right) \right\|^2 \right]$$

$$= -\frac{\eta \left\| \nabla f(x^k) \right\|}{2c} \mathbb{E} \left[ \left\| \mathrm{clip}_c \left( \mathbf{g}(x^k, \boldsymbol{\xi}^k) \right) \right\|^2 \right] (1 - \eta L_1 c) - \frac{c\eta}{4} \left\| \nabla f(x^k) \right\|$$

$$+ \frac{\eta^2 L_0}{2} \mathbb{E} \left[ \left\| \mathrm{clip}_c \left( \mathbf{g}(x^k, \boldsymbol{\xi}^k) \right) \right\|^2 \right] + \frac{\eta \sigma^2 M}{2cB}$$

$$\leq -\frac{c\eta}{4} \left\| \nabla f(x^k) \right\| - \frac{\eta}{2} \mathbb{E} \left[ \left\| \mathrm{clip}_c \left( \mathbf{g}(x^k, \boldsymbol{\xi}^k) \right) \right\|^2 \right] (1 - \eta(L_0 + L_1 c))$$

$$+ \frac{\eta \sigma^2 M}{2cB}$$

$$\leq -\frac{c\eta}{4}\left\|\nabla f(x^k)\right\| + \frac{\eta\sigma^2 M}{2cB}. \tag{40}$$

Using the convexity assumption of the function, we have the following:

$$f(x^k) - f^* \leq \left\langle \nabla f(x^k), x^k - x^* \right\rangle \overset{equation\ 6}{\leq} \left\|\nabla f(x^k)\right\| \left\|x^k - x^*\right\| \leq \left\|\nabla f(x^k)\right\| \underbrace{\max_{k \in [0,N-1]} \left\|x^k - x^*\right\|}_{R}.$$

Hence we have:

$$\left\|\nabla f(x^k)\right\| \geq \frac{f(x^k) - f^*}{R}. \tag{41}$$

Then substituting equation 41 into equation 40 we obtain:

$$\mathbb{E}\left[f(x^{k+1})\right] - f(x^k) \leq -\frac{\eta c}{4}\left\|\nabla f(x^k)\right\| + \frac{\eta\sigma^2 M}{2cB} \leq -\frac{\eta c}{4R}(f(x^k) - f^*) + \frac{\eta\sigma^2 M}{2cB}.$$

This inequality is equivalent to the trailing inequality:

$$\mathbb{E}\left[f(x^{k+1})\right] - f^* \leq \left(1 - \frac{\eta c}{4R}\right)(f(x^k) - f^*) + \frac{\eta\sigma^2 M}{2cB}.$$

Then for $k = 0, 1, 2, ..., N-1$ iterations that satisfy the conditions $\left\|\nabla f(x^k)\right\| \geq c \geq \sqrt{2}\zeta$, then ClipSGD with biased gradient oracle has

$$\mathbb{E}\left[f(x^N)\right] - f^* \leq \left(1 - \frac{\eta}{2R}\right)^N (f(x^0) - f^*) + \frac{\sigma^2 MR}{cB}.$$

**In the case $c \leq \sqrt{2}\zeta$**   We have in equation 39:

$$-\eta\left\langle \nabla f(x^k), \mathbb{E}\left[\text{clip}_c\left(\mathbf{g}(x^k, \boldsymbol{\xi}^k)\right)\right]\right\rangle \overset{equation\ 39}{\leq} -\frac{c\eta}{2}\left\|\nabla f(x^k)\right\| - \frac{\eta}{2\alpha}\mathbb{E}\left[\left\|\text{clip}_c\left(\mathbf{g}(x^k, \boldsymbol{\xi}^k)\right)\right\|^2\right]$$

$$+ \frac{\eta\sigma^2 M}{2cB} + \frac{\eta\left\|\nabla f(x^k)\right\|\zeta^2}{2c}$$

$$= -\frac{c\eta}{2}\left\|\nabla f(x^k)\right\| - \frac{\eta}{2\alpha}\mathbb{E}\left[\left\|\text{clip}_c\left(\mathbf{g}(x^k, \boldsymbol{\xi}^k)\right)\right\|^2\right]$$

$$+ \frac{\eta M}{2c}\left(\frac{\sigma^2}{B} + \zeta^2\right).$$

Plugging this into equation 38 and choosing $\eta \leq \frac{1}{4(L_0 + L_1 c)}$ we have:

$$\mathbb{E}\left[f(x^{k+1})\right] - f(x^k) \overset{equation\ 38}{\leq} -\frac{c\eta}{2}\left\|\nabla f(x^k)\right\| - \frac{\eta\left\|\nabla f(x^k)\right\|}{2c}\mathbb{E}\left[\left\|\text{clip}_c\left(\mathbf{g}(x^k, \boldsymbol{\xi}^k)\right)\right\|^2\right]$$

$$+ \frac{\eta^2(L_0 + L_1\left\|\nabla f(x^k)\right\|)}{2}\mathbb{E}\left[\left\|\text{clip}_c\left(\mathbf{g}(x^k, \boldsymbol{\xi}^k)\right)\right\|^2\right] + \frac{\eta M}{2c}\left(\frac{\sigma^2}{B} + \zeta^2\right)$$

$$= -\frac{c\eta}{2}\left\|\nabla f(x^k)\right\| - \frac{\eta\left\|\nabla f(x^k)\right\|}{2c}\mathbb{E}\left[\left\|\text{clip}_c\left(\mathbf{g}(x^k, \boldsymbol{\xi}^k)\right)\right\|^2\right](1 - \eta L_1 c)$$

$$+ \frac{\eta^2 L_0}{2}\mathbb{E}\left[\left\|\text{clip}_c\left(\mathbf{g}(x^k, \boldsymbol{\xi}^k)\right)\right\|^2\right] + \frac{\eta M}{2c}\left(\frac{\sigma^2}{B} + \zeta^2\right)$$

$$\leq -\frac{c\eta}{2}\left\|\nabla f(x^k)\right\| - \frac{\eta}{2}\mathbb{E}\left[\left\|\text{clip}_c\left(\mathbf{g}(x^k, \boldsymbol{\xi}^k)\right)\right\|^2\right](1 - \eta(L_0 + L_1 c))$$

$$+ \frac{\eta M}{2c}\left(\frac{\sigma^2}{B} + \zeta^2\right)$$

$$\leq -\frac{c\eta}{2}\left\|\nabla f(x^k)\right\| + \frac{\eta M}{2c}\left(\frac{\sigma^2}{B} + \zeta^2\right). \tag{42}$$

Using the convexity assumption of the function, we have the following:

$$f(x^k) - f^* \leq \langle \nabla f(x^k), x^k - x^* \rangle \overset{equation\ 6}{\leq} \left\| \nabla f(x^k) \right\| \left\| x^k - x^* \right\| \leq \left\| \nabla f(x^k) \right\| \underbrace{\max_{k \in [0, N-1]} \left\| x^k - x^* \right\|}_{R}.$$

Hence we have:

$$\left\| \nabla f(x^k) \right\| \geq \frac{f(x^k) - f^*}{R}. \tag{43}$$

Then substituting equation 43 into equation 42 we obtain:

$$\mathbb{E}\left[ f(x^{k+1}) \right] - f(x^k) \leq -\frac{\eta c}{2} \left\| \nabla f(x^k) \right\| + \frac{\eta M}{2c} \left( \frac{\sigma^2}{B} + \zeta^2 \right) \leq -\frac{\eta c}{2R} (f(x^k) - f^*) + \frac{\eta M}{2c} \left( \frac{\sigma^2}{B} + \zeta^2 \right).$$

This inequality is equivalent to the trailing inequality:

$$\mathbb{E}\left[ f(x^{k+1}) \right] - f^* \leq \left( 1 - \frac{\eta c}{2R} \right) (f(x^k) - f^*) + \frac{\eta M}{2c} \left( \frac{\sigma^2}{B} + \zeta^2 \right).$$

Then for $k = 0, 1, 2, ..., N-1$ iterations that satisfy the conditions $\left\| \nabla f(x^k) \right\| \geq c$ and $c \leq \sqrt{2}\zeta$, then ClipSGD

$$\mathbb{E}\left[ f(x^N) \right] - f^* \leq \left( 1 - \frac{\eta c}{2R} \right)^N (f(x^0) - f^*) + \frac{MR}{c^2} \left( \frac{\sigma^2}{B} + \zeta^2 \right).$$

### D.1.2 Second case: $\frac{c}{3} \leq \left\| \nabla f(x^k) \right\| \leq c$

In this case $\nabla f(x^k) = \text{clip}_c \left( \nabla f(x^k) \right)$ with $\alpha = \min \left\{ 1, \frac{c}{\|\nabla f(x^k)\|} \right\} = 1$, therefore we have the following

$$-\eta \left\langle \nabla f(x^k), \text{clip}_c \left( \mathbf{g}(x^k, \boldsymbol{\xi}^k) \right) \right\rangle \overset{equation\ 5}{=} -\frac{\alpha \eta}{2} \left\| \nabla f(x^k) \right\|^2 - \frac{\eta}{2\alpha} \left\| \text{clip}_c \left( \mathbf{g}(x^k, \boldsymbol{\xi}^k) \right) \right\|^2$$

$$+ \frac{\eta}{2\alpha} \left\| \text{clip}_c \left( \mathbf{g}(x^k, \boldsymbol{\xi}^k) \right) - \alpha \nabla f(x^k) \right\|^2$$

$$= -\frac{\eta}{2} \left\| \nabla f(x^k) \right\|^2 - \frac{\eta}{2} \left\| \text{clip}_c \left( \mathbf{g}(x^k, \boldsymbol{\xi}^k) \right) \right\|^2$$

$$+ \frac{\eta}{2} \left\| \text{clip}_c \left( \mathbf{g}(x^k, \boldsymbol{\xi}^k) \right) - \text{clip}_c \left( \nabla f(x^k) \right) \right\|^2$$

$$\leq -\frac{c\eta}{6} \left\| \nabla f(x^k) \right\| - \frac{\eta}{2} \left\| \text{clip}_c \left( \mathbf{g}(x^k, \boldsymbol{\xi}^k) \right) \right\|^2$$

$$+ \frac{\eta}{2} \left\| \text{clip}_c \left( \mathbf{g}(x^k, \boldsymbol{\xi}^k) \right) - \text{clip}_c \left( \nabla f(x^k) \right) \right\|^2.$$

Using that clipping is a projection on onto a convex set, namely ball with radius $c$, and thus is Lipshitz operator with Lipshitz constant 1, we can obtain:

$$-\eta \left\langle \nabla f(x^k), \mathbb{E}\left[ \text{clip}_c \left( \mathbf{g}(x^k, \boldsymbol{\xi}^k) \right) \right] \right\rangle \leq -\frac{c\eta}{6} \left\| \nabla f(x^k) \right\| - \frac{\eta}{2} \mathbb{E}\left[ \left\| \text{clip}_c \left( \mathbf{g}(x^k, \boldsymbol{\xi}^k) \right) \right\|^2 \right]$$

$$+ \frac{\eta}{2} \mathbb{E}\left[ \left\| \mathbf{g}(x^k, \boldsymbol{\xi}^k) - \nabla f(x^k) \right\|^2 \right]$$

$$\overset{equation\ 9}{=} -\frac{c\eta}{6} \left\| \nabla f(x^k) \right\| - \frac{\eta}{2} \mathbb{E}\left[ \left\| \text{clip}_c \left( \mathbf{g}(x^k, \boldsymbol{\xi}^k) \right) \right\|^2 \right]$$

$$+ \frac{\eta}{2} \mathbb{E}\left[ \left\| \mathbf{g}(x^k, \boldsymbol{\xi}^k) - \mathbb{E}\left[ \mathbf{g}(x^k, \boldsymbol{\xi}^k) \right] \right\|^2 \right]$$

$$+ \frac{\eta}{2} \left\| \mathbf{b}(x^k) \right\|^2$$

$$\leq -\frac{c\eta}{6} \left\| \nabla f(x^k) \right\| - \frac{\eta}{2} \mathbb{E}\left[ \left\| \text{clip}_c \left( \mathbf{g}(x^k, \boldsymbol{\xi}^k) \right) \right\|^2 \right]$$

$$+ \frac{\eta}{2} \left( \frac{\sigma^2}{B} + \zeta^2 \right)$$

$$= -\frac{c\eta}{6} \left\| \nabla f(x^k) \right\| - \frac{\eta}{2} \mathbb{E} \left[ \left\| \text{clip}_c \left( \mathbf{g}(x^k, \boldsymbol{\xi}^k) \right) \right\|^2 \right]$$

$$+ \frac{\eta}{2} \left( \frac{\sigma^2}{B} + \zeta^2 \right).$$

Plugging this into equation 38 and choosing $\eta \leq \frac{1}{4(L_0 + L_1 c)}$ we have:

$$\mathbb{E} \left[ f(x^{k+1}) \right] - f(x^k) \overset{equation\ 38}{\leq} -\frac{c\eta}{6} \left\| \nabla f(x^k) \right\| - \frac{\eta}{2} \mathbb{E} \left[ \left\| \text{clip}_c \left( \mathbf{g}(x^k, \boldsymbol{\xi}^k) \right) \right\|^2 \right]$$

$$+ \frac{\eta^2 (L_0 + L_1 \left\| \nabla f(x^k) \right\|)}{2} \mathbb{E} \left[ \left\| \text{clip}_c \left( \mathbf{g}(x^k, \boldsymbol{\xi}^k) \right) \right\|^2 \right] + \frac{\eta}{2} \left( \frac{\sigma^2}{B} + \zeta^2 \right)$$

$$= -\frac{c\eta}{6} \left\| \nabla f(x^k) \right\| - \frac{\eta}{2} \mathbb{E} \left[ \left\| \text{clip}_c \left( \mathbf{g}(x^k, \boldsymbol{\xi}^k) \right) \right\|^2 \right] \left( 1 - \eta (L_0 + L_1 \left\| \nabla f(x^k) \right\|) \right)$$

$$+ \frac{\eta}{2} \left( \frac{\sigma^2}{B} + \zeta^2 \right)$$

$$\leq -\frac{c\eta}{6} \left\| \nabla f(x^k) \right\| + \frac{\eta}{2} \left( \frac{\sigma^2}{B} + \zeta^2 \right). \tag{44}$$

Using the convexity assumption of the function, we have the following:

$$f(x^k) - f^* \leq \left\langle \nabla f(x^k), x^k - x^* \right\rangle \overset{equation\ 6}{\leq} \left\| \nabla f(x^k) \right\| \left\| x^k - x^* \right\| \leq \left\| \nabla f(x^k) \right\| \underbrace{\max_{k \in [0, N-1]} \left\| x^k - x^* \right\|}_{R}.$$

Hence we have:

$$\left\| \nabla f(x^k) \right\| \geq \frac{f(x^k) - f^*}{R}. \tag{45}$$

Then substituting equation 45 into equation 44 we obtain:

$$\mathbb{E} \left[ f(x^{k+1}) \right] - f(x^k) \leq -\frac{\eta c}{6} \left\| \nabla f(x^k) \right\| + \frac{\eta}{2} \left( \frac{\sigma^2}{B} + \zeta^2 \right) \leq -\frac{\eta c}{6R} (f(x^k) - f^*) + \frac{\eta}{2} \left( \frac{\sigma^2}{B} + \zeta^2 \right).$$

This inequality is equivalent to the trailing inequality:

$$\mathbb{E} \left[ f(x^{k+1}) \right] - f^* \leq \left( 1 - \frac{\eta c}{6R} \right) (f(x^k) - f^*) + \frac{\eta}{2} \left( \frac{\sigma^2}{B} + \zeta^2 \right).$$

Then for $k = 0, 1, 2, ..., N-1$ iterations that satisfy the conditions $\frac{c}{2} \leq \left\| \mathbf{g}(x^k, \boldsymbol{\xi}^k) \right\| \leq c$, then ClipSGD with biased gradient oracle has

$$\mathbb{E} \left[ f(x^N) \right] - f^* \leq \left( 1 - \frac{\eta c}{6R} \right)^N (f(x^0) - f^*) + \frac{3R}{c} \left( \frac{\sigma^2}{B} + \zeta^2 \right).$$

### D.1.3 THIRD CASE: $\left\| \nabla f(x^k) \right\| \leq \frac{c}{3}$

We introduce an indicative function:

$$\aleph_k = \mathbb{1} \left\{ \left\| \mathbf{g}(x^k, \boldsymbol{\xi}^k) \right\| > c \right\}. \tag{46}$$

Then the following is true:

$$\mathbb{E} \left[ \aleph_k \right] = \mathbb{E} \left[ \aleph_k^2 \right] = \mathcal{P} \left[ \left\| \mathbf{g}(x^k, \boldsymbol{\xi}^k) \right\| > c \right] \overset{①}{\leq} \mathcal{P} \left[ \left\| \mathbf{g}(x^k, \boldsymbol{\xi}^k) - \mathbb{E} \left[ \mathbf{g}(x^k, \boldsymbol{\xi}^k) \right] \right\| > \frac{c}{3} \right] \overset{②}{\leq} \frac{9\sigma^2}{c^2 B}, \tag{47}$$

where in ① we used $\left\| \mathbf{g}(x^k, \boldsymbol{\xi}^k) \right\| \leq \left\| \mathbf{g}(x^k, \boldsymbol{\xi}^k) - \mathbb{E} \left[ \mathbf{g}(x^k, \boldsymbol{\xi}^k) \right] \right\| + \left\| \mathbb{E} \left[ \mathbf{g}(x^k, \boldsymbol{\xi}^k) \right] \right\| \leq \left\| \mathbf{g}(x^k, \boldsymbol{\xi}^k) - \mathbb{E} \left[ \mathbf{g}(x^k, \boldsymbol{\xi}^k) \right] \right\| + \frac{c}{2}$, where assume that $\zeta \leq \frac{2c}{3}$: and in ② we used Markov's inequality.

Let $r_{k+1} = \mathbb{E}\left[\left\|x^{k+1} - x^*\right\|\right]$ and $F_{k+1} = \mathbb{E}\left[f(x^{k+1}) - f^*\right]$, then given that

$$\text{clip}_c\left(\mathbf{g}(x^k, \boldsymbol{\xi}^k)\right) = \mathbf{g}(x^k, \boldsymbol{\xi}^k)(1 - \aleph_k) + \frac{c}{\|\mathbf{g}(x^k, \boldsymbol{\xi}^k)\|}\mathbf{g}(x^k, \boldsymbol{\xi}^k)\aleph_k$$

$$= \mathbf{g}(x^k, \boldsymbol{\xi}^k) + \left(\frac{c}{\|\mathbf{g}(x^k, \boldsymbol{\xi}^k)\|} - 1\right)\mathbf{g}(x^k, \boldsymbol{\xi}^k)\aleph_k$$

we get with $\eta \le \frac{1}{4(L_0 + L_1 c)}$:

$$r_{k+1}^2 = r_k^2 - 2\eta\left\langle\mathbb{E}\left[\text{clip}_c\left(\mathbf{g}(x^k, \boldsymbol{\xi}^k)\right)\right], x^k - x^*\right\rangle + \eta^2\mathbb{E}\left[\left\|\text{clip}_c\left(\mathbf{g}(x^k, \boldsymbol{\xi}^k)\right)\right\|^2\right]$$

$$= r_k^2 - 2\eta\left\langle\nabla f(x^k), x^k - x^*\right\rangle - 2\eta\left\langle\mathbb{E}\left[\left(\frac{c}{\|\mathbf{g}(x^k, \boldsymbol{\xi}^k)\|} - 1\right)\mathbf{g}(x^k, \boldsymbol{\xi}^k)\aleph_k\right], x^k - x^*\right\rangle$$

$$+ \eta^2\mathbb{E}\left[\left\|\text{clip}_c\left(\mathbf{g}(x^k, \boldsymbol{\xi}^k)\right)\right\|^2\right] + 2\eta\left\langle\mathbf{b}(x^k), x^k - x^*\right\rangle$$

$$\overset{equation\ 6}{\le} r_k^2 - 2\eta\left\langle\nabla f(x^k), x^k - x^*\right\rangle + 2\eta\left\|\mathbb{E}\left[\left(\frac{c}{\|\mathbf{g}(x^k, \boldsymbol{\xi}^k)\|} - 1\right)\mathbf{g}(x^k, \boldsymbol{\xi}^k)\aleph_k\right]\right\|\left\|x^k - x^*\right\|$$

$$+ \eta^2\mathbb{E}\left[\left\|\text{clip}_c\left(\mathbf{g}(x^k, \boldsymbol{\xi}^k)\right)\right\|^2\right] + 2\eta\left\|\mathbf{b}(x^k)\right\|\left\|x^k - x^*\right\|$$

$$\overset{①}{\le} r_k^2 - 2\eta F_k + 2\eta\left\|\mathbb{E}\left[\left(\frac{c}{\|\mathbf{g}(x^k, \boldsymbol{\xi}^k)\|} - 1\right)\mathbf{g}(x^k, \boldsymbol{\xi}^k)\aleph_k\right]\right\|\left\|x^0 - x^*\right\|$$

$$+ \eta^2\mathbb{E}\left[\left\|\text{clip}_c\left(\mathbf{g}(x^k, \boldsymbol{\xi}^k)\right)\right\|^2\right] + 2\eta\left\|\mathbf{b}(x^k)\right\|\left\|x^0 - x^*\right\|$$

$$\overset{equation\ 7}{\le} r_k^2 - 2\eta F_k + 2\eta\left\|\mathbb{E}\left[\left(\frac{c}{\|\mathbf{g}(x^k, \boldsymbol{\xi}^k)\|} - 1\right)\mathbf{g}(x^k, \boldsymbol{\xi}^k)\aleph_k\right]\right\|\left\|x^0 - x^*\right\|$$

$$+ 2\eta^2\mathbb{E}\left[\left\|\text{clip}_c\left(\mathbf{g}(x^k, \boldsymbol{\xi}^k)\right) - \nabla f(x^k)\right\|^2\right] + 2\eta^2\left\|\nabla f(x^k)\right\|^2 + 2\eta\zeta\left\|x^0 - x^*\right\|$$

$$= r_k^2 - 2\eta F_k + 2\eta\left\|\mathbb{E}\left[\left(\frac{c}{\|\mathbf{g}(x^k, \boldsymbol{\xi}^k)\|} - 1\right)\mathbf{g}(x^k, \boldsymbol{\xi}^k)\aleph_k\right]\right\|R$$

$$+ 2\eta^2\mathbb{E}\left[\left\|\text{clip}_c\left(\mathbf{g}(x^k, \boldsymbol{\xi}^k)\right) - \text{clip}_c\left(\nabla f(x^k)\right)\right\|^2\right] + 2\eta^2\left\|\nabla f(x^k)\right\|^2 + 2\eta\zeta R$$

$$\overset{②}{\le} r_k^2 - 2\eta F_k + 2\eta\left\|\mathbb{E}\left[\left(\frac{c}{\|\mathbf{g}(x^k, \boldsymbol{\xi}^k)\|} - 1\right)\mathbf{g}(x^k, \boldsymbol{\xi}^k)\aleph_k\right]\right\|R$$

$$+ 2\eta^2\mathbb{E}\left[\left\|\mathbf{g}(x^k, \boldsymbol{\xi}^k) - \nabla f(x^k)\right\|^2\right] + 2\eta^2\left\|\nabla f(x^k)\right\|^2 + 2\eta\zeta R$$

$$\overset{equation\ 9}{=} r_k^2 - 2\eta F_k + 2\eta\left\|\mathbb{E}\left[\left(\frac{c}{\|\mathbf{g}(x^k, \boldsymbol{\xi}^k)\|} - 1\right)\mathbf{g}(x^k, \boldsymbol{\xi}^k)\aleph_k\right]\right\|R$$

$$+ 2\eta^2\mathbb{E}\left[\left\|\mathbf{g}(x^k, \boldsymbol{\xi}^k) - \mathbb{E}\left[\mathbf{g}(x^k, \boldsymbol{\xi}^k)\right]\right\|^2\right] + 2\eta^2\left\|\nabla f(x^k)\right\|^2 + 2\eta\zeta R + 2\eta^2\left\|\mathbf{b}(x^k)\right\|^2$$

$$\le r_k^2 - 2\eta F_k + 2\eta\left\|\mathbb{E}\left[\left(\frac{c}{\|\mathbf{g}(x^k, \boldsymbol{\xi}^k)\|} - 1\right)\mathbf{g}(x^k, \boldsymbol{\xi}^k)\aleph_k\right]\right\|R$$

$$+ \frac{2\eta^2\sigma^2}{B} + 2\eta^2\left\|\nabla f(x^k)\right\|^2 + 2\eta\zeta R + 2\eta^2\zeta^2$$

$$\overset{equation\ 10}{\le} r_k^2 - 2\eta F_k + 2\eta\left\|\mathbb{E}\left[\left(\frac{c}{\|\mathbf{g}(x^k, \boldsymbol{\xi}^k)\|} - 1\right)\mathbf{g}(x^k, \boldsymbol{\xi}^k)\aleph_k\right]\right\|R$$

$$+ \frac{2\eta^2\sigma^2}{B} + 4\eta^2\left(L_0 + L_1\left\|\nabla f(x^k)\right\|\right)F_k + 2\eta\zeta R + 2\eta^2\zeta^2$$

$$\le r_k^2 - 2\eta F_k + 2\eta\left\|\mathbb{E}\left[\left(\frac{c}{\|\mathbf{g}(x^k, \boldsymbol{\xi}^k)\|} - 1\right)\mathbf{g}(x^k, \boldsymbol{\xi}^k)\aleph_k\right]\right\|R$$

$$+ \frac{2\eta^2\sigma^2}{B} + 4\eta^2\left(L_0 + L_1 c\right)F_k + 2\eta\zeta R + 2\eta^2\zeta^2$$

$$= r_k^2 - 2\eta F_k\left(1 - 2\eta\left(L_0 + L_1 c\right)\right) + \frac{2\eta^2\sigma^2}{B} + 2\eta\zeta R + 2\eta^2\zeta^2$$

$$+ 2\eta \left\| \mathbb{E}\left[\left(\frac{c}{\|\mathbf{g}(x^k, \boldsymbol{\xi}^k)\|} - 1\right) \mathbf{g}(x^k, \boldsymbol{\xi}^k)\aleph_k\right]\right\| R$$

$$\leq r_k^2 - \eta F_k + \frac{2\eta^2\sigma^2}{B} + 2\eta\zeta R + 2\eta^2\zeta^2$$

$$+ 2\eta \left\| \mathbb{E}\left[\left(\frac{c}{\|\mathbf{g}(x^k, \boldsymbol{\xi}^k)\|} - 1\right) \mathbf{g}(x^k, \boldsymbol{\xi}^k)\aleph_k\right]\right\| R. \tag{48}$$

Let's find the upper bound of the last summand:

$$2\eta R \left\| \mathbb{E}\left[\left(\frac{c}{\|\mathbf{g}(x^k, \boldsymbol{\xi}^k)\|} - 1\right) \mathbf{g}(x^k, \boldsymbol{\xi}^k)\aleph_k\right]\right\|$$

$$\stackrel{equation\ 46}{\leq} 2\eta R \mathbb{E}\left[\|\mathbf{g}(x^k, \boldsymbol{\xi}^k)\| \cdot \left(1 - \frac{c}{\|\mathbf{g}(x^k, \boldsymbol{\xi}^k)\|}\right)\aleph_k\right]$$

$$\leq 2\eta R \mathbb{E}\left[\|\mathbf{g}(x^k, \boldsymbol{\xi}^k)\| \cdot \aleph_k\right]$$

$$\leq 2\eta R \left(\mathbb{E}\left[\|\mathbf{g}(x^k, \boldsymbol{\xi}^k) - \mathbb{E}\left[\mathbf{g}(x^k, \boldsymbol{\xi}^k)\right]\|\right] \cdot \aleph_k\right] + \|\nabla f(x^k)\| \mathbb{E}\left[\aleph_k\right] + \|\mathbf{b}(x^k)\| \mathbb{E}\left[\aleph_k\right]\right)$$

$$\leq 2\eta R \left(\sqrt{\mathbb{E}\left[\|\mathbf{g}(x^k, \boldsymbol{\xi}^k) - \mathbb{E}\left[\mathbf{g}(x^k, \boldsymbol{\xi}^k)\right]\|^2\right] \cdot \mathbb{E}\left[\aleph_k^2\right]} + \frac{2c}{3}\mathbb{E}\left[\aleph_k\right]\right)$$

$$\stackrel{equation\ 47}{\leq} 2\eta R \left(\frac{3\sigma^2}{cB} + \frac{2c}{3} \cdot \frac{9\sigma^2}{c^2 B}\right)$$

$$= \frac{18\eta\sigma^2 R}{cB}. \tag{49}$$

Substituting into the initial formula and rearrange the summands, we obtain

$$\eta F_k \stackrel{equation\ 48}{\leq} r_k^2 - r_{k+1}^2 + \frac{2\eta^2\sigma^2}{B} + 2\eta\zeta R + 2\eta^2\zeta^2 + 2\eta \left\| \mathbb{E}\left[\left(\frac{c}{\|\nabla f(x^k, \boldsymbol{\xi}^k)\|} - 1\right) \nabla f(x^k, \boldsymbol{\xi}^k)\aleph_k\right]\right\| R$$

$$\stackrel{equation\ 49}{\leq} r_k^2 - r_{k+1}^2 + \frac{2\eta^2\sigma^2}{B} + \frac{18\eta\sigma^2 R}{cB} + 2\eta\zeta R + 2\eta^2\zeta^2$$

Combining all the cases considered, we obtain the convergence rate of ClipSGD with biased gradient oracle:

$$\mathbb{E}\left[f(x^N)\right] - f^* \leq F_N \cdot \mathbb{1}\left[\mathcal{T}_1\right] + F_N \cdot \mathbb{1}\left[\mathcal{T}_2\right]$$

$$\lesssim \left(1 - \frac{\eta c}{R}\right)^K F_0 + \frac{R^2}{\eta(N-K)} + \left(\frac{MR}{c^2} + \frac{R}{c} + \eta\right) \cdot \left(\frac{\sigma^2}{B} + \zeta^2\right) + R\zeta, \tag{50}$$

where $\mathcal{T}_1$ describes case $\|\nabla f(x^k)\| \geq \frac{c}{3}$, and $\mathcal{T}_2$ describes case $\|\nabla f(x^k)\| < \frac{c}{3}$.

## D.2 CONVERGENCE RESULTS FOR ZO-ClipSGD

In order to obtain convergence results for ZO-ClipSGD it is necessary to estimate the bias and variance of the gradient approximation equation 4.

**Bias of gradient approximation** Using the variational representation of the Euclidean norm, and definition of gradient approximation equation 4 we can write:

$$\|\mathbb{E}\left[\mathbf{g}(x, \{\xi, e\})\right] - \nabla f(x)\| = \left\| \mathbb{E}\left[\frac{d}{2\gamma}\left(\tilde{f}(x+\gamma e, \xi) - \tilde{f}(x-\gamma e, \xi)\right)e\right] - \nabla f(x)\right\|$$

$$\stackrel{\text{①}}{=} \left\| \mathbb{E}\left[\frac{d}{\gamma}\left(f(x+\gamma e, \xi) + \delta(x+\gamma e)\right)e\right] - \nabla f(x)\right\|$$

$$\stackrel{\text{②}}{\leq} \left\| \mathbb{E}\left[\frac{d}{\gamma}f(x+\gamma e, \xi)e\right] - \nabla f(x)\right\| + \frac{d\Delta}{\gamma}$$

$$\stackrel{\text{③}}{=} \|\mathbb{E}\left[\nabla f(x+\gamma u, \xi)\right] - \nabla f(x)\| + \frac{d\Delta}{\gamma}$$

$$
= \sup_{z \in S^d(1)} \mathbb{E}\left[\|\nabla_z f(x + \gamma u, \xi) - \nabla_z f(x)\|\right] + \frac{d\Delta}{\gamma}
$$

$$
\overset{equation\ 8}{\leq} \left(L_0 + L_1 \|\nabla f(x^k)\|\right) \gamma \mathbb{E}\left[\|u\|\right] + \frac{d\Delta}{\gamma}
$$

$$
\leq \left(L_0 + L_1 M\right)\gamma + \frac{d\Delta}{\gamma}, \tag{51}
$$

where $u \in B^d(1)$, ① = the equality is obtained from the fact, namely, distribution of $e$ is symmetric, ② = the inequality is obtain from bounded noise $|\delta(x)| \leq \Delta$, ③ = the equality is obtained from a version of Stokes' theorem (see Section 13.3.5, Exercise 14a, Zorich & Paniagua, 2016).

**Bounding second moment (variance) of gradient approximation** By definition gradient approximation equation 4 and Wirtinger-Poincare inequality equation 12 we have

$$
\mathbb{E}\left[\|\mathbf{g}(x, \{\xi, e\}) - \mathbb{E}\left[\mathbf{g}(x, \{\xi, e\})\right]\|^2\right]
$$

$$
\leq \mathbb{E}\left[\|\mathbf{g}(x, \{\xi, e\})\|^2\right]
$$

$$
= \frac{d^2}{4\gamma^2} \mathbb{E}\left[\left\|\left(\tilde{f}(x + \gamma e, \xi) - \tilde{f}(x - \gamma e, \xi)\right) e\right\|^2\right]
$$

$$
= \frac{d^2}{4\gamma^2} \mathbb{E}\left[\left(f(x + \gamma e, \xi) - f(x - \gamma e, \xi) + \delta(x + \gamma e) - \delta(x - \gamma e)\right)^2\right]
$$

$$
\overset{equation\ 7}{\leq} \frac{d^2}{2\gamma^2}\left(\mathbb{E}\left[\left(f(x + \gamma e, \xi) - f(x - \gamma e, \xi)\right)^2\right] + 2\Delta^2\right)
$$

$$
\overset{equation\ 12}{\leq} \frac{d^2}{2\gamma^2}\left(\frac{\gamma^2}{d}\mathbb{E}\left[\|\nabla f(x + \gamma e, \xi) + \nabla f(x - \gamma e, \xi)\|^2\right] + 2\Delta^2\right)
$$

$$
= \frac{d^2}{2\gamma^2}\left(\frac{\gamma^2}{d}\mathbb{E}\left[\|\nabla f(x + \gamma e, \xi) + \nabla f(x - \gamma e, \xi) \pm 2\nabla f(x, \xi)\|^2\right] + 2\Delta^2\right)
$$

$$
\overset{equation\ 8}{\leq} 4d\mathbb{E}\left[\|\nabla f(x, \xi)\|^2\right] + 4dL^2\gamma^2\mathbb{E}\left[\|e\|^2\right] + \frac{d^2\Delta^2}{\gamma^2}
$$

$$
\overset{①}{\leq} 4d\tilde{\sigma}^2 + 4d\left(L_0 + L_1\|\nabla f(x^k)\|\right)^2\gamma^2\mathbb{E}\left[\|e\|^2\right] + \frac{d^2\Delta^2}{\gamma^2}
$$

$$
\leq 4d\tilde{\sigma}^2 + 4d\left(L_0 + L_1 M\right)^2\gamma^2 + \frac{d^2\Delta^2}{\gamma^2}, \tag{52}
$$

where ① = the inequality is obtain from $\mathbb{E}\left[\|\nabla f(x, \xi)\|^2\right] \leq \tilde{\sigma}^2$.

### D.2.1 PROOF OF THEOREM 5.2

In order to obtain the convergence rate of ZO-ClipSGD in the convex setting, we need to substitute the obtained estimates equation 51 and equation 52 into the convergence rate of ClipSGD equation 50 instead of $\zeta$ and $\sigma^2$, respectively. Given that $\frac{MR}{c^2} + \frac{R}{c} + \eta \lesssim \frac{MR}{c^2}$ at small $c$, then the convergence of ZO-ClipSGD in the convex setup is as follows:

$$
\mathbb{E}\left[f(x^N)\right] - f^* \lesssim \underbrace{\left(1 - \frac{\eta}{R}\right)^K \left(f(x^0) - f^*\right)}_{①} + \underbrace{\frac{R^2}{\eta(N - K)}}_{②} + \underbrace{\frac{dMR\tilde{\sigma}^2}{c^2 B}}_{③} + \underbrace{\frac{dMR\left(L_0 + L_1 M\right)^2\gamma^2}{c^2 B}}_{④}
$$

$$
+ \underbrace{\frac{d^2 MR\Delta^2}{c^2 B\gamma^2}}_{⑤} + \underbrace{\frac{MR\left(L_0 + L_1 M\right)^2\gamma^2}{c^2}}_{⑥} + \underbrace{\frac{d^2 MR\Delta^2}{c^2\gamma^2}}_{⑦}
$$

$$
+ \underbrace{\left(L_0 + L_1 M\right)\gamma R}_{⑧} + \underbrace{\frac{d\Delta R}{\gamma}}_{⑨}.
$$

**From term ①**, we find the $K$:

$$① : \quad \left(1 - \frac{\eta c}{R}\right)^K (f(x^0) - f^*) \le \varepsilon \quad \Rightarrow \quad K \ge \frac{R}{\eta c} \log \frac{f(x^0) - f^*}{\varepsilon}. \tag{53}$$

**From term ②**, we find the number of iterations $N$ required for Algorithm 3 in convex setup to achieve $\varepsilon$-accuracy:

$$② : \quad \frac{R^2}{\eta(N - K)} \le \varepsilon \quad \Rightarrow \quad N \overset{equation\ 53}{\ge} \frac{R^2}{\eta \varepsilon} + \frac{R}{\eta c} \log \frac{f(x^0) - f^*}{\varepsilon};$$

$$N = \mathcal{O}\left(\frac{R^2}{\eta \varepsilon} + \frac{R}{\eta c} \log \frac{1}{\varepsilon}\right). \tag{54}$$

**From terms ③**, we find the batch size $B$:

$$③ : \quad \frac{dMR\tilde{\sigma}^2}{c^2 B} \le \varepsilon \quad \Rightarrow \quad B \ge \frac{dMR\tilde{\sigma}^2}{\varepsilon c^2};$$

$$B = \mathcal{O}\left(\frac{dMR\tilde{\sigma}^2}{\varepsilon c^2}\right). \tag{55}$$

**From terms ④, ⑥ and ⑧** we find the smoothing parameter $\gamma$:

$$④ : \quad \frac{dMR(L_0 + L_1 M)^2 \gamma^2}{c^2 B} \le \varepsilon \quad \Rightarrow \quad \gamma \le \sqrt{\frac{\varepsilon c^2 B}{dMR(L_0 + L_1 M)^2}} \overset{equation\ 55}{=} \frac{\tilde{\sigma}}{(L_0 + L_1 M)};$$

$$⑥ : \quad \frac{MR(L_0 + L_1 M)^2 \gamma^2}{c^2} \le \varepsilon \quad \Rightarrow \quad \gamma \le \frac{\sqrt{\varepsilon} c}{\sqrt{MR}(L_0 + L_1 M)};$$

$$⑧ : \quad (L_0 + L_1 M) R\gamma \le \varepsilon \quad \Rightarrow \quad \gamma \le \frac{\varepsilon}{R(L_0 + L_1 M)};$$

$$\gamma \le \frac{1}{(L_0 + L_1 M)} \min\left\{\tilde{\sigma}, \frac{\sqrt{\varepsilon} c}{\sqrt{MR}}, \frac{\varepsilon}{R}\right\} = \frac{\varepsilon}{R(L_0 + L_1 M)}. \tag{56}$$

**From the remaining terms ⑤, ⑦ and ⑨**, we find the maximum allowable level of adversarial noise $\Delta$ that still guarantees the convergence of the ZO-ClipSGD to desired accuracy $\varepsilon$ in convex setup:

$$⑤ : \quad \frac{d^2 MR\Delta^2}{c^2 B\gamma^2} \le \varepsilon \quad \Rightarrow \quad \Delta \le \frac{\sqrt{\varepsilon} c\gamma \sqrt{B}}{d\sqrt{MR}} \overset{equation\ 55, equation\ 56}{=} \frac{\varepsilon \tilde{\sigma}}{\sqrt{d}(L_0 + L_1 M) R};$$

$$⑦ : \quad \frac{d^2 MR\Delta^2}{\gamma^2 c^2} \le \varepsilon \quad \Rightarrow \quad \Delta \le \sqrt{\frac{\gamma^2 c^2 \varepsilon}{d^2 MR}} \overset{equation\ 56}{=} \frac{\varepsilon^{3/2} c}{d(L_0 + L_1 M)\sqrt{M} R^{3/2}};$$

$$⑨ : \quad \frac{d\Delta R}{\gamma} \le \varepsilon \quad \Rightarrow \quad \Delta \le \sqrt{\frac{\gamma \varepsilon}{dR}} \overset{equation\ 56}{=} \frac{\varepsilon^2}{d(L_0 + L_1 M) R^2};$$

$$\Delta \le \frac{\varepsilon}{\sqrt{d}(L_0 + L_1 M) R} \min\left\{\tilde{\sigma}, \frac{\sqrt{\varepsilon} c}{\sqrt{d}\sqrt{MR}}, \frac{\varepsilon}{\sqrt{d}R}\right\}$$

$$= \frac{\varepsilon}{\sqrt{d}(L_0 + L_1 M) R} \min\left\{\tilde{\sigma}, \frac{\varepsilon}{\sqrt{d}R}\right\}. \tag{57}$$

In this way, the ZO-ClipSGD achieves $\varepsilon$-accuracy: $\mathbb{E}\left[f(x^N) - f^*\right] \le \varepsilon$ in convex setup after

$$N \overset{equation\ 54}{=} \mathcal{O}\left(\frac{R^2}{\eta \varepsilon} + \frac{R}{\eta c} \log \frac{1}{\varepsilon}\right), \quad T = N \cdot B \overset{equation\ 54, equation\ 55}{=} \mathcal{O}\left(\frac{d\tilde{\sigma}^2 MR^2}{\varepsilon c^2 \eta}\left(\frac{1}{c} \log \frac{1}{\varepsilon} + \frac{R}{\varepsilon}\right)\right)$$

number of iterations, total number of zero-order oracle calls and at

$$\Delta \overset{equation\ 57}{\lesssim} \frac{\varepsilon}{\sqrt{d}(L_0 + L_1 M) R} \min\left\{\tilde{\sigma}, \frac{\varepsilon}{\sqrt{d}R}\right\}$$

the maximum level of noise with smoothing parameter $\frac{\varepsilon}{(L_0 + L_1 M) R}$ equation 56.

# E  ZERO-ORDER NORMALIZED STOCHASTIC GRADIENT DESCENT METHOD

This section consists of two parts: 1) a generalization of the convergence result of NSGD (Algorithm 2) to the biased gradient oracle $\mathbf{g}(x^k, \boldsymbol{\xi}^k) = \nabla f(x^k, \boldsymbol{\xi}^k) + \mathbf{b}(x^k)$, where $\mathbf{b}(x^k)$ is biased bounded by $\zeta \geq 0 : \left\| \mathbf{b}(x^k) \right\| \leq \zeta$; 2) deriving convergence estimates of ZO-NSGD directly.

## E.1  BIASED NORMALIZED STOCHASTIC GRADIENT DESCENT METHOD (PROOF OF THE LEMMA 5.3)

Let's introduce the notation $G(x^k, \boldsymbol{\xi}^k) = \frac{\mathbf{g}(x^k, \boldsymbol{\xi}^k)}{\|\mathbf{g}(x^k, \boldsymbol{\xi}^k)\|}$, then using $(L_0, L_1)$-smoothness (see Assumption 1.2):

$$
f(x^{k+1}) - f(x^k) \overset{equation\ 8}{\leq} \left\langle \nabla f(x^k), x^{k+1} - x^k \right\rangle + \frac{L_0 + L_1 \left\| \nabla f(x^k) \right\|}{2} \left\| x^{k+1} - x^k \right\|^2
$$

$$
= -\eta \left\langle \nabla f(x^k), G(x^k, \boldsymbol{\xi}^k) \right\rangle + \frac{\eta^2 (L_0 + L_1 \left\| \nabla f(x^k) \right\|)}{2} \left\| G(x^k, \boldsymbol{\xi}^k) \right\|^2. \quad (58)
$$

Next, we consider 4 cases of the relation $\left\| \nabla f(x^k) \right\|$ and $\left\| \mathbf{g}(x^k, \boldsymbol{\xi}^k) \right\|$ with respect to the hyperparameter $\lambda$.

### E.1.1  FIRST CASE: $\left\| \nabla f(x^k) \right\| \geq \lambda$ AND $\left\| \mathbf{g}(x^k, \boldsymbol{\xi}^k) \right\| \geq \lambda$

Let us evaluate first summand of equation 58 with $\alpha = \left\| \nabla f(x^k) \right\|^{-1}$:

$$
-\eta \left\langle \nabla f(x^k), G(x^k, \boldsymbol{\xi}^k) \right\rangle \overset{equation\ 5}{=} -\frac{\alpha \eta}{2} \left\| \nabla f(x^k) \right\|^2 - \frac{\eta}{2\alpha} \left\| G(x^k, \boldsymbol{\xi}^k) \right\|^2
$$

$$
+ \frac{\eta}{2\alpha} \left\| G(x^k, \boldsymbol{\xi}^k) - \alpha \nabla f(x^k) \right\|^2
$$

$$
= -\frac{\eta}{2} \left\| \nabla f(x^k) \right\| - \frac{\eta}{2\alpha} \left\| G(x^k, \boldsymbol{\xi}^k) \right\|^2
$$

$$
+ \frac{\eta}{2\lambda^2 \alpha} \left\| \lambda G(x^k, \boldsymbol{\xi}^k) - \lambda \alpha \nabla f(x^k) \right\|^2
$$

$$
= -\frac{\eta}{2} \left\| \nabla f(x^k) \right\| - \frac{\eta}{2\alpha} \left\| G(x^k, \boldsymbol{\xi}^k) \right\|^2
$$

$$
+ \frac{\eta}{2\lambda^2 \alpha} \left\| \mathrm{clip}_\lambda \left( \mathbf{g}(x^k, \boldsymbol{\xi}^k) \right) - \mathrm{clip}_\lambda \left( \nabla f(x^k) \right) \right\|^2
$$

Using that clipping is a projection on onto a convex set, namely ball with radius $\lambda$, and thus is Lipshitz operator with Lipshitz constant 1, we can obtain:

$$
-\eta \left\langle \nabla f(x^k), \mathbb{E} \left[ G(x^k, \boldsymbol{\xi}^k) \right] \right\rangle \leq -\frac{\eta}{2} \left\| \nabla f(x^k) \right\| - \frac{\eta}{2\alpha} \mathbb{E} \left[ \left\| G(x^k, \boldsymbol{\xi}^k) \right\|^2 \right]
$$

$$
+ \frac{\eta}{2\lambda^2 \alpha} \mathbb{E} \left[ \left\| \mathbf{g}(x^k, \boldsymbol{\xi}^k) - \nabla f(x^k) \right\|^2 \right]. \quad (59)
$$

**In the case:** $0 \leq \zeta \leq \frac{\lambda}{\sqrt{2}}$. Using this in equation 59, we have the following with $\eta_k \leq \frac{\left\| \nabla f(x^k) \right\|}{2(L_0 + L_1 \|\nabla f(x^k)\|)}$:

$$
\mathbb{E} \left[ f(x^{k+1}) \right] - f(x^k) \overset{equation\ 58}{\leq} -\eta \left\langle \nabla f(x^k), \mathbb{E} \left[ G(x^k, \boldsymbol{\xi}^k) \right] \right\rangle + \frac{\eta^2 (L_0 + L_1 \left\| \nabla f(x^k) \right\|)}{2} \mathbb{E} \left[ \left\| G(x^k, \boldsymbol{\xi}^k) \right\|^2 \right]
$$

$$
\overset{equation\ 59}{\leq} -\frac{\eta}{2} \left\| \nabla f(x^k) \right\| - \frac{\eta}{2\alpha} \mathbb{E} \left[ \left\| G(x^k, \boldsymbol{\xi}^k) \right\|^2 \right] + \frac{\eta}{2\lambda^2 \alpha} \mathbb{E} \left[ \left\| \mathbf{g}(x^k, \boldsymbol{\xi}^k) - \nabla f(x^k) \right\|^2 \right]
$$

$$
+ \frac{\eta^2 (L_0 + L_1 \left\| \nabla f(x^k) \right\|)}{2} \mathbb{E} \left[ \left\| G(x^k, \boldsymbol{\xi}^k) \right\|^2 \right]
$$

$$
= -\frac{\eta}{2} \left\| \nabla f(x^k) \right\| + \frac{\eta}{2\lambda^2 \alpha} \mathbb{E} \left[ \left\| \mathbf{g}(x^k, \boldsymbol{\xi}^k) - \nabla f(x^k) \right\|^2 \right]
$$

$$
- \frac{\eta}{2} \mathbb{E} \left[ \left\| G(x^k, \boldsymbol{\xi}^k) \right\|^2 \right] \left( 1 - \frac{\eta (L_0 + L_1 \left\| \nabla f(x^k) \right\|)}{\left\| \nabla f(x^k) \right\|} \right)
$$

$$\leq -\frac{\eta}{2} \left\| \nabla f(x^k) \right\| + \frac{\eta}{2\lambda^2\alpha} \mathbb{E}\left[ \left\| \mathbf{g}(x^k, \boldsymbol{\xi}^k) - \nabla f(x^k) \right\|^2 \right]$$

$$\stackrel{equation\ 9}{=} -\frac{\eta}{2} \left\| \nabla f(x^k) \right\| + \frac{\eta}{2\lambda^2\alpha} \mathbb{E}\left[ \left\| \mathbf{g}(x^k, \boldsymbol{\xi}^k) - \mathbb{E}\left[ \mathbf{g}(x^k, \boldsymbol{\xi}^k) \right] \right\| \right] + \frac{\eta}{2\lambda^2\alpha} \left\| \mathbf{b}(x^k) \right\|^2$$

$$\leq -\frac{\eta}{2} \left\| \nabla f(x^k) \right\| + \frac{\eta\sigma^2}{2\lambda^2\alpha B} + \frac{\eta\zeta^2}{2\lambda^2\alpha}$$

$$\leq -\frac{\eta}{2} \left\| \nabla f(x^k) \right\| + \frac{\eta}{4} \left\| \nabla f(x^k) \right\| + \frac{\eta\sigma^2 M}{2\lambda^2 B}$$

$$= -\frac{\eta}{4} \left\| \nabla f(x^k) \right\| + \frac{\eta\sigma^2 M}{2\lambda^2 B}. \tag{60}$$

The step size will be constant, depending on the hyperparameter $\lambda$:

$$\frac{\left\| \nabla f(x^k) \right\|}{2\left( L_0 + L_1 \left\| \nabla f(x^k) \right\| \right)} = \frac{1}{2\left( L_0 \frac{1}{\left\| \nabla f(x^k) \right\|} + L_1 \right)} = \frac{\lambda}{2\left( L_0 \frac{\lambda}{\left\| \nabla f(x^k) \right\|} + L_1\lambda \right)} \geq \frac{\lambda}{2\left( L_0 + L_1\lambda \right)}.$$

Thus, $\eta_k = \eta \leq \frac{\lambda}{2(L_0 + L_1\lambda)}$.

Using the convexity assumption of the function, we have the following:

$$f(x^k) - f^* \leq \left\langle \nabla f(x^k), x^k - x^* \right\rangle \stackrel{equation\ 6}{\leq} \left\| \nabla f(x^k) \right\| \left\| x^k - x^* \right\| \leq \left\| \nabla f(x^k) \right\| \underbrace{\max_{k \in [0, N-1]} \left\| x^k - x^* \right\|}_{R}.$$

Hence we have:

$$\left\| \nabla f(x^k) \right\| \geq \frac{f(x^k) - f^*}{R}. \tag{61}$$

Then substituting equation 61 into equation 60 we obtain:

$$\mathbb{E}\left[ f(x^{k+1}) \right] - f(x^k) \leq -\frac{\eta}{4} \left\| \nabla f(x^k) \right\| + \frac{\eta\sigma^2 M}{2\lambda^2 B} \leq -\frac{\eta}{4R}(f(x^k) - f^*) + \frac{\eta\sigma^2 M}{2\lambda^2 B}.$$

This inequality is equivalent to the trailing inequality:

$$\mathbb{E}\left[ f(x^{k+1}) \right] - f^* \leq \left( 1 - \frac{\eta}{4R} \right) (f(x^k) - f^*) + \frac{\eta\sigma^2 M}{2\lambda^2 B}.$$

Then for $k = 0, 1, 2, ..., N-1$ iterations that satisfy the conditions $\left\| \mathbf{g}(x^k, \boldsymbol{\xi}^k) \right\| \geq \sqrt{2}\zeta$ and $\left\| \nabla f(x^k) \right\| \geq \sqrt{2}\zeta$ NSGD with biased gradient oracle shows linear convergence:

$$\mathbb{E}\left[ f(x^N) \right] - f^* \leq \left( 1 - \frac{\eta}{4R} \right)^N (f(x^0) - f^*) + \frac{2\sigma^2 M R}{\lambda^2 B}.$$

**In the case:** $\frac{\lambda}{\sqrt{2}} \leq \zeta$. Using this in equation 59, we have the following with $\eta_k \leq \frac{\left\| \nabla f(x^k) \right\|}{2(L_0 + L_1 \left\| \nabla f(x^k) \right\|)}$:

$$\mathbb{E}\left[ f(x^{k+1}) \right] - f(x^k) \stackrel{equation\ 58}{\leq} -\eta \left\langle \nabla f(x^k), \mathbb{E}\left[ G(x^k, \boldsymbol{\xi}^k) \right] \right\rangle + \frac{\eta^2 (L_0 + L_1 \left\| \nabla f(x^k) \right\|)}{2} \mathbb{E}\left[ \left\| G(x^k, \boldsymbol{\xi}^k) \right\|^2 \right]$$

$$\stackrel{equation\ 59}{\leq} -\frac{\eta}{2} \left\| \nabla f(x^k) \right\| - \frac{\eta}{2\alpha} \mathbb{E}\left[ \left\| G(x^k, \boldsymbol{\xi}^k) \right\|^2 \right] + \frac{\eta}{2\lambda^2\alpha} \mathbb{E}\left[ \left\| \mathbf{g}(x^k, \boldsymbol{\xi}^k) - \nabla f(x^k) \right\|^2 \right]$$

$$+ \frac{\eta^2 (L_0 + L_1 \left\| \nabla f(x^k) \right\|)}{2} \mathbb{E}\left[ \left\| G(x^k, \boldsymbol{\xi}^k) \right\|^2 \right]$$

$$= -\frac{\eta}{2} \left\| \nabla f(x^k) \right\| + \frac{\eta}{2\lambda^2\alpha} \mathbb{E}\left[ \left\| \mathbf{g}(x^k, \boldsymbol{\xi}^k) - \nabla f(x^k) \right\|^2 \right]$$

$$- \frac{\eta}{2} \mathbb{E}\left[ \left\| G(x^k, \boldsymbol{\xi}^k) \right\|^2 \right] \left( 1 - \frac{\eta(L_0 + L_1 \left\| \nabla f(x^k) \right\|)}{\left\| \nabla f(x^k) \right\|} \right)$$

$$\leq -\frac{\eta}{2}\left\|\nabla f(x^k)\right\| + \frac{\eta}{2\lambda^2\alpha}\mathbb{E}\left[\left\|\mathbf{g}(x^k,\boldsymbol{\xi}^k) - \nabla f(x^k)\right\|^2\right]$$

$$\overset{equation\ 9}{=} -\frac{\eta}{2}\left\|\nabla f(x^k)\right\| + \frac{\eta}{2\lambda^2\alpha}\mathbb{E}\left[\left\|\mathbf{g}(x^k,\boldsymbol{\xi}^k) - \mathbb{E}\left[\mathbf{g}(x^k,\boldsymbol{\xi}^k)\right]\right\|\right] + \frac{\eta}{2\lambda^2\alpha}\left\|\mathbf{b}(x^k)\right\|^2$$

$$\leq -\frac{\eta}{2}\left\|\nabla f(x^k)\right\| + \frac{\eta\sigma^2}{2\lambda^2\alpha B} + \frac{\eta\zeta^2}{2\lambda^2\alpha}$$

$$\leq -\frac{\eta}{2}\left\|\nabla f(x^k)\right\| + \frac{\eta\sigma^2 M}{2\lambda^2 B} + \frac{\eta\zeta^2 M}{2\lambda^2}$$

$$= -\frac{\eta}{2}\left\|\nabla f(x^k)\right\| + \frac{\eta\sigma^2 M}{2\lambda^2 B} + \frac{\eta\zeta^2 M}{2\lambda^2}. \tag{62}$$

The step size will be constant, depending on the hyperparameter $\lambda$:

$$\frac{\left\|\nabla f(x^k)\right\|}{2\left(L_0 + L_1\left\|\nabla f(x^k)\right\|\right)} = \frac{1}{2\left(L_0\frac{1}{\|\nabla f(x^k)\|} + L_1\right)} = \frac{\lambda}{2\left(L_0\frac{\lambda}{\|\nabla f(x^k)\|} + L_1\lambda\right)} \geq \frac{\lambda}{2\left(L_0 + L_1\lambda\right)}.$$

Thus, $\eta_k = \eta \leq \frac{\lambda}{2(L_0 + L_1\lambda)}$.

Using the convexity assumption of the function, we have the following:

$$f(x^k) - f^* \leq \left\langle \nabla f(x^k), x^k - x^* \right\rangle \overset{equation\ 6}{\leq} \left\|\nabla f(x^k)\right\|\left\|x^k - x^*\right\| \leq \left\|\nabla f(x^k)\right\|\underbrace{\max_{k\in[0,N-1]}\left\|x^k - x^*\right\|}_{R}.$$

Hence we have:

$$\left\|\nabla f(x^k)\right\| \geq \frac{f(x^k) - f^*}{R}. \tag{63}$$

Then substituting equation 63 into equation 62 we obtain:

$$\mathbb{E}\left[f(x^{k+1})\right] - f(x^k) \leq -\frac{\eta}{2}\left\|\nabla f(x^k)\right\| + \frac{\eta\sigma^2 M}{2\lambda^2 B} + \frac{\eta\zeta^2 M}{2\lambda^2} \leq -\frac{\eta}{2R}(f(x^k) - f^*) + \frac{\eta\sigma^2 M}{2\lambda^2 B} + \frac{\eta\zeta^2 M}{2\lambda^2}.$$

This inequality is equivalent to the trailing inequality:

$$\mathbb{E}\left[f(x^{k+1})\right] - f^* \leq \left(1 - \frac{\eta}{2R}\right)\left(f(x^k) - f^*\right) + \frac{\eta\sigma^2 M}{2\lambda^2 B} + \frac{\eta\zeta^2 M}{2\lambda^2}.$$

Then for $k = 0, 1, 2, ..., N - 1$ iterations that satisfy the conditions $\left\|\mathbf{g}(x^k,\boldsymbol{\xi}^k)\right\| \geq \lambda$ and $\left\|\nabla f(x^k)\right\| \geq \lambda$ and $\zeta \geq \sqrt{2}\lambda$ NSGD with biased gradient oracle shows linear convergence:

$$\mathbb{E}\left[f(x^N)\right] - f^* \leq \left(1 - \frac{\eta}{2R}\right)^N\left(f(x^0) - f^*\right) + \frac{\sigma^2 M R}{\lambda^2 B} + \frac{\zeta^2 M R}{\lambda^2}.$$

### E.1.2  SECOND CASE: $\left\|\nabla f(x^k)\right\| \leq \lambda$ AND $\left\|\mathbf{g}(x^k,\boldsymbol{\xi}^k)\right\| \geq \lambda$

Let us evaluate first summand of equation 58 with $\alpha = \lambda^{-1}$:

$$-\eta\left\langle \nabla f(x^k), G(x^k,\boldsymbol{\xi}^k)\right\rangle \overset{equation\ 5}{=} -\frac{\alpha\eta}{2}\left\|\nabla f(x^k)\right\|^2 - \frac{\eta}{2\alpha}\left\|G(x^k,\boldsymbol{\xi}^k)\right\|^2$$

$$+ \frac{\eta}{2\alpha}\left\|G(x^k,\boldsymbol{\xi}^k) - \alpha\nabla f(x^k)\right\|^2$$

$$\leq -\frac{\eta}{2}\left\|\nabla f(x^k)\right\| - \frac{\eta}{2\alpha}\left\|G(x^k,\boldsymbol{\xi}^k)\right\|^2$$

$$+ \frac{\eta}{2\lambda}\left\|\lambda G(x^k,\boldsymbol{\xi}^k) - \nabla f(x^k)\right\|^2$$

$$= -\frac{\eta}{2}\left\|\nabla f(x^k)\right\| - \frac{\eta}{2\alpha}\left\|G(x^k,\boldsymbol{\xi}^k)\right\|^2$$

$$+ \frac{\eta}{2\lambda} \left\| \text{clip}_\lambda \left( \mathbf{g}(x^k, \boldsymbol{\xi}^k) \right) - \text{clip}_\lambda \left( \nabla f(x^k) \right) \right\|^2$$

Using that clipping is a projection on onto a convex set, namely ball with radius $\lambda$, and thus is Lipshitz operator with Lipshitz constant 1, we can obtain:

$$-\eta \left\langle \nabla f(x^k), \mathbb{E}\left[ G(x^k, \boldsymbol{\xi}^k) \right] \right\rangle \leq -\frac{\eta}{2} \left\| \nabla f(x^k) \right\| - \frac{\eta}{2\alpha} \mathbb{E}\left[ \left\| G(x^k, \boldsymbol{\xi}^k) \right\|^2 \right]$$
$$+ \frac{\eta}{2\lambda} \mathbb{E}\left[ \left\| \mathbf{g}(x^k, \boldsymbol{\xi}^k) - \nabla f(x^k) \right\|^2 \right]. \qquad (64)$$

Using this, we have the following with $\eta_k \leq \frac{\|\nabla f(x^k)\|}{2(L_0 + L_1 \|\nabla f(x^k)\|)}$:

$$\mathbb{E}\left[ f(x^{k+1}) \right] - f(x^k) \overset{equation\ 58}{\leq} -\eta \left\langle \nabla f(x^k), \mathbb{E}\left[ G(x^k, \boldsymbol{\xi}^k) \right] \right\rangle + \frac{\eta^2 (L_0 + L_1 \|\nabla f(x^k)\|)}{2} \mathbb{E}\left[ \left\| G(x^k, \boldsymbol{\xi}^k) \right\|^2 \right]$$

$$\overset{equation\ 64}{\leq} -\frac{\eta}{2} \left\| \nabla f(x^k) \right\| - \frac{\eta}{2\alpha} \mathbb{E}\left[ \left\| G(x^k, \boldsymbol{\xi}^k) \right\|^2 \right] + \frac{\eta}{2\lambda} \mathbb{E}\left[ \left\| \mathbf{g}(x^k, \boldsymbol{\xi}^k) - \nabla f(x^k) \right\|^2 \right]$$

$$+ \frac{\eta^2 (L_0 + L_1 \|\nabla f(x^k)\|)}{2} \mathbb{E}\left[ \left\| G(x^k, \boldsymbol{\xi}^k) \right\|^2 \right]$$

$$\overset{equation\ 9}{=} -\frac{\eta}{2} \left\| \nabla f(x^k) \right\| + \frac{\eta}{2\lambda} \mathbb{E}\left[ \left\| \mathbf{g}(x^k, \boldsymbol{\xi}^k) - \mathbb{E}\left[ \mathbf{g}(x^k, \boldsymbol{\xi}^k) \right] \right\|^2 \right] + \frac{\eta}{2\lambda} \left\| \mathbf{b}(x^k) \right\|^2$$

$$- \frac{\eta}{2} \mathbb{E}\left[ \left\| G(x^k, \boldsymbol{\xi}^k) \right\|^2 \right] \left( 1 - \frac{\eta (L_0 + L_1 \|\nabla f(x^k)\|)}{\|\nabla f(x^k)\|} \right)$$

$$\leq -\frac{\eta}{2} \left\| \nabla f(x^k) \right\| + \frac{\eta \sigma^2}{2\lambda B} + \frac{\eta \zeta^2}{2\lambda}. \qquad (65)$$

The step size will be constant, depending on the hyperparameter $\lambda$:

$$\frac{\|\nabla f(x^k)\|}{2 \left( L_0 + L_1 \|\nabla f(x^k)\| \right)} = \frac{1}{2 \left( L_0 \frac{1}{\|\nabla f(x^k)\|} + L_1 \right)} = \frac{\lambda}{2 \left( L_0 \frac{\lambda}{\|\nabla f(x^k)\|} + L_1 \lambda \right)} \geq \frac{\lambda}{2 \left( L_0 + L_1 \lambda \right)}.$$

Thus, $\eta_k = \eta \leq \frac{\lambda}{2(L_0 + L_1 \lambda)}$.

Using the convexity assumption of the function, we have the following:

$$f(x^k) - f^* \leq \left\langle \nabla f(x^k), x^k - x^* \right\rangle \overset{equation\ 6}{\leq} \left\| \nabla f(x^k) \right\| \left\| x^k - x^* \right\| \leq \left\| \nabla f(x^k) \right\| \underbrace{\max_{k \in [0, N-1]} \left\| x^k - x^* \right\|}_{R}.$$

Hence we have:

$$\left\| \nabla f(x^k) \right\| \geq \frac{f(x^k) - f^*}{R}. \qquad (66)$$

Then substituting equation 66 into equation 65 we obtain:

$$\mathbb{E}\left[ f(x^{k+1}) \right] - f(x^k) \leq -\frac{\eta}{2} \left\| \nabla f(x^k) \right\| + \frac{\eta \sigma^2}{2\lambda B} + \frac{\eta \zeta^2}{2\lambda} \leq -\frac{\eta}{2R} (f(x^k) - f^*) + \frac{\eta \sigma^2}{2\lambda B} + \frac{\eta \zeta^2}{2\lambda}.$$

This inequality is equivalent to the trailing inequality:

$$\mathbb{E}\left[ f(x^{k+1}) \right] - f^* \leq \left( 1 - \frac{\eta}{2R} \right) (f(x^k) - f^*) + \frac{\eta}{2\lambda} \left( \frac{\sigma^2}{B} + \zeta^2 \right).$$

Then for $k = 0, 1, 2, ..., N - 1$ iterations that satisfy the conditions $\left\| \nabla f(x^k) \right\| \leq \lambda$ and $\left\| \mathbf{g}(x^k, \boldsymbol{\xi}^k) \right\| \geq \lambda$ NSGD with biased gradient oracle shows linear convergence:

$$\mathbb{E}\left[ f(x^N) \right] - f^* \leq \left( 1 - \frac{\eta}{2R} \right)^N (f(x^0) - f^*) + \frac{R}{\lambda} \left( \frac{\sigma^2}{B} + \zeta^2 \right).$$

### E.1.3 THIRD CASE: $\left\|\nabla f(x^k)\right\| \leq \lambda$ AND $\left\|\mathbf{g}(x^k, \boldsymbol{\xi}^k)\right\| \leq \lambda$

Using this in equation 58, we have the following with $\eta_k \leq \frac{\left\|\nabla f(x^k)\right\|}{2(L_0 + L_1 \|\nabla f(x^k)\|)}$ and $\alpha = \left\|\nabla f(x^k)\right\|^{-1}$:

$$\mathbb{E}\left[f(x^{k+1})\right] - f(x^k) \overset{equation\ 58}{\leq} -\eta \left\langle \nabla f(x^k), \mathbb{E}\left[G(x^k, \boldsymbol{\xi}^k)\right]\right\rangle + \frac{\eta^2 (L_0 + L_1 \left\|\nabla f(x^k)\right\|)}{2}\mathbb{E}\left[\left\|G(x^k, \boldsymbol{\xi}^k)\right\|^2\right]$$

$$\overset{equation\ 5}{=} -\frac{\eta\alpha}{2}\left\|\nabla f(x^k)\right\|^2 - \frac{\eta}{2\alpha}\mathbb{E}\left[\left\|G(x^k, \boldsymbol{\xi}^k)\right\|^2\right] + \frac{\eta}{2\alpha}\mathbb{E}\left[\left\|G(x^k, \boldsymbol{\xi}^k) - \alpha\nabla f(x^k)\right\|^2\right]$$

$$+ \frac{\eta^2 (L_0 + L_1 \left\|\nabla f(x^k)\right\|)}{2}\mathbb{E}\left[\left\|G(x^k, \boldsymbol{\xi}^k)\right\|^2\right]$$

$$= -\frac{\eta}{2}\left\|\nabla f(x^k)\right\| + \frac{\eta}{2\alpha}\mathbb{E}\left[\left\|G(x^k, \boldsymbol{\xi}^k) - \alpha\nabla f(x^k)\right\|^2\right]$$

$$- \frac{\eta}{2}\mathbb{E}\left[\left\|G(x^k, \boldsymbol{\xi}^k)\right\|^2\right]\left(1 - \frac{\eta(L_0 + L_1 \left\|\nabla f(x^k)\right\|)}{\|\nabla f(x^k)\|}\right)$$

$$\leq -\frac{\eta}{2}\left\|\nabla f(x^k)\right\| + \frac{\eta}{2\alpha}\mathbb{E}\left[\left\|G(x^k, \boldsymbol{\xi}^k) - \alpha\nabla f(x^k)\right\|^2\right]$$

$$\leq -\frac{\eta}{2}\left\|\nabla f(x^k)\right\| + \frac{\eta}{\alpha}\mathbb{E}\left[\left\|G(x^k, \boldsymbol{\xi}^k)\right\|^2 + \left\|\alpha\nabla f(x^k)\right\|^2\right]$$

$$= -\frac{\eta}{2}\left\|\nabla f(x^k)\right\| + \frac{\eta}{\alpha}\mathbb{E}\left[\left\|\frac{\mathbf{g}(x^k, \boldsymbol{\xi}^k)}{\|\mathbf{g}(x^k, \boldsymbol{\xi}^k)\|}\right\|^2 + \left\|\frac{\nabla f(x^k)}{\|\nabla f(x^k)\|}\right\|^2\right]$$

$$= -\frac{\eta}{2}\left\|\nabla f(x^k)\right\| + \frac{2\eta\lambda\left\|\nabla f(x^k)\right\|}{\lambda}$$

$$\leq -\frac{\eta}{2}\left\|\nabla f(x^k)\right\| + 2\eta\lambda. \tag{67}$$

The step size will be constant, depending on the hyperparameter $\lambda$:

$$\frac{\left\|\nabla f(x^k)\right\|}{2\left(L_0 + L_1 \|\nabla f(x^k)\|\right)} = \frac{1}{2\left(L_0 \frac{1}{\|\nabla f(x^k)\|} + L_1\right)} = \frac{\lambda}{2\left(L_0 \frac{\lambda}{\|\nabla f(x^k)\|} + L_1\lambda\right)} \geq \frac{\lambda}{2\left(L_0 + L_1\lambda\right)}.$$

Thus, $\eta_k = \eta \leq \frac{\lambda}{2(L_0 + L_1\lambda)}$.

Using the convexity assumption of the function, we have the following:

$$f(x^k) - f^* \leq \left\langle \nabla f(x^k), x^k - x^*\right\rangle \overset{equation\ 6}{\leq} \left\|\nabla f(x^k)\right\| \left\|x^k - x^*\right\| \leq \left\|\nabla f(x^k)\right\| \underbrace{\max_{k \in [0, N-1]} \left\|x^k - x^*\right\|}_{R}.$$

Hence we have:

$$\left\|\nabla f(x^k)\right\| \geq \frac{f(x^k) - f^*}{R}. \tag{68}$$

Then substituting equation 68 into equation 67 we obtain:

$$\mathbb{E}\left[f(x^{k+1})\right] - f(x^k) \leq -\frac{\eta}{2}\left\|\nabla f(x^k)\right\| + 2\eta\lambda \leq -\frac{\eta}{2R}(f(x^k) - f^*) + 2\eta\lambda.$$

This inequality is equivalent to the trailing inequality:

$$\mathbb{E}\left[f(x^{k+1})\right] - f^* \leq \left(1 - \frac{\eta}{2R}\right)(f(x^k) - f^*) + 2\eta\lambda.$$

Then for $k = 0, 1, 2, ..., N-1$ iterations that satisfy the conditions $\left\|\nabla f(x^k)\right\| \leq \lambda$ NSGD with biased gradient oracle shows linear convergence:

$$\mathbb{E}\left[f(x^N)\right] - f^* \leq \left(1 - \frac{\eta}{2R}\right)^N (f(x^0) - f^*) + \lambda R.$$

### E.1.4 FOURTH CASE: $\left\|\nabla f(x^k)\right\| \geq \lambda$ AND $\left\|\mathbf{g}(x^k, \boldsymbol{\xi}^k)\right\| \leq \lambda$

Using this in equation 58, we have the following with $\eta_k \leq \frac{\left\|\nabla f(x^k)\right\|}{2(L_0 + L_1 \|\nabla f(x^k)\|)}$ and $\alpha = \lambda^{-1}$:

$$
\begin{aligned}
\mathbb{E}\left[f(x^{k+1})\right] - f(x^k) &\overset{equation\ 58}{\leq} -\eta \left\langle \nabla f(x^k), \mathbb{E}\left[G(x^k, \boldsymbol{\xi}^k)\right]\right\rangle \\
&\quad + \frac{\eta^2(L_0 + L_1 \left\|\nabla f(x^k)\right\|)}{2} \mathbb{E}\left[\left\|G(x^k, \boldsymbol{\xi}^k)\right\|^2\right] \\
&\overset{equation\ 5}{=} -\frac{\eta\alpha}{2}\left\|\nabla f(x^k)\right\|^2 - \frac{\eta}{2\alpha}\left\|\mathbb{E}\left[G(x^k, \boldsymbol{\xi}^k)\right]\right\|^2 \\
&\quad + \frac{\eta}{2\alpha}\left\|\mathbb{E}\left[G(x^k, \boldsymbol{\xi}^k)\right] - \alpha\nabla f(x^k)\right\|^2 \\
&\quad + \frac{\eta^2(L_0 + L_1 \left\|\nabla f(x^k)\right\|)}{2} \mathbb{E}\left[\left\|G(x^k, \boldsymbol{\xi}^k)\right\|^2\right] \\
&= -\frac{\eta}{2\lambda}\left\|\nabla f(x^k)\right\|^2 + \frac{\eta}{2\lambda}\left\|\mathbb{E}\left[\lambda G(x^k, \boldsymbol{\xi}^k)\right] - \nabla f(x^k)\right\|^2 \\
&\quad + \frac{\eta^2(L_0 + L_1 \left\|\nabla f(x^k)\right\|)}{2} \\
&= -\frac{\eta}{2\lambda}\left\|\nabla f(x^k)\right\|^2 + \frac{\eta}{\lambda}\left\|\mathbb{E}\left[\frac{\lambda\mathbf{g}(x^k, \boldsymbol{\xi}^k)}{\|\mathbf{g}(x^k, \boldsymbol{\xi}^k)\|} - \mathbf{g}(x^k, \boldsymbol{\xi}^k)\right]\right\|^2 \\
&\quad + \frac{\eta}{\lambda}\left\|\mathbf{b}(x^k)\right\|^2 + \frac{\eta^2(L_0 + L_1 \left\|\nabla f(x^k)\right\|)}{2} \\
&= -\frac{\eta}{2\lambda}\left\|\nabla f(x^k)\right\|^2 + \frac{\eta}{2\lambda}\left\|\mathbb{E}\left[\left(\frac{\lambda}{\|\mathbf{g}(x^k, \boldsymbol{\xi}^k)\|} - 1\right)\mathbf{g}(x^k, \boldsymbol{\xi}^k)\right]\right\|^2 \\
&\quad + \frac{\eta}{\lambda}\left\|\mathbf{b}(x^k)\right\|^2 + \frac{\eta^2(L_0 + L_1 \left\|\nabla f(x^k)\right\|)}{2} \\
&\leq -\frac{\eta}{2\lambda}\left\|\nabla f(x^k)\right\|^2 + \frac{\eta}{2\lambda}\mathbb{E}\left[\left(\frac{\lambda}{\|\mathbf{g}(x^k, \boldsymbol{\xi}^k)\|} - 1\right)^2 \left\|\mathbf{g}(x^k, \boldsymbol{\xi}^k)\right\|^2\right] \\
&\quad + \frac{\eta}{\lambda}\left\|\mathbf{b}(x^k)\right\|^2 + \frac{\eta^2(L_0 + L_1 \left\|\nabla f(x^k)\right\|)}{2} \\
&\leq -\frac{\eta}{2\lambda}\left\|\nabla f(x^k)\right\|^2 + \frac{\eta}{2\lambda}\mathbb{E}\left[\frac{\lambda^2}{\|\mathbf{g}(x^k, \boldsymbol{\xi}^k)\|^2}\left\|\mathbf{g}(x^k, \boldsymbol{\xi}^k)\right\|^2\right] \\
&\quad + \frac{\eta}{\lambda}\left\|\mathbf{b}(x^k)\right\|^2 + \frac{\eta^2(L_0 + L_1 \left\|\nabla f(x^k)\right\|)}{2} \\
&= -\frac{\eta}{2\lambda}\left\|\nabla f(x^k)\right\|^2 + \frac{\eta^2(L_0 + L_1 \left\|\nabla f(x^k)\right\|)}{2} + \frac{\eta\lambda}{2} + \frac{\eta}{\lambda}\left\|\mathbf{b}(x^k)\right\|^2 \\
&\leq -\frac{\eta}{2}\left\|\nabla f(x^k)\right\| + \frac{\eta^2(L_0 + L_1 \left\|\nabla f(x^k)\right\|)}{2} + \frac{\eta\lambda}{2} + \frac{\eta}{\lambda}\left\|\mathbf{b}(x^k)\right\|^2 \\
&= -\frac{\eta}{2}\left\|\nabla f(x^k)\right\|\left(1 - \frac{\eta(L_0 + L_1 \left\|\nabla f(x^k)\right\|)}{\|\nabla f(x^k)\|}\right) + \frac{\eta\lambda}{2} + \frac{\eta}{\lambda}\left\|\mathbf{b}(x^k)\right\|^2 \\
&\leq -\frac{\eta}{4}\left\|\nabla f(x^k)\right\| + \frac{\eta\lambda}{2} + \frac{\eta\zeta^2}{\lambda}. \quad (69)
\end{aligned}
$$

The step size will be constant, depending on the hyperparameter $\lambda$:

$$
\frac{\left\|\nabla f(x^k)\right\|}{2\left(L_0 + L_1 \|\nabla f(x^k)\|\right)} = \frac{1}{2\left(L_0 \frac{1}{\|\nabla f(x^k)\|} + L_1\right)} = \frac{\lambda}{2\left(L_0 \frac{\lambda}{\|\nabla f(x^k)\|} + L_1\lambda\right)} \geq \frac{\lambda}{2\left(L_0 + L_1\lambda\right)}.
$$

Thus, $\eta_k = \eta \leq \frac{\lambda}{2(L_0 + L_1\lambda)}$.

Using the convexity assumption of the function, we have the following:

$$f(x^k) - f^* \leq \langle \nabla f(x^k), x^k - x^* \rangle \overset{equation\ 6}{\leq} \left\| \nabla f(x^k) \right\| \left\| x^k - x^* \right\| \leq \left\| \nabla f(x^k) \right\| \underbrace{\max_{k \in [0, N-1]} \left\| x^k - x^* \right\|}_{R}.$$

Hence we have:

$$\left\| \nabla f(x^k) \right\| \geq \frac{f(x^k) - f^*}{R}. \tag{70}$$

Then substituting equation 70 into equation 69 we obtain:

$$\mathbb{E}\left[ f(x^{k+1}) \right] - f(x^k) \leq -\frac{\eta}{4} \left\| \nabla f(x^k) \right\| + \frac{\eta \lambda}{2} + \frac{\eta \zeta^2}{\lambda} \leq -\frac{\eta}{4R}(f(x^k) - f^*) + \frac{\eta \lambda}{2} + \frac{\eta \zeta^2}{\lambda}.$$

This inequality is equivalent to the trailing inequality:

$$\mathbb{E}\left[ f(x^{k+1}) \right] - f^* \leq \left( 1 - \frac{\eta}{4R} \right) \left( f(x^k) - f^* \right) + \frac{\eta \lambda}{2} + \frac{\eta \zeta^2}{\lambda}.$$

Then for $k = 0, 1, 2, ..., N - 1$ iterations that satisfy the conditions $\left\| \nabla f(x^k) \right\| \geq \lambda$ and $\left\| \mathbf{g}(x^k, \boldsymbol{\xi}^k) \right\| \leq \lambda$ NSGD with biased gradient oracle shows linear convergence:

$$\mathbb{E}\left[ f(x^N) \right] - f^* \leq \left( 1 - \frac{\eta}{4R} \right)^N (f(x^0) - f^*) + 2\lambda R + \frac{2\zeta^2 R}{\lambda}.$$

Combining all the cases considered, we obtain the convergence rate of NSGD with biased gradient oracle:

$$\mathbb{E}\left[ f(x^N) \right] - f^* \lesssim \left( 1 - \frac{\eta}{R} \right)^N (f(x^0) - f^*) + \frac{MR}{\lambda^2} \left( \frac{\sigma^2}{B} + \zeta^2 \right) + \lambda R. \tag{71}$$

## E.2 CONVERGENCE RESULTS FOR ZO-NSGD (PROOF OF THE THEOREM 5.4)

In order to obtain the convergence rate of ZO-NSGD in the convex setting, we need to substitute the obtained estimates equation 51 and equation 52 into the convergence rate of NSGD equation 71 instead of $\zeta$ and $\sigma^2$, respectively. Then the convergence of ZO-NSGD in the convex setup is as follows:

$$\mathbb{E}\left[ f(x^N) \right] - f^* \lesssim \underbrace{\left( 1 - \frac{\eta}{R} \right)^N (f(x^0) - f^*)}_{①} + \underbrace{\frac{dMR\tilde{\sigma}^2}{\lambda^2 B}}_{②} + \underbrace{\frac{dMR \left( L_0 + L_1 M \right)^2 \gamma^2}{\lambda^2 B}}_{③} + \underbrace{\frac{d^2 MR\Delta^2}{\lambda^2 B \gamma^2}}_{④}$$

$$+ \underbrace{\frac{MR \left( L_0 + L_1 M \right)^2 \gamma^2}{\lambda^2}}_{⑤} + \underbrace{\frac{d^2 MR\Delta^2}{\lambda^2 \gamma^2}}_{⑥} + \underbrace{\lambda R}_{⑦}.$$

**From term ⑦**, we find the hyperparameter $\lambda$:

$$① : \quad \lambda R \leq \varepsilon \quad \Rightarrow \quad \lambda \leq \frac{\varepsilon}{R}. \tag{72}$$

**From term ①**, we find the number of iterations $N$ required for Algorithm 4 in convex setup to achieve $\varepsilon$-accuracy:

$$① : \quad \left( 1 - \frac{\eta}{R} \right)^N (f(x^0) - f^*) \leq \varepsilon \quad \Rightarrow \quad N \geq \frac{R}{\eta} \log \frac{(f(x^0) - f^*)}{\varepsilon};$$

$$N = \tilde{\mathcal{O}} \left( \frac{R}{\eta} \right). \tag{73}$$

**From terms ②**, we find the batch size $B$:

$$② : \quad \frac{dMR\tilde{\sigma}^2}{\lambda^2 B} \leq \varepsilon \quad \Rightarrow \quad B \overset{equation\ 72}{\geq} \frac{dMR^3 \tilde{\sigma}^2}{\varepsilon^3};$$

$$B = \mathcal{O}\left(\frac{dMR^3\tilde{\sigma}^2}{\varepsilon^3}\right). \tag{74}$$

**From terms ③ and ⑤** we find the smoothing parameter $\gamma$:

$$③: \quad \frac{dMR\left(L_0 + L_1M\right)^2\gamma^2}{\lambda^2 B} \leq \varepsilon \quad \Rightarrow \quad \gamma \leq \sqrt{\frac{\varepsilon\lambda^2 B}{dMR\left(L_0 + L_1M\right)^2}} \stackrel{equation\ 74, equation\ 72}{=} \frac{\tilde{\sigma}}{(L_0 + L_1M)};$$

$$⑤: \quad \frac{MR\left(L_0 + L_1M\right)^2\gamma^2}{\lambda^2} \leq \varepsilon \quad \Rightarrow \quad \gamma \leq \frac{\sqrt{\varepsilon^3}}{\sqrt{M}R^{3/2}\left(L_0 + L_1M\right)};$$

$$\gamma \leq \frac{1}{(L_0 + L_1M)}\min\left\{\tilde{\sigma}, \frac{\varepsilon^{3/2}}{\sqrt{M}R^{3/2}}\right\} = \frac{\varepsilon^{3/2}}{(L_0 + L_1M)\sqrt{M}R^{3/2}}. \tag{75}$$

**From the remaining terms ④ and ⑥**, we find the maximum allowable level of adversarial noise $\Delta$ that still guarantees the convergence of the ZO-NSGD to desired accuracy $\varepsilon$ in convex setup:

$$④: \quad \frac{d^2MR\Delta^2}{\lambda^2 B\gamma^2} \leq \varepsilon \quad \Rightarrow \quad \Delta \leq \frac{\sqrt{\varepsilon}\lambda\gamma\sqrt{B}}{d\sqrt{MR}} \stackrel{equation\ 74, equation\ 75, equation\ 72}{=} \frac{\varepsilon^{3/2}\tilde{\sigma}}{\sqrt{d}\left(L_0 + L_1M\right)R^{3/2}};$$

$$⑥: \quad \frac{d^2MR\Delta^2}{\gamma^2\lambda^2} \leq \varepsilon \quad \Rightarrow \quad \Delta \leq \sqrt{\frac{\gamma^2\lambda^2\varepsilon}{d^2MR}} \stackrel{equation\ 72, equation\ 75}{=} \frac{\varepsilon^3}{d\left(L_0 + L_1M\right)R^3};$$

$$\Delta \leq \frac{\varepsilon^{3/2}}{\sqrt{d}\left(L_0 + L_1M\right)R^{3/2}}\min\left\{\tilde{\sigma}, \frac{\varepsilon^{3/2}}{\sqrt{d}R^{3/2}}\right\}. \tag{76}$$

In this way, the ZO-NSGD achieves $\varepsilon$-accuracy: $\mathbb{E}\left[f(x^N) - f^*\right] \leq \varepsilon$ in convex setup after

$$N \stackrel{equation\ 73}{=} \tilde{\mathcal{O}}\left(\frac{R}{\eta}\right), \quad T = N \cdot B \stackrel{equation\ 73, equation\ 74}{=} \mathcal{O}\left(\frac{d\tilde{\sigma}^2 MR^4}{\varepsilon^3\eta}\right)$$

number of iterations, total number of zero-order oracle calls and at

$$\Delta \stackrel{equation\ 76}{\lesssim} \frac{\varepsilon^{3/2}}{\sqrt{d}\left(L_0 + L_1M\right)R^{3/2}}\min\left\{\tilde{\sigma}, \frac{\varepsilon^{3/2}}{\sqrt{d}R^{3/2}}\right\}$$

the maximum level of noise with smoothing parameter $\frac{\varepsilon^{3/2}}{(L_0 + L_1M)\sqrt{M}R^{3/2}}$ equation 75.

# F ADDITIONAL CLARIFICATION

In this section, we would like to clarify the convergence in the case $L_0 = 0$ (Remark 1.3). In this case the problem does not reach a minimum (hence $R = \arg\inf f(x) = +\infty$). Therefore, we exemplify the special case of NSGD (when $\left\|\nabla f(x^k, \boldsymbol{\xi}^k)\right\| \geq \sqrt{2}\sigma$ and $\left\|\nabla f(x^k)\right\| \geq \sqrt{2}\sigma$), shows that it is possible to achieve the desired accuracy $\varepsilon$ in a finite number of iterations.

Let's introduce the notation $G(x^k, \boldsymbol{\xi}^k) = \frac{\nabla f(x^k, \boldsymbol{\xi}^k)}{\left\|\nabla f(x^k, \boldsymbol{\xi}^k)\right\|}$, then using $(L_0, L_1)$-smoothness (see Assumption 1.2):

$$f(x^{k+1}) - f(x^k) \stackrel{equation\ 8}{\leq} \left\langle\nabla f(x^k), x^{k+1} - x^k\right\rangle + \frac{L_0 + L_1\left\|\nabla f(x^k)\right\|}{2}\left\|x^{k+1} - x^k\right\|^2$$

$$= -\eta\left\langle\nabla f(x^k), G(x^k, \boldsymbol{\xi}^k)\right\rangle + \frac{\eta^2(L_0 + L_1\left\|\nabla f(x^k)\right\|)}{2}\left\|G(x^k, \boldsymbol{\xi}^k)\right\|^2. \tag{77}$$

Let us evaluate first summand of equation 77 with $\alpha = \left\|\nabla f(x^k)\right\|^{-1}$:

$$-\eta\left\langle\nabla f(x^k), G(x^k, \boldsymbol{\xi}^k)\right\rangle \stackrel{equation\ 5}{=} -\frac{\alpha\eta}{2}\left\|\nabla f(x^k)\right\|^2 - \frac{\eta}{2\alpha}\left\|G(x^k, \boldsymbol{\xi}^k)\right\|^2$$

$$+ \frac{\eta}{2\alpha}\left\|G(x^k, \boldsymbol{\xi}^k) - \alpha\nabla f(x^k)\right\|^2$$

$$= -\frac{\eta}{2} \left\| \nabla f(x^k) \right\| - \frac{\eta}{2\alpha} \left\| G(x^k, \boldsymbol{\xi}^k) \right\|^2$$
$$+ \frac{\eta}{2\lambda^2\alpha} \left\| \lambda G(x^k, \boldsymbol{\xi}^k) - \lambda\alpha\nabla f(x^k) \right\|^2$$
$$= -\frac{\eta}{2} \left\| \nabla f(x^k) \right\| - \frac{\eta}{2\alpha} \left\| G(x^k, \boldsymbol{\xi}^k) \right\|^2$$
$$+ \frac{\eta}{2\lambda^2\alpha} \left\| \mathrm{clip}_\lambda \left( \nabla f(x^k, \boldsymbol{\xi}^k) \right) - \mathrm{clip}_\lambda \left( \nabla f(x^k) \right) \right\|^2$$

Using that clipping is a projection on onto a convex set, namely ball with radius $\lambda$, and thus is Lipshitz operator with Lipshitz constant 1, we can obtain:

$$-\eta \left\langle \nabla f(x^k), \mathbb{E} \left[ G(x^k, \boldsymbol{\xi}^k) \right] \right\rangle \leq -\frac{\eta}{2} \left\| \nabla f(x^k) \right\| - \frac{\eta}{2\alpha} \mathbb{E} \left[ \left\| G(x^k, \boldsymbol{\xi}^k) \right\|^2 \right]$$
$$+ \frac{\eta}{2\lambda^2\alpha} \mathbb{E} \left[ \left\| \nabla f(x^k, \boldsymbol{\xi}^k) - \nabla f(x^k) \right\|^2 \right]. \qquad (78)$$

Using this in equation 78, we have the following with $\eta_k \leq \frac{\left\| \nabla f(x^k) \right\|}{2(L_0 + L_1 \left\| \nabla f(x^k) \right\|)}$:

$$\mathbb{E} \left[ f(x^{k+1}) \right] - f(x^k) \overset{equation\ 77}{\leq} -\eta \left\langle \nabla f(x^k), \mathbb{E} \left[ G(x^k, \boldsymbol{\xi}^k) \right] \right\rangle + \frac{\eta^2 (L_0 + L_1 \left\| \nabla f(x^k) \right\|)}{2} \mathbb{E} \left[ \left\| G(x^k, \boldsymbol{\xi}^k) \right\|^2 \right]$$
$$\overset{equation\ 78}{\leq} -\frac{\eta}{2} \left\| \nabla f(x^k) \right\| - \frac{\eta}{2\alpha} \mathbb{E} \left[ \left\| G(x^k, \boldsymbol{\xi}^k) \right\|^2 \right] + \frac{\eta}{2\lambda^2\alpha} \mathbb{E} \left[ \left\| \nabla f(x^k, \boldsymbol{\xi}) - \nabla f(x^k) \right\|^2 \right]$$
$$+ \frac{\eta^2 (L_0 + L_1 \left\| \nabla f(x^k) \right\|)}{2} \mathbb{E} \left[ \left\| G(x^k, \boldsymbol{\xi}^k) \right\|^2 \right]$$
$$= -\frac{\eta}{2} \left\| \nabla f(x^k) \right\| + \frac{\eta}{2\lambda^2\alpha} \mathbb{E} \left[ \left\| \nabla f(x^k, \boldsymbol{\xi}^k) - \nabla f(x^k) \right\|^2 \right]$$
$$- \frac{\eta}{2} \mathbb{E} \left[ \left\| G(x^k, \boldsymbol{\xi}^k) \right\|^2 \right] \left( 1 - \frac{\eta(L_0 + L_1 \left\| \nabla f(x^k) \right\|)}{\left\| \nabla f(x^k) \right\|} \right)$$
$$\leq -\frac{\eta}{2} \left\| \nabla f(x^k) \right\| + \frac{\eta\sigma^2}{2\lambda^2\alpha}$$
$$\leq -\frac{\eta}{2} \left\| \nabla f(x^k) \right\| + \frac{\eta}{4} \left\| \nabla f(x^k) \right\|$$
$$= -\frac{\eta}{4} \left\| \nabla f(x^k) \right\|. \qquad (79)$$

The step size will be constant, depending on the hyperparameter $\lambda$:

$$\frac{\left\| \nabla f(x^k) \right\|}{2 \left( L_0 + L_1 \left\| \nabla f(x^k) \right\| \right)} = \frac{1}{2 \left( L_0 \frac{1}{\left\| \nabla f(x^k) \right\|} + L_1 \right)} = \frac{\lambda}{2 \left( L_0 \frac{\lambda}{\left\| \nabla f(x^k) \right\|} + L_1\lambda \right)} \geq \frac{\lambda}{2 \left( L_0 + L_1\lambda \right)}.$$

Thus, $\eta_k = \eta \leq \frac{\lambda}{2(L_0 + L_1\lambda)}$.

We introduce the hyperparameter of the algorithm $R_s = \left\| x^0 - s \right\|$. Then using the convexity assumption of the function, we have the following:

$$f(x^k) - f(s) \leq \left\langle \nabla f(x^k), x^k - s \right\rangle$$
$$\overset{equation\ 6}{\leq} \left\| \nabla f(x^k) \right\| \left\| x^k - s \right\|$$
$$\leq \left\| \nabla f(x^k) \right\| \underbrace{\left\| x^0 - s \right\|}_{R_s}.$$

Hence we have:
$$\left\| \nabla f(x^k) \right\| \geq \frac{f(x^k) - f(s)}{R_s}. \qquad (80)$$

Then substituting equation 80 into equation 79 we obtain:

$$\mathbb{E}\left[f(x^{k+1})\right] - f(x^k) \leq -\frac{\eta}{4}\left\|\nabla f(x^k)\right\| \leq -\frac{\eta}{4R_s}(f(x^k) - f(s)).$$

This inequality is equivalent to the trailing inequality:

$$\mathbb{E}\left[f(x^{k+1})\right] - f^* \leq \left(1 - \frac{\eta}{4R_s}\right)\left(f(x^k) - f^*\right) + \frac{\eta}{4R_s}(f(s) - f^*).$$

Then for $k = 0, 1, 2, ..., N - 1$ iterations that satisfy the conditions $\left\|\nabla f(x^k, \boldsymbol{\xi}^k)\right\| \geq \sqrt{2}\sigma$ and $\left\|\nabla f(x^k)\right\| \geq \sqrt{2}\sigma$ NSGD shows linear convergence:

$$f(x^N) - f^* \leq \left(1 - \frac{\eta}{4R_s}\right)^N (f(x^0) - f^*) + f(s) - f^*.$$

Thus, we have shown that it is indeed possible to converge to a linear rate of convergence on logistic regression using the hyperparameter $R_s$.