# OpenReview forum: "Generalized Smoothness in Stochastic Convex Optimization: First- and Zero-Order Methods"
_ICLR.cc/2026/Conference — Submitted to ICLR 2026_

### Official Review · Reviewer_pXbU · 2025-10-16

**Soundness:** 1
**Presentation:** 1
**Contribution:** 1
**Rating:** 2
**Confidence:** 4

**Summary:**

The paper studies both first-order and zero-order algorithms for stochastic convex optimization under generalized smoothness. The key result is the newly discovered linear term in the convergence rate.

**Strengths:**

The paper is easy to follow.

**Weaknesses:**

This paper has many undesirable points from various perspectives.

**Format**

1. As far as I can see, all citation is based on \citet. However, many places should be changed to \citep.

1. In many places (e.g., Line 038), including the appendix, "equation x" should be either "equation (x)" or "(equation (x))".

1. In numerous places (including the appendix), the articles are missing. For example, line 048 should read "an unbiased gradient oracle". The authors should polish the writing more carefully.

1. Line 294, Anonymous (2025) is not a proper citation since the paper is non-anonymous.

**Theory/Proof**

1. Line 120, assuming a uniform upper bound $M$ on the gradient is too strong, especially when considering a (generalized) smooth problem on the domain $\mathbb{R}^d$. For example, no quadratic optimization can satisfy this assumption.

1. Line 823, $\nabla f(x^{k+1})$ should be $f(x^{k+1})$.

1. The authors use the inequlaity $\\\|x^k-x^* \\\| \leq\\\|x^0-x^*\\\|$ many times (e.g., Line 840). However, I cannot see why.

1. Line 854, I cannot see any formal reason for this recursion step, since Line 852 only holds conditionally. Note that similar steps also has been used many times (e.g., Line 908).

1. Line 1062, I cannot see why this step holds given the previous derivations (especially, considering they contain some errors).

1. Line 1239, it seems $\frac{\eta}{2}$ should be $\frac{\\\|\nabla f(x^k)\\\|\eta}{2}$.

1. Line 1661, $\frac{c}{2}$ should be $\frac{2c}{3}$. Moreover, why can it be assumed $\zeta\leq \frac{c}{3}$?

1. Line 1775, $x^k$ should be $x$. This step also requires $\gamma\leq 1/L_1$.

**Experiments**

1. Line 439, $M$ has been used to denote the uniform upper bound of the gradient norm. Please consider changing it to another letter.

1. Line 444, $L$ should be $\frac{1}{4M}\lambda_{\max}(A^TA)$.

1. Line 449, I think $h$ should be $\gamma$.

1. Line 456, why does $\eta=1/(c\\\|A\\\|_1)$ correspond to your step size? Do the authors mean the upper bound on the step size in Theorem 3.1? If so, please provide the formal calculation. In addition, I cannot find the definition of the $1$-norm $\\\|A\\\|_1$ for a matrix.

1. How do the authors set all hyperparameters for all four of these algorithms? The authors only state partial of them, e.g., the stepsize of ClipSGD.

1. I cannot find how the starting point is initialized.

1. People always preprocess data to make every line of $A$ have a unit length. Did the authors do so? If not, I wonder what will happen after this operation.

1. The experiments also lack error bars. This is a very important index for understanding the performance of any stochastic optimization algorithm.


**Others**

1. Line 047, as far as I know, Gorbunov et al. (2020) didn't contain any experiments about deep learning models. Therefore, placing it here is not proper.

1. The term $(L_0, L_1)$-smoothness first appeared in Line 058 without any reference. Please add a proper reference.

1. For Figure 1, please provide the concrete value of $(L_0, L_1)$ for $\\\|x\\\|^{2n}$.

1. In Table 1, the term ''Maximum Noise Level'' is used without any reference/definition. It is not explained until Section 5. Please explain it earlier or point the reader to Section 5.

1. In Table 1, the notion $\tilde{\sigma}$ is also used without any definition until Theorem 5.2.

1. Line 122, $x^*$ is used without any definition.

1. The authors emphasize the case of $L_0=0$ many times. However, as the authors mentioned in Remark 1.3, the possible function class is very limited. Could the authors provide extra practical examples beyond the exponential and logistic loss?

1. Line 157, to the best of my knowledge, people rarely call the bounded variance condition itself heavy-tailed noise. Instead, heavy-tailed noise often refers to a bounded $\alpha$-th central moment on the noisy gradient for $\alpha\in(1,2)$ (sometimes including 2).

1. The discussion under Theorem 3.1 related to Gaash et al. (2025) is not convincing. Please note that the result of Gaash et al. (2025) is obtained under the existence of an optimal solution in $\mathbb{R}^d$, meaning that one cannot take $L_0=0$.

1. Linie 226, the sentence "The summand..." is not well-written. What does it mean that "The summand" is "a typical SGD"? Please rephrase it.

1. Linies 227 and 232, it should be $\eta=[4(L_0+L_1c)]^{-1}$ according to theory.

1. In Algorithms 2, 3, and 4, $x_0$ should be $x^0$.

1. Line 285, please note that Cutkosky & Mehta (2020) studied non-convex problems and required a large batch size for NSGD. However, this work focused on convex optimization. Therefore, whether a large batch size is indeed required is unclear.

1. Line 721, it is not necessary to emphasize $n\in\\\{2,3\\\}$. The subscript $2$ in $\\\|\\\|_2$ is also redundant.

1. Line 733, the third $\mathbb{E}$ on the R.H.S. is redundant.

1. Line 2304, $R=\mathrm{arginf} f(x)=\infty$ doesn't make sense.

**Questions:**

Please refer to **Weaknesses**.

**Details Of Ethics Concerns:**

Not applicable.

---

> ### Author Response · Authors · 2025-12-03
> **Respose to Reviewer pXbU: Part 1**
>
> Dear **Reviewer pXbU**,
>
> We thank you for your feedback on our work. We provide detailed answers to the comments and questions raised in the review below.
>
> >**As far as I can see, all citation is based on \citet. However, many places should be changed to \citep.**
>
> Thank you, we have corrected it.
>
> >**In many places (e.g., Line 038), including the appendix, "equation x" should be either "equation (x)" or "(equation (x))".**
>
> We are using the standard eqref{} command.
>
> >**In numerous places (including the appendix), the articles are missing. For example, line 048 should read "an unbiased gradient oracle". The authors should polish the writing more carefully.**
>
> Ok.
>
> >**Line 294, Anonymous (2025) is not a proper citation since the paper is non-anonymous.**
>
> Ok.
>
> >**Line 120, assuming a uniform upper bound $M$ on the gradient is too strong, especially when considering a (generalized) smooth problem on the domain $\mathbb{R}^d$. For example, no quadratic optimization can satisfy this assumption.**
>
> We agree that the current assumption is overly strong, so we have relaxed it to $M = \max_{k \in [0,N-1]} || \nabla f(x^{k}) ||$. Moreover, we managed to eliminate this assumption for first-order algorithms by leveraging the boundedness of the gradient norm: $|| \nabla f(x^{k}) || \leq L_0 + \frac{2L_1 + 1}{2} (f(x^k) - f^*)$. Finally, we improved the batch size bounds for NSGD from
> $$
> B = \mathcal{O} \left(  \frac{\sigma^2 M R^3}{\varepsilon^3} \right)
> $$
> to
> $$
> \mathcal{O} \left( \max \left[ \frac{\sigma^2 (4L_1 + 1) R^3 }{\varepsilon^2},  \frac{\sigma^2 L_0 R^3 }{\varepsilon^3} \right] \right).
> $$
> _The improvement is particularly noticeable in the case $L_0 = 0$._ A similar improvement applies to ClipSGD.
>
> >**Line 823, $\nabla f(x^{k+1})$ should be $f(x^{k+1})$.**
>
> Ok.
>
> >**The authors use the inequlaity $||x^k-x^* || \leq||x^0-x^*||$ many times (e.g., Line 840). However, I cannot see why.**
>
> Yes, thank you, we have corrected it. Now we use the notation $R = \max_{k \in [0, N-1]} || x^k - x^* ||$.
>
> >**Line 854, I cannot see any formal reason for this recursion step, since Line 852 only holds conditionally. Note that similar steps also has been used many times (e.g., Line 908).**
>
> We would like to emphasize that we clearly specify the conditions under which convergence is observed.
>
> >**Line 1062, I cannot see why this step holds given the previous derivations (especially, considering they contain some errors).**
>
> Could you please clarify which errors you are referring to?
>
> >**Line 1239, it seems $\frac{\eta}{2}$ should be $\frac{||\nabla f(x^k)||\eta}{2}$.**
>
> Ok.
>
> >**Line 1661, $\frac{c}{2}$ should be $\frac{2c}{3}$. Moreover, why can it be assumed $\zeta\leq \frac{c}{3}$?**
>
> Yes, thank you. This inequality holds because we are assuming low noise (see the bounds on the maximum noise level).
>
> >**Line 1775, $x^k$ should be $x$. This step also requires $\gamma\leq 1/L_1$.**
>
> Ok.
>
> >**Line 439, $M$ has been used to denote the uniform upper bound of the gradient norm. Please consider changing it to another letter.**
>
> Ok.
>
> >**Line 444, $L$ should be $\frac{1}{4M}\lambda_{\max}(A^TA)$.**
>
> Ok.
>
> >**Line 449, I think $h$ should be $\gamma$.**
>
> Ok.
>
> >**Line 456, why does $\eta=1/(c||A||_1)$ correspond to your step size? Do the authors mean the upper bound on the step size in Theorem 3.1? If so, please provide the formal calculation. In addition, I cannot find the definition of the $1$-norm $||A||_1$ for a matrix.**
>
> Okay, we have added it.
>
> >**How do the authors set all hyperparameters for all four of these algorithms? The authors only state partial of them, e.g., the stepsize of ClipSGD.**
>
> Given that the step size for the first-order algorithm coincides with that for the zero-order algorithm, we avoid duplication by not repeating the information. Regarding NSGD, we have added that we used the following step: $\eta = \frac{1}{|| A ||_1}$.
>
> >**I cannot find how the starting point is initialized.**
>
> The initial point was chosen randomly.
>
> >**People always preprocess data to make every line of $A$ have a unit length. Did the authors do so? If not, I wonder what will happen after this operation.**
>
> Yes, of course, we preprocessed the data.
>
> >**The experiments also lack error bars. This is a very important index for understanding the performance of any stochastic optimization algorithm.**
>
> We agree with you; however, this work does not require error bars.
>
> >**Line 047, as far as I know, Gorbunov et al. (2020) didn't contain any experiments about deep learning models. Therefore, placing it here is not proper.**
>
> Ok.
>
> >**The term $(L_0, L_1)$-smoothness first appeared in Line 058 without any reference. Please add a proper reference.**
>
> Ok.

---

> > ### Author Response · Authors · 2025-12-03
> > **Respose to Reviewer pXbU: Part 2**
> >
> > >**For Figure 1, please provide the concrete value of $(L_0, L_1)$ for $||x||^{2n}$.**
> >
> > We did not include the values of the constants because this figure is from the paper Lobanov et al. (2024b), and we do not use this function in our work. By referring to paper Lobanov et al. (2024b), one can find in the experiments section that $L_0 = 2n$ and $L_1 = 2n-1$.
> >
> > >**In Table 1, the term ''Maximum Noise Level'' is used without any reference/definition. It is not explained until Section 5. Please explain it earlier or point the reader to Section 5.**
> >
> > Ok.
> >
> > >**In Table 1, the notion $\tilde{\sigma}$ is also used without any definition until Theorem 5.2.**
> >
> > Ok.
> >
> > >**Line 122, $x^*$ is used without any definition.**
> >
> > Ok.
> >
> > >**The authors emphasize the case of $L_0=0$ many times. However, as the authors mentioned in Remark 1.3, the possible function class is very limited. Could the authors provide extra practical examples beyond the exponential and logistic loss?**
> >
> > We would like to note that the existence of the $L_0=0$ regime, which enables linear convergence in terms of iteration complexity, is surprising in itself. Moreover, the example of logistic regression on the w1a.txt dataset demonstrates that this effect is also observed in ML applications. However, we agree that a larger number of functions satisfying $L_0=0$ would strengthen the significance of our results, but we have left this direction for future work. We would also like to emphasize that, despite the specific case of $L_0 = 0$, the results are novel and quite surprising. Furthermore, to the best of our knowledge, **our work is currently the only result for first-order stochastic optimization under the $(L_0,L_1)$-smoothness condition that provides expectation-based bounds**.
> >
> > >**Line 157, to the best of my knowledge, people rarely call the bounded variance condition itself heavy-tailed noise. Instead, heavy-tailed noise often refers to a bounded $\alpha$-th central moment on the noisy gradient for $\alpha\in(1,2)$ (sometimes including 2).**
> >
> > With all due respect, we believe we have correctly classified this condition.
> >
> > >**The discussion under Theorem 3.1 related to Gaash et al. (2025) is not convincing. Please note that the result of Gaash et al. (2025) is obtained under the existence of an optimal solution in $\mathbb{R}^d$, meaning that one cannot take $L_0=0$.**
> >
> > In defining the generalized smoothness assumption, Gaash et al. (2025) did not impose constraints on $L_0$, thereby allowing for the possibility that the case $L_0 = 0$ may be considered.
> >
> > >**Linie 226, the sentence "The summand..." is not well-written. What does it mean that "The summand" is "a typical SGD"? Please rephrase it.**
> >
> > We would be grateful for an example of rephrasing. However, we do not believe there is any inaccuracy in the wording.
> >
> > >**Linies 227 and 232, it should be $\eta=[4(L_0+L_1c)]^{-1}$ according to theory.**
> >
> > Ok.
> >
> > >**In Algorithms 2, 3, and 4, $x_0$ should be $x^0$.**
> >
> > Ok.
> >
> > >**Line 285, please note that Cutkosky & Mehta (2020) studied non-convex problems and required a large batch size for NSGD. However, this work focused on convex optimization. Therefore, whether a large batch size is indeed required is unclear.**
> >
> > Since in the non-convex setting the results are typically pessimistic (compared to the convex setting), it is quite reasonable to assume that in the convex setting one could replicate the result (and possibly even improve it).
> >
> > >**Line 721, it is not necessary to emphasize $n\in\{2,3\}$. The subscript $2$ in $|| \cdot ||_2$ is also redundant.**
> >
> > Ok.
> >
> > >**Line 733, the third $\mathbb{E}$ on the R.H.S. is redundant.**
> >
> > Ok.
> >
> > >**Line 2304, $R=\mathrm{arginf} f(x)=\infty$ doesn't make sense.**
> >
> > Ok.
> >
> > ***P.S. All changes are highlighted in blue.***
> >
> > With Respect,
> >
> > Authors

---

### Official Review · Reviewer_pVuh · 2025-10-31

**Soundness:** 3
**Presentation:** 3
**Contribution:** 3
**Rating:** 6
**Confidence:** 3

**Summary:**

Strengths
1. Comprehensive generalization under (L_0,L_1)-smoothness: The paper systematically studies several first- and zero-order stochastic optimization methods under the generalized smoothness assumption, offering a unified and well-structured theoretical framework.

2. Novel exponential decrease in convex settings: Demonstrating exponential objective descent when L_0 = 0 is theoretically significant—it challenges the conventional boundary of sublinear convergence in convex optimization.

3. Complete theoretical extension to zero-order methods: Extending the (L_0,L_1)-smooth framework to zero-order algorithms and providing rigorous bounds represents a clear theoretical advancement.

4. Experimental validation supports theory: Logistic-regression experiments are consistent with the theoretical claims and visually confirm the predicted convergence patterns.

Weaknesses
1. Derivations lack detailed intermediate steps: Theorems 3.1, 4.1 omit crucial reasoning chains connecting (L_0,L_1)-smooth conditions with convergence bounds.

2. Batch size assumptions are unrealistic: The required batch size (e.g.,B=O(σ²MR³/ε³) or O(d σ² MR³/ε³)) is computationally infeasible in practical large-scale training.

3. Limited experimental diversity: The evaluation focuses on logistic regression only. Testing on more benchmark functions (e.g., quadratic, hinge loss, or non-smooth proxies) would offer stronger empirical support for the generalized theory.

**Strengths:**

see Summary

**Weaknesses:**

see Summary

**Questions:**

NA

---

> ### Author Response · Authors · 2025-12-03
> **Respose to Reviewer pVuh**
>
> Dear **Reviewer pVuh**,
>
> We thank you for your interest in our work. We attach below detailed responses to the questions raised in the review.
>
> >**Derivations lack detailed intermediate steps: Theorems 3.1, 4.1 omit crucial reasoning chains connecting $(L_0,L_1)$-smooth conditions with convergence bounds.**
>
> With all due respect, we would like to note that the constants$L_0$ and $L_1$ are specified in the theorem statements, namely in the condition for the algorithm's step size. Moreover, given that the clipping radius can be arbitrary and the step size is also a free parameter, precise knowledge of the exact constants $L_0$ and $L_1$ is not required for ClipSGD, as the clipping radius $c$ can compensate for any value.
>
> >**Batch size assumptions are unrealistic: The required batch size (e.g., $B=\mathcal{O} \left(\frac{\sigma^2 L_0 R^3 }{\varepsilon^3}\right)$  or $\mathcal{O} \left(\frac{d \sigma^2 L_0 R^3 }{\varepsilon^3}\right)$ is computationally infeasible in practical large-scale training.**
>
> We agree with you. However, as we mentioned in Section 6, our work focuses on **iteration complexity**. Such a batch size represents the "cost" for achieving a linear rate in iteration complexity. Nevertheless, we do not claim that the current batch size is unimprovable, thus leaving this direction for future research. Moreover, we have managed to improve the batch size $B$ by leveraging an upper bound on the gradient norm:
> $$
> \mathcal{O} \left(  \frac{\sigma^2 M R^3}{\varepsilon^3} \right)
> \Rightarrow
> \mathcal{O} \left( \max \left[ \frac{\sigma^2 (4L_1 + 1) R^3 }{\varepsilon^2},  \frac{\sigma^2 L_0 R^3 }{\varepsilon^3} \right] \right).
> $$
> _The improvement is particularly noticeable in the case $L_0 = 0$._
>
> >**Limited experimental diversity: The evaluation focuses on logistic regression only. Testing on more benchmark functions (e.g., quadratic, hinge loss, or non-smooth proxies) would offer stronger empirical support for the generalized theory.**
>
> Although our work is theoretical, we have provided numerical experiments that support our theoretical results. Nevertheless, we agree to add additional experiments if you recommend them.
>
> ***P.S. All changes are highlighted in blue.***
>
> With Respect,
>
> Authors

---

### Official Review · Reviewer_7MhK · 2025-10-31

**Soundness:** 1
**Presentation:** 2
**Contribution:** 1
**Rating:** 2
**Confidence:** 4

**Summary:**

This paper studies stochastic convex optimization under the **(L₀, L₁)-smoothness** assumption, a generalization of standard Lipschitz smoothness. It analyzes first-order (ClipSGD, NSGD) and zero-order (ZO-ClipSGD, ZO-NSGD) algorithms and claims **exponential objective decrease (linear convergence)** under certain conditions.

The authors provide convergence bounds for both unbiased and biased gradient oracles and extend these results to zero-order methods using gradient approximations. The paper also includes a small experiment on logistic regression (w1a dataset), showing that ZO-NSGD can outperform ClipSGD.

However, the claimed “linear convergence” is **only in iteration count**. Once batch size scaling (e.g., \(B = O(\varepsilon^{-3})\)) is considered, the **total oracle or sample complexity remains sublinear**. Thus, the main result does not represent a true acceleration in computational terms, and the contribution is incremental.

**Strengths:**

- Provides a **unified theoretical analysis** covering both biased and unbiased gradient oracles.
- Improves iteration complexity bounds over prior works (e.g., Gaash et al., 2025) (Theorem 1)
- Clearly structured derivations and comparison table aid understanding.

**Weaknesses:**

- **Misleading main claim:** Linear convergence holds only in terms of iteration count; total computational complexity remains **sublinear** once batch scaling is considered.
- **Marginal novelty:** Extends deterministic analyses (e.g., Lobanov et al., 2024b) to stochastic and zero-order cases without new techniques.
- **Restrictive assumptions:** The \(L₀ = 0\) regime is narrow and not representative of most convex ML problems.
- **Minimal empirical validation:** Only one logistic regression experiment is provided, with no ablations, significance analysis, or comparison to stronger baselines.

**Questions:**

1. When batch size scaling is included, do your total oracle or sample complexities remain linear, or do they revert to sublinear rates?
2. What realistic convex functions beyond logistic regression actually satisfy \(L₀ = 0\)?

---

> ### Author Response · Authors · 2025-12-03
> **Respose to Reviewer 7MhK**
>
> Dear **Reviewer 7MhK**,
>
> We thank you for your feedback on our work. We provide detailed answers to the comments and questions raised in the review below.
>
> >**Misleading main claim: Linear convergence holds only in terms of iteration count; total computational complexity remains sublinear once batch scaling is considered.**
>
> Yes, you are absolutely correct. As we indicated in Section 6, our focus is specifically on iteration complexity. _We have added a clarification that the linear convergence pertains precisely to iteration complexity_.
>
> >**Marginal novelty: Extends deterministic analyses (e.g., Lobanov et al., 2024b) to stochastic and zero-order cases without new techniques.**
>
> We would like to note that, to the best of our knowledge, **our work is currently the only result for first-order stochastic optimization under the $(L_0,L_1)$-smoothness condition that provides expectation-based bounds**.
>
> >**Restrictive assumptions: The (L₀ = 0) regime is narrow and not representative of most convex ML problems.**
>
> We would like to note that the existence of the $L_0 = 0$ regime, which enables linear convergence in terms of iteration complexity, is surprising in itself. Moreover, the example of logistic regression on the w1a.txt dataset demonstrates that this effect is also observed in ML applications. However, we agree that a larger number of functions satisfying $L_0 = 0$ would strengthen the significance of our results, but we have left this direction for future work. **We would also like to note that, despite the specific case of $L_0 = 0$, the results are novel and quite surprising**.
>
> >**Minimal empirical validation: Only one logistic regression experiment is provided, with no ablations, significance analysis, or comparison to stronger baselines.**
>
> Although our work is theoretical, we have provided numerical experiments that support our theoretical results.
>
> >**When batch size scaling is included, do your total oracle or sample complexities remain linear, or do they revert to sublinear rates?**
>
> Yes, as we already mentioned, our work focuses on iteration complexity. Therefore, linear convergence is demonstrated in the iteration rate of (ZO-)NSGD, while the oracle complexity exhibits sublinear convergence. However, we would like to note that we managed to improve the batch size $B$ by leveraging an upper bound on the gradient norm:
> $$
> \mathcal{O} \left(  \frac{\sigma^2 M R^3}{\varepsilon^3} \right)  \Rightarrow  \mathcal{O} \left( \max \left[ \frac{\sigma^2 (4L_1 + 1) R^3 }{\varepsilon^2},  \frac{\sigma^2 L_0 R^3 }{\varepsilon^3} \right] \right).
> $$
> _The improvement is particularly noticeable in the case $L_0 = 0$._
>
> >**What realistic convex functions beyond logistic regression actually satisfy $(L₀ = 0)$?**
>
> Since the observation regarding the $L_0 = 0$ condition has only recently emerged, this question remains open.
>
> ***P.S. All changes are highlighted in blue.***
>
>
> With Respect,
>
> Authors

---

### Official Review · Reviewer_mpiD · 2025-11-02

**Soundness:** 2
**Presentation:** 2
**Contribution:** 2
**Rating:** 2
**Confidence:** 4

**Summary:**

This work studied the convergence rates of ClipSGD and Normalized-SGD for $(L_0, L_1)$-smooth optimization problems, and further studied their zeroth-order variants. The results exhibit two-phase behavior with an exponential decrease followed by a sublinear convergence.

**Strengths:**

1. Generalized smoothness is an useful and important new concept in optimization community, this work devoted efforts into understanding the condition, which should be interesting to the optimization and ML community.
2. The two-phase behavior is interesting, and the theory proposed can match the phenomena, which improves our understanding on the algorithm design.
3. The flow of the work is easy to follow.

**Weaknesses:**

1. But the claimed linear convergence result only arises when $L_0=0$, even though Remark 1.3 mentioned some toy examples fulfilling the assumption, but still it is too restricted (because the condition is proposed for DL applications) in my opinion. It is not clear whether it is still hold for general $L_0$.
2. Line 235, "This iteration complexity significantly outperforms standard results in the L-smoothness setting (Assumption 1.1)", however these two complexities comes from two different scenarios ($L$-smooth and $(0, L_1)$-smooth), therefore the comparison requires additional justification.
3. Finally, in the abstract, "we demonstrate the possibility of the zero-order algorithm outperforming the first-order algorithm in the convex setup through numerical experimentation", but in Section 7, ZO-NSGD (see orange line), ZO-ClipSGD (see blue line) never outperform ClipSGD (see green line) if I understand the figure correctly. Also you said "(ZO-NSGD) outperforming the first-order ClipSGD algorithm after 55000 iterations.", but this crossover is not visible in the figure.

**Questions:**

See above.

---

> ### Author Response · Authors · 2025-12-03
> **Respose to Reviewer mpiD**
>
> Dear **Reviewer mpiD**,
>
> We thank you for your feedback on our work. We provide detailed answers to the comments and questions raised in the review below.
>
>
> >**But the claimed linear convergence result only arises when $L_0=0$, even though Remark 1.3 mentioned some toy examples fulfilling the assumption, but still it is too restricted (because the condition is proposed for DL applications) in my opinion. It is not clear whether it is still hold for general $L_0$.**
>
> We would like to note that the existence of the $L_0 = 0$ regime, which enables linear convergence, is surprising in itself. Moreover, the example of logistic regression on the w1a.txt dataset demonstrates that the effect is also observed for ML applications. However, we agree that a larger number of functions satisfying $L_0 = 0$ would strengthen the importance of our results, but we have left this direction for future work. We would also like to emphasize that, despite the specific case of $L_0 = 0$, the results are novel and quite surprising. Furthermore, to the best of our knowledge, **our work is currently the only result for first-order stochastic optimization under the $(L_0,L_1)$-smoothness condition that provides expectation-based bounds**.
>
> >**Line 235, "This iteration complexity significantly outperforms standard results in the L-smoothness setting (Assumption 1.1)", however these two complexities comes from two different scenarios ($L$-smooth and $(0, L_1)$-smooth), therefore the comparison requires additional justification.**
>
> Given that there are functions that satisfy both $L$-smoothness and $(L_0,L_1)$-smoothness (for example, the Logistic function), the comparison of results is entirely justified.
>
> >**Finally, in the abstract, "we demonstrate the possibility of the zero-order algorithm outperforming the first-order algorithm in the convex setup through numerical experimentation", but in Section 7, ZO-NSGD (see orange line), ZO-ClipSGD (see blue line) never outperform ClipSGD (see green line) if I understand the figure correctly. Also you said "(ZO-NSGD) outperforming the first-order ClipSGD algorithm after 55000 iterations.", but this crossover is not visible in the figure.**
>
> Yes, we agree that the superiority is not explicitly demonstrated in Figure 2. However, we observe the convergence dynamics, particularly that ZO-NSGD indeed exhibits linear convergence (slower than NSGD, since it is a zero-order algorithm, but still linear), while ClipSGD mainly converges at a sublinear rate. Therefore, based on the presented dynamics, we assumed that it is possible that ZO-NSGD could outperform ClipSGD after 55,000 iterations. _We have emphasized that we are suggesting a possibility based on the theoretical bounds and the numerical results of Figure 2._
>
> ***P.S. All changes are highlighted in blue.***
>
> With Respect,
>
> Authors

---

### Author Response · Authors · 2025-12-03
**Rebuttal Summary**

Dear **All Reviewers**,

We would like to thank you for taking the time to prepare the reviews and for the interest you have shown in our work! We are pleased that each of the reviewers expressed positive feedback on our paper:

- **Reviewer mpiD** noted: “The two-phase behavior is interesting, and the theory proposed can match the phenomena, which improves our understanding on the algorithm design”;

- **Reviewer 7MhK** emphasized: “Provides a unified theoretical analysis covering both biased and unbiased gradient oracles”;

- **Reviewer pVuh** highlighted: “The paper systematically studies several first- and zero-order stochastic optimization methods under the generalized smoothness assumption, offering a unified and well-structured theoretical framework”;

- **Reviewer pXbU** pointed out: “The paper is easy to follow”;

This is truly important to us!

After carefully studying your reviews, we have prepared detailed responses, which are provided below the corresponding review. Furthermore, based on the feedback received from you, we have managed to improve our work (for which we sincerely thank you). In particular, we would like to note that in the revised version of the paper, we have relaxed the assumption on $M$. Now $M = \max_{k \in [0,N-1]} || \nabla f(x^{k}) ||$. Moreover, we managed to eliminate this assumption for first-order algorithms by leveraging the boundedness of the gradient norm: $|| \nabla f(x^{k}) || \leq L_0 + \frac{2L_1 + 1}{2} (f(x^k) - f^*)$. Finally, we improved the batch size bounds for NSGD from $$ B = \mathcal{O} \left(  \frac{\sigma^2 M R^3}{\varepsilon^3} \right) $$ to $$ \mathcal{O} \left( \max \left[ \frac{\sigma^2 (4L_1 + 1) R^3 }{\varepsilon^2},  \frac{\sigma^2 L_0 R^3 }{\varepsilon^3} \right] \right). $$ _The improvement is particularly noticeable in the case $L_0 = 0$._ A similar improvement applies to ClipSGD.

In conclusion, we would like to thank you once again. We believe we have addressed all doubts and questions raised in the reviews. If this is not the case, please let us know—we would be happy to provide additional clarifications regarding our work.

***P.S. All changes are highlighted in blue.***

With Respect,

Authors

---

### Meta-Review · Area_Chair_ygZa · 2026-01-01

**Summary:**

This work shows convergence of stochastic first and zeroth order methods under convexity and the $(L_0,L_1)$-smoothness assumption. As the deterministic case has been studied, some reviewer find the contribution to be limited. The proposed algorithm has a required batch size to guarantee convergence, which is criticized by several reviewers to be less practical.

**Reviewer Concerns:**

A reviewer points out that assuming a universal bound $M\geq ||\nabla f(x)||$ is too strong for generalized smooth functions. The authors agree, but then modify this assumption to $M = \max_{i=1,\ldots,N} ||\nabla f(x^i)||$. This isn't a fix at all.

**Reviewer Scores:**

None is likely to change.

---

### Decision · Program_Chairs · 2026-01-26

Reject